# Expanding Small-Scale Datasets with Guided Imagination

**Yifan Zhang**[1*]  **Daquan Zhou**[2*]  **Bryan Hooi**[1]  **Kai Wang**[1]  **Jiashi Feng**[2]

[1]National University of Singapore    [2]ByteDance

## Abstract

The power of DNNs relies heavily on the quantity and quality of training data. However, collecting and annotating data on a large scale is often expensive and time-consuming. To address this issue, we explore a new task, termed *dataset expansion*, aimed at expanding a ready-to-use small dataset by automatically creating new labeled samples. To this end, we present a Guided Imagination Framework (GIF) that leverages cutting-edge generative models like DALL-E2 and Stable Diffusion (SD) to "imagine" and create informative new data from the input seed data. Specifically, GIF conducts data imagination by optimizing the latent features of the seed data in the semantically meaningful space of the prior model, resulting in the creation of photo-realistic images with *new* content. To guide the imagination towards creating informative samples for model training, we introduce two key criteria, *i.e., class-maintained information boosting* and *sample diversity promotion*. These criteria are verified to be essential for effective dataset expansion: GIF-SD obtains 13.5% higher model accuracy on natural image datasets than unguided expansion with SD. With these essential criteria, GIF successfully expands small datasets in various scenarios, boosting model accuracy by 36.9% on average over six natural image datasets and by 13.5% on average over three medical datasets. The source code is available at https://github.com/Vanint/DatasetExpansion.

## 1  Introduction

A substantial number of training samples is essential for unleashing the power of deep networks [14]. However, such requirements often impede small-scale data applications from fully leveraging deep learning solutions. Manual collection and labeling of large-scale datasets are often expensive and time-consuming in small-scale scenarios [52]. To address data scarcity while minimizing costs, we explore a novel task, termed **Dataset Expansion**. As depicted in Figure 1 (left), dataset expansion aims to create an automatic data generation pipeline that can expand a small dataset into a larger and more informative one for model training. This task particularly focuses on enhancing the quantity and quality of the small-scale dataset by creating informative new samples. This differs from conventional data augmentations that primarily focus on increasing data size through transformations, often without creating samples that offer fundamentally new content. The expanded dataset is expected to be versatile, fit for training various network architectures, and promote model generalization.

We empirically find that naive applications of existing methods cannot address the problem well (cf. Table 1 and Figure 4). Firstly, data augmentation [12, 15, 109], involving applying pre-set transformations to images, can be used for dataset expansion. However, these transformations primarily alter the surface visual characteristics of an image, but cannot create images with novel content (cf. Figure 5a). As a result, the new information introduced is limited, and the performance gains tend to saturate quickly as more data is generated. Secondly, we have explored pre-trained generative models (*e.g.,* Stable Diffusion (SD) [59]) to create images for model training. However,

---

*Corresponding to: yifan.zhang@u.nus.edu, zhoudaquan21@gmail.com

37th Conference on Neural Information Processing Systems (NeurIPS 2023).

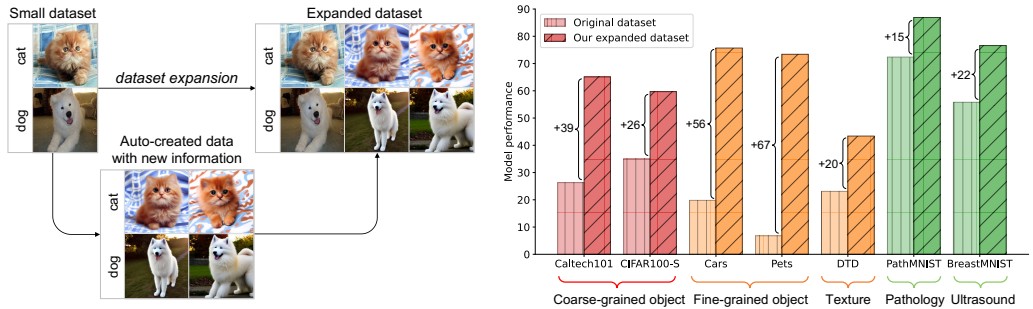

Figure 1: **Dataset Expansion** aims to create data with *new* information to enrich small datasets for training DNNs better (left). As shown, the created images by our method are all class-relevant but diverse (*e.g.,* new sitting postures of cats, new patterns of cushions). This enables ResNet-50 trained on our expanded datasets to perform much better than the one trained on the original datasets (right).

these pre-trained generative models are usually category-agnostic to the target dataset, so they cannot ensure that the synthetic samples carry the correct labels and are beneficial to model training.

Different from them, our solution is inspired by human learning with imagination. Upon observing an object, humans can readily imagine its different variants in various shapes, colors, or contexts, relying on their prior understanding of the world [75, 80]. Such an imagination process is highly valuable for dataset expansion as it does not merely tweak an object's appearance, but leverages rich prior knowledge to create object variants infused with new information. In tandem with this, recent generative models like SD and DALL-E2 [56] have demonstrated exceptional abilities in capturing the data distribution of extremely vast datasets [4, 66] and generating photo-realistic images with rich content. This motivates us to explore their use as prior models to establish our computational data imagination pipeline for dataset expansion, *i.e.,* imagining different sample variants given seed data. However, the realization of this idea is non-trivial and is complicated by two key challenges: how to generate samples with correct labels, and how to ensure the created samples boost model training.

To handle these challenges, we conduct a series of exploratory studies (cf. Section 3) and make two key findings. First, CLIP [54], which offers excellent zero-shot classification abilities, can map latent features of category-agnostic generative models to the specific label space of the target dataset. This enables us to generate samples with correct labels. Second, we discover two informativeness criteria that are crucial for generating effective training data: 1) *class-maintained information boosting* to ensure that the imagined data are class-consistent with the seed data and bring new information; 2) *sample diversity promotion* to encourage the imagined samples to have diversified content.

In light of the above findings, we propose a **Guided Imagination Framework** (GIF) for dataset expansion. Specifically, given a seed image, GIF first extracts its latent feature with the prior (generative) model. Unlike data augmentation that imposes variation over raw images, GIF optimizes the "variation" over latent features. Thanks to the guidance carefully designed by our discovered criteria, the latent feature is optimized to provide more information while maintaining its class. This enables GIF to create informative new samples, with class-consistent semantics yet higher content diversity, for dataset expansion. Such expansion is shown to benefit model generalization.

As DALL-E2 [56] and SD are powerful in generating images, and MAE [24] excels at reconstructing images, we explore their use as prior models of GIF for data imagination. We evaluate our methods on small-scale natural and medical image datasets. As shown in Figure 1 (right), compared to ResNet-50 trained on the original dataset, our method improves its performance by a large margin across various visual tasks. Specifically, with the designed guidance, GIF obtains 36.9% accuracy gains on average over six natural image datasets and 13.5% gains on average over three medical datasets. Moreover, the expansion efficiency of GIF is much higher than data augmentation, *i.e.,* 5× expansion by GIF-SD outperforms even 20× expansion by Cutout, GridMask and RandAugment on Cars and DTD datasets. In addition, the expanded datasets also benefit out-of-distribution performance of models, and can be directly used to train various architectures (*e.g.,* ResNeXt [85], WideResNet [97], and MobileNet [64]), leading to consistent performance gains. We also empirically show that GIF is more applicable than CLIP to handle real small-data scenarios, particularly with non-natural image domains (*e.g.,* medicine) and hardware constraints (*e.g.,* limited supportable model sizes). Please note that GIF is much faster and more cost-effective than human data collection for dataset expansion.

## 2 Related Work

**Learning with synthetic images.** Training with synthetic data is a promising direction [26, 34, 110]. For example, DatasetGANs [42, 105] explore GAN models [17, 33] to generate images for segmentation model training. However, they require a sufficiently large dataset for in-domain GAN training, which is not feasible in small-data scenarios. Also, as the generated images are without labels, they need manual annotations on generated images to train a label generator for annotating synthetic images. Similarly, many recent studies [2, 3, 22, 39, 63, 78, 82, 94] also explored generative models to generate new data for model training. However, these methods cannot ensure that the synthesized data bring sufficient new information and accurate labels for the target small datasets. Moreover, training GANs from scratch [3, 39, 63, 78, 94], especially with very limited data, often fails to converge or produce meaningful results. In contrast, our dataset expansion aims to expand a small dataset into a larger labeled one in a fully automatic manner without involving human annotators. As such, our method emerges as a more effective way to expand small datasets.

**Data augmentation.** Augmentation employs manually specified rules, such as image manipulation [91], erasing [15, 109], mixup [28, 100, 101], and transformation selection [11, 12] to boost model generalization [69]. Despite certain benefits, these methods enrich images by pre-defined transformations, which only locally vary the pixel values of images and cannot generate images with highly diversified content. Moreover, the effectiveness of augmented data is not always guaranteed due to random transformations. In contrast, our approach harnesses generative models trained on large datasets and guides them to generate more informative and diversified images, thus resulting in more effective and efficient dataset expansion. For additional related studies, please refer to Appendix A.

## 3 Problem and Preliminary Studies

**Problem statement**. To address data scarcity, we explore a novel *dataset expansion* task. Without loss of generality, we consider image classification, where a small-scale training dataset $\mathcal{D}_o = \{x_i, y_i\}_{i=1}^{n_o}$ is given. Here, $x_i$ denotes an instance with class label $y_i$, and $n_o$ denotes the number of samples. The goal of dataset expansion is to generate a set of new synthetic samples $\mathcal{D}_s = \{x'_j, y'_j\}_{j=1}^{n_s}$ to enlarge the original dataset, such that a DNN model trained on the expanded dataset $\mathcal{D}_o \cup \mathcal{D}_s$ outperforms the model trained on $\mathcal{D}_o$ significantly. The key is that the synthetic sample set $\mathcal{D}_s$ should be highly related to the original dataset $\mathcal{D}_o$ and *bring sufficient new information* to boost model training.

### 3.1 A proposal for computational imagination models

Given an object, humans can easily imagine its different variants, like the object in various colors, shapes, or contexts, based on their accumulated prior knowledge about the world [75, 80]. This imagination process is highly useful for dataset expansion, as it does not simply perturb the object's appearance but applies rich prior knowledge to create variants with new information. In light of this, we seek to build a computational model to simulate this imagination process for dataset expansion.

Deep generative models, known for their capacity to capture the distribution of a dataset, become our tool of choice. By drawing on their prior distribution knowledge, we can generate new samples resembling the characteristics of their training datasets. More importantly, recent generative models, such as Stable Diffusion (SD) and DALL-E2, have demonstrated remarkable abilities in capturing the distribution of extremely large datasets and generating photo-realistic images with diverse content. This inspires us to explore their use as prior models to construct our data imagination pipeline.

Specifically, given a pre-trained generative model $G$ and a seed example $(x, y)$ from the target dataset, we formulate the imagination of $x'$ from $x$ as $x' = G(f(x) + \delta)$. Here, $f(\cdot)$ is an image encoder of $G$ to transform the raw image into an embedding for imagination with the generative model. $\delta$ is a perturbation applied to $f(x)$ such that $G$ can generate $x'$ different from $x$. A simple choice of $\delta$ would be Gaussian random noise, which, however, cannot generate highly informative samples. In the following subsection, we will discuss how to optimize $\delta$ to provide useful guidance.

It is worth noting that we do not aim to construct a biologically plausible imagination model that strictly follows the dynamics and rules of the human brain. Instead, we draw inspiration from the imaginative activities of the human brain and propose a pipeline to leverage well pre-trained generative models to explore dataset expansion.

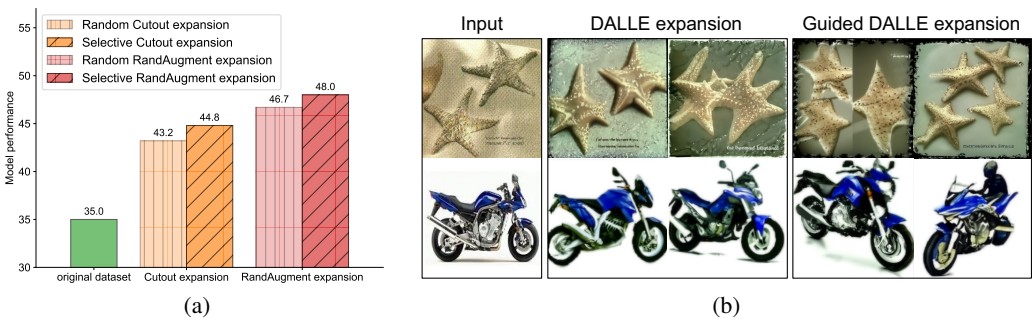

Figure 2: **Effectiveness of the informativeness criteria for sample creation**. (a) Comparison between random expansion and our selective expansion on CIFAR100-Subset. (b) Visualization of DALLE expansion with and without diversity guidance.

## 3.2 How to guide imagination for effective expansion?

Our data imagination pipeline leverages generative models to create new samples from seed data. However, it is unclear what types of samples are effective and how to optimize $\delta$ accordingly in the pipeline to create data that are useful for model training. Our key insight is that the newly created sample $x'$ should *introduce new information compared to the seed sample x, while preserving the same class semantics as the seed sample*. To achieve these properties, we explore the following preliminary studies and discover two key criteria: (1) class-maintained information boosting, and (2) sample diversity promotion.

**Class-maintained informativeness boosting**. When enhancing data informativeness, it is non-trivial to ensure that the generated $x'$ has the same label $y$ as the seed sample $x$, since it is difficult to maintain the class semantics after perturbation in the latent space $f(x)$. To overcome this, we explore CLIP [54] for its well-known image-text alignment ability: CLIP's image encoder can project an image $x$ to an embedding space aligned with the language embedding of its class name $y$ [72, 81]. Therefore, we can leverage CLIP's embedding vectors of all class names as a zero-shot classifier to guide the generation of samples that maintain the same class semantics as seed data. Meanwhile, the entropy of the zero-shot prediction can serve as a measure to boost the classification informativeness of the generated data.

To pinpoint whether the criteria of class-maintained information boosting helps to generate more informative samples, we start with exploratory experiments on a subset of CIFAR100 [41]. Here, the subset is built for simulating small-scale datasets by randomly sampling 100 instances per class from CIFAR100. We first synthesize samples based on existing data augmentation methods (*i.e.,* RandAugment and Cutout [15]) and expand CIFAR100-Subset by 5×. Meanwhile, we conduct selective augmentation expansion based on our criteria (*i.e.,* selecting the samples with the same zero-shot prediction but higher prediction entropy compared to seed samples) until we reach the required expansion ratio per seed sample. As shown in Figure 2a, selective expansion outperforms random expansion by 1.3% to 1.6%, meaning that the selected samples are more informative for model training. Compared to random augmentation, selective expansion filters out the synthetic data with lower prediction entropy and those with higher entropy but inconsistent classes. The remaining data thus preserve the same class semantics but bring more information gain, leading to better expansion effectiveness.

**Sample diversity promotion.** To prevent the "imagination collapse" issue that generative models yield overly similar or duplicate samples, we delve further into the criterion of sample diversity promotion. To study its effectiveness, we resort to a powerful generative model (*i.e.,* DALL-E2) as the prior model to generate images and expand CIFAR100-Subset by 5×, where the guided expansion scheme and the implementation of diversity promotion will be introduced in the following section. As shown in Figure 2b, the generated images with diversity guidance are more diversified: starfish images have more diverse object numbers, and motorbike images have more diverse angles of view and even a new driver. This leads to 1.4% additional accuracy gains on CIFAR100-Subset (cf. Table 20 in Appendix F.5), demonstrating that the criterion of sample diversity promotion is effective in bringing diversified information to boost model training.

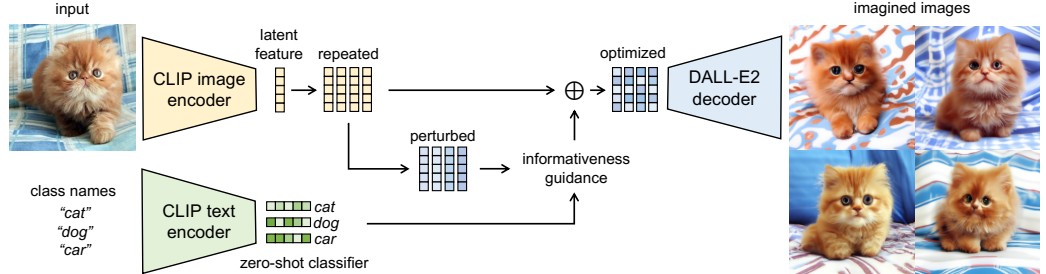

Figure 3: Illustration of the proposed GIF method based on DALL-E2 [56], which expands small datasets by creating informative new samples with guided imagination. Here, we resort to DALL-E2 as the prior model, in which the image/text encoders are CLIP image/text encoders while the decoder is the diffusion model of DALL-E2. Moreover, $\oplus$ denotes guided residual multiplicative perturbation. More implementation details of GIF-DALLE, GIF-SD and GIF-MAE are provided in Appendix D.

## 4 GIF: A Guided Imagination Framework for Dataset Expansion

In light of the aforementioned studies, we propose a Guided Imagination Framework (GIF) for dataset expansion. This framework guides the imagination of prior generative models based on the identified criteria. Given a seed image $x$ from a target dataset, we first extract its latent feature $f(x)$ via the encoder of the generative model. Different from data augmentation that directly imposes variations over raw RGB images, GIF optimizes the "variations" over latent features. Thanks to the aforementioned criteria as guidance, the optimized latent features result in samples that preserve the correct class semantics while introducing new information beneficial for model training.

**Overall pipeline**. To detail the framework, we use DALL-E2 [56] as a prior generative model for illustration. As shown in Figure 3, DALL-E2 adopts CLIP image/text encoders $f_{\text{CLIP-I}}$ and $f_{\text{CLIP-T}}$ as its image/text encoders and uses a pre-trained diffusion model $G$ as its image decoder. To create a set of new images $x'$ from the seed image $x$, GIF first repeats its latent feature $f = f_{\text{CLIP-I}}(x)$ for $K$ times, with $K$ being the expansion ratio. For each latent feature $f$, we inject perturbation over it with randomly initialized noise $z \sim \mathcal{U}(0, 1)$ and bias $b \sim \mathcal{N}(0, 1)$. Here, to prevent out-of-control imagination, we conduct residual multiplicative perturbation on the latent feature $f$ and enforce an $\varepsilon$-ball constraint on the perturbation as follows:

$$f' = \mathcal{P}_{f,\epsilon}((1+z)f + b), \tag{1}$$

where $\mathcal{P}_{f,\epsilon}(\cdot)$ means to project the perturbed feature $f'$ to the $\varepsilon$-ball of the original latent feature, *i.e.,* $\|f' - f\|_\infty \le \varepsilon$. Note that each latent feature has independent $z$ and $b$. Following our explored criteria, GIF optimizes $z$ and $b$ over the latent feature space as follows:

$$z', b' \leftarrow \underset{z,b}{\arg\max} \ \mathcal{S}_{inf} + \mathcal{S}_{div}, \tag{2}$$

where $\mathcal{S}_{inf}$ and $\mathcal{S}_{div}$ correspond to the class-maintained informativeness score and the sample diversity score, respectively, which will be elaborated below. This latent feature optimization is the key step for achieving guided imagination. After updating the noise $z'$ and bias $b'$ for each latent feature, GIF obtains a set of new latent features by Eq. (1), which are then used to create new samples through the decoder $G$.

**Class-maintained informativeness**. To boost the informativeness of the generated data without changing class labels, we resort to CLIP's zero-shot classification abilities. Specifically, we first use $f_{\text{CLIP-T}}$ to encode the class name $y$ of a sample $x$ and take the embedding $w_y = f_{\text{CLIP-T}}(y)$ as the zero-shot classifier of class $y$. Each latent feature $f_{\text{CLIP-I}}(x)$ can be classified according to its cosine similarity to $w_y$, *i.e.,* the affinity score of $x$ belonging to class $y$ is $\hat{s}_y = \cos(f(x), w_y)$, which forms a prediction probability vector $s = \sigma([\hat{s}_1, \ldots, \hat{s}_C])$ for the total $C$ classes of the target dataset based on softmax $\sigma(\cdot)$. The prediction of the perturbed feature $s'$ can be obtained in the same way. We then design $\mathcal{S}_{inf}$ to improve the information entropy of the perturbed feature while maintaining its class semantics as the seed sample:

$$\mathcal{S}_{inf} = s'_j + (s\log(s) - s'\log(s')), \ \ \text{s.t.} \ \ j = \arg\max(s).$$

Specifically, $s'_j$ denotes the zero-shot classification score of the perturbed feature $s'$ regarding the predicted label of the original latent feature $j = \arg\max(s)$. Here, the zero-shot prediction of the

original data serves as an anchor to regularize the class semantics of the perturbed features in CLIP's embedding space, thus encouraging class consistency between the generated samples and the seed sample. Moreover, $s \log(s) - s' \log(s')$ means contrastive entropy increment, which encourages the perturbed feature to have higher prediction entropy and helps to improve the classification informativeness of the generated image.

**Sample diversity**. To promote the diversity of the generated samples, we design $\mathcal{S}_{div}$ as the Kullback–Leibler (KL) divergence among all perturbed latent features of a seed sample: $\mathcal{S}_{div} = \mathcal{D}_{KL}(f' \| \bar{f})$, where $f'$ denotes the current perturbed latent feature and $\bar{f}$ indicates the mean over the $K$ perturbed features of the seed sample. To enable measuring KL divergence between features, inspired by [84], we apply the softmax function to transform feature vectors into probability vectors for KL divergence.

**Theoretical analysis**. We then analyze our method to conceptually understand its benefits to model generalization. We resort to $\delta$-cover [62] to analyze how data diversity influences the generalization error bound. Specifically, "a dataset $E$ is a $\delta$-cover of a dataset $S$" means a set of balls with radius $\delta$ centered at each sample of the dataset $E$ can cover the entire dataset $S$. According to the property of $\delta$-cover, we define the dataset diversity by $\delta$-diversity as the inverse of the minimal $\delta_{min}$, *i.e.,* $\delta_{div} = \frac{1}{\delta_{min}}$. Following the same assumptions of [67], we have the following result.

**Theorem 4.1.** *Let $A$ denote a learning algorithm that outputs a set of parameters given a dataset $\mathcal{D} = \{x_i, y_i\}_{i \in [n]}$ with $n$ i.i.d. samples drawn from distribution $\mathcal{P}_\mathcal{Z}$. Assume the hypothesis function is $\lambda^\eta$-Lipschitz continuous, the loss function $\ell(x, y)$ is $\lambda^\ell$-Lipschitz continuous for all $y$, and is bounded by $L$, with $\ell(x_i, y_i; A) = 0$ for all $i \in [n]$. If $\mathcal{D}$ constitutes a $\delta$-cover of $\mathcal{P}_\mathcal{Z}$, then with probability at least $1 - \gamma$, the generalization error bound satisfies:*

$$\left| \mathbb{E}_{x,y \sim \mathcal{P}_\mathcal{Z}}[\ell(x, y; A)] - \frac{1}{n} \sum_{i \in [n]} \ell(x_i, y_i; A) \right| \overset{c}{\leq} \frac{\lambda^\ell + \lambda^\eta LC}{\delta_{div}},$$

*where $C$ is a constant, and the symbol $\overset{c}{\leq}$ indicates "smaller than" up to an additive constant.*

Please refer to Appendix C for proofs. This theorem shows that the generalization error is bounded by the inverse of $\delta$-diversity. That is, the more diverse samples are created, the more improvement of generalization performance would be made in model training. In real small-data applications, data scarcity leads the covering radius $\delta$ to be very large and thus the $\delta$-diversity is low, which severely affects model generalization. Simply increasing the data number (*e.g.,* via data repeating) does not help generalization since it does not increase $\delta$-diversity. In contrast, our GIF applies two key criteria (*i.e.,* "informativeness boosting" and "sample diversity promotion") to create informative and diversified new samples. The expanded dataset thus has higher data diversity than random augmentation, which helps to increase $\delta$-diversity and thus boosts model generalization. This advantage can be verified by Table 2 and Figure 4.

**Implementing GIF with different prior models**. To enable effective expansion, we explore three prior models for guided imagination: DALL-E2 [56] and Stable Diffusion [59] are advanced image generative methods, while MAE [24] is skilled at reconstructing images. We call the resulting methods GIF-DALLE, GIF-SD, and GIF-MAE, respectively. We introduce their high-level ideas below, while their method details are provided in Appendix D.

GIF-DALLE adheres strictly to the above pipeline for guided imagination, while we slightly modify the pipeline in GIF-SD and GIF-MAE since their image encoders are different from the CLIP image encoder. Given a seed sample, GIF-SD and GIF-MAE first generate the latent feature via the encoders of their prior models, and then conduct random noise perturbation following Eq. (1). Here, GIF-SD has one more step than GIF-MAE before noise perturbation, *i.e.,* conducting prompt-guided diffusion for its latent feature, where the rule of the prompt design will be elaborated in Appendix D.2. Based on the perturbed feature, GIF-SD and GIF-MAE generate an intermediate image via their decoders, and apply CLIP to conduct zero-shot predictions for both the seed and intermediate images to compute the informativeness guidance (*i.e.,* Eq. (2)) for optimizing latent features. Note that, in GIF-SD and GIF-MAE, we execute *channel-level* noise perturbation since we find it facilitates the generation of content-consistent samples with a greater variety of image styles (cf. Appendix B.2).

Table 1: Accuracy of ResNet-50 trained from scratch on small datasets and their expanded datasets by various methods. Here, CIFAR100-Subset is expanded by 5×, Pets is expanded by 30×, and all other natural image datasets are expanded by 20×. All medical image datasets are expanded by 5×. Moreover, MAE, DALL-E2 and SD (Stable Diffusion) are the baselines of directly using them to expand datasets without our GIF. In addition, CLIP indicates its zero-shot performance. All performance values are averaged over three runs. Please see Appendix F for more comparisons.

| Dataset | Natural image datasets | | | | | | | Medical image datasets | | | |
|---|---|---|---|---|---|---|---|---|---|---|---|
| | Caltech101 | Cars | Flowers | DTD | CIFAR100-S | Pets | Average | PathMNIST | BreastMNIST | OrganSMNIST | Average |
| *Original* | 26.3 | 19.8 | 74.1 | 23.1 | 35.0 | 6.8 | 30.9 | 72.4 | 55.8 | 76.3 | 68.2 |
| CLIP | 82.1 | 55.8 | 65.9 | 41.7 | 41.6 | 85.4 | 62.1 | 10.7 | 51.8 | 7.7 | 23.4 |
| Distillation of CLIP | 33.2 | 18.9 | 75.1 | 25.6 | 37.8 | 11.1 | 33.6 | 77.3 | 60.2 | 77.4 | 71.6 |
| *Expanded* | | | | | | | | | | | |
| Cutout [15] | 51.5 | 25.8 | 77.8 | 24.2 | 44.3 | 38.7 | 43.7 (+12.8) | 78.8 | 66.7 | 78.3 | 74.6 (+6.4) |
| GridMask [8] | 51.6 | 28.4 | 80.7 | 25.3 | 48.2 | 37.6 | 45.3 (+14.4) | 78.4 | 66.8 | 78.9 | 74.7 (+6.5) |
| RandAugment [12] | 57.8 | 43.2 | 83.8 | 28.7 | 46.7 | 48.0 | 51.4 (+20.5) | 79.2 | 68.7 | 79.6 | 75.8 (+7.6) |
| MAE [24] | 50.6 | 25.9 | 76.3 | 27.6 | 44.3 | 39.9 | 44.1 (+13.2) | 81.7 | 63.4 | 78.6 | 74.6 (+6.4) |
| DALL-E2 [56] | 61.3 | 48.3 | 84.1 | 34.5 | 52.1 | 61.7 | 57.0 (+26.1) | 82.8 | 70.8 | 79.3 | 77.6 (+9.4) |
| SD [59] | 51.1 | 51.7 | 78.8 | 33.2 | 52.9 | 57.9 | 54.3 (+23.4) | 85.1 | 73.8 | 78.9 | 79.3 (+11.1) |
| GIF-MAE (ours) | 58.4 | 44.5 | 84.4 | 34.2 | 52.7 | 52.4 | 54.4 (+23.5) | 82.0 | 73.3 | 80.6 | 78.6 (+10.4) |
| GIF-DALLE (ours) | 63.0 | 53.1 | 88.2 | 39.5 | 54.5 | 66.4 | 60.8 (+29.9) | 84.4 | 76.6 | 80.5 | 80.5 (+12.3) |
| GIF-SD (ours) | **65.1** | **75.7** | **88.3** | **43.4** | **61.1** | **73.4** | **67.8** (+36.9) | **86.9** | **77.4** | **80.7** | **81.7** (+13.5) |

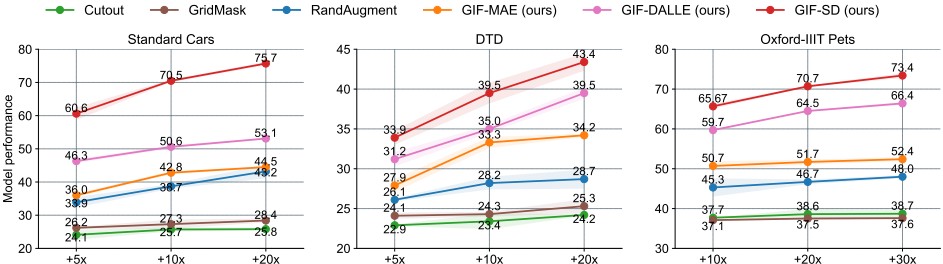

Figure 4: Accuracy of ResNet-50 trained from scratch on the expanded datasets with different expansion ratios. The results on other datasets are reported in Appendix F.1.

# 5 Experiments

**Datasets**. We evaluate GIF on six small-scale natural image datasets and three medical datasets. Natural datasets cover a variety of tasks: object classification (Caltech-101 [18], CIFAR100-Subset [41]), fine-grained classification (Cars [40], Flowers [49], Pets [50]) and texture classification (DTD [10]). Here, CIFAR100-Subset is an artificial dataset for simulating small-scale datasets by randomly sampling 100 instances per class from the original CIFAR100. Moreover, medical datasets [90] cover a wide range of image modalities, such as breast ultrasound (BreastMNIST), colon pathology (PathMNIST), and Abdominal CT (OrganSMNIST). Please refer to Appendix E for data statistics.

**Compared methods**. As there is no algorithm devoted to dataset expansion, we take representative data augmentation methods as baselines, including RandAugment, Cutout, and GridMask [8]. We also compare to directly using prior models (*i.e.,* DALL-E2, SD, and MAE) for dataset expansion. Besides, CLIP has outstanding zero-shot abilities, and some recent studies explore distilling CLIP to facilitate model training. Hence, we also compare to zero-shot prediction and knowledge distillation (KD) of CLIP on the target datasets. The implementation details of GIF are provided in Appendix D.

## 5.1 Results of small-scale dataset expansion

**Expansion effectiveness.** As shown in Table 1, compared with the models trained on original datasets, GIF-SD boosts their accuracy by an average of 36.9% across six natural image datasets and 13.5% across three medical datasets. This verifies the superior capabilities of GIF over other methods for enhancing small datasets, particularly in the data-starved field of medical image understanding. This also suggests that dataset expansion is a promising direction for real small-data applications. Here, the reason why GIF-SD outperforms GIF-DALLE is that GIF-DALLE only exploits the image-to-image variation ability of DALL-E2, while GIF-SD further applies text prompts to diversify samples.

**Expansion efficiency.** Our GIF is more *sample efficient* than data augmentations, in terms of the accuracy gain brought by each created sample. As shown in Figure 4, 5× expansion by GIF-SD outperforms even 20× expansion by various data augmentations on Cars and DTD datasets, implying

Table 2: Corruption accuracy of ResNet-50 trained from scratch on CIFAR100-S and our $5\times$ expanded dataset, under 15 types of corruption in CIFAR100-C with the severity level 3. More results regarding other severity levels are provided in Appendix F.2.

| Dataset | Noise | | | Blur | | | | Weather | | | | Digital | | | | Average |
|---|---|---|---|---|---|---|---|---|---|---|---|---|---|---|---|---|
| | Gauss. | Shot | Impul. | Defoc. | Glass | Motion | Zoom | Snow | Frost | Fog | Brit. | Contr. | Elastic | Pixel | JPEG | |
| *Original* | 12.8 | 17.0 | 12.5 | 30.5 | 31.7 | 25.2 | 28.6 | 26.5 | 19.0 | 18.6 | 28.3 | 11.5 | 29.5 | 33.6 | 28.8 | 23.6 |
| *5×-expanded* by RandAugment | 16.7 | 21.9 | 27.5 | 42.2 | **42.5** | 35.8 | 40.2 | 36.9 | 31.9 | 30.0 | 43.1 | 20.4 | 41.2 | 44.7 | 37.6 | 34.2 (+10.6) |
| *5×-expanded* by GIF-SD | 29.7 | 36.4 | 32.7 | 51.9 | 32.4 | 39.2 | 46.0 | 45.3 | 38.1 | 47.1 | 55.7 | 37.3 | 48.6 | 53.2 | 49.4 | 43.3 (+19.7) |
| *20×-expanded* by GIF-SD | **31.8** | **39.2** | **34.7** | **58.4** | 33.4 | **43.1** | **51.9** | **51.7** | **47.4** | **55.0** | **63.3** | **46.5** | **54.9** | **58.0** | **53.6** | **48.2** (+24.6) |

Table 3: Accuracy of various architectures trained on $5\times$ expanded Cars. The results on other datasets are given in Appendix F.3.

| Dataset | ResNeXt-50 | WideResNet-50 | MobilteNet-v2 | Avg. |
|---|---|---|---|---|
| *Original* | $18.4_{\pm 0.5}$ | $32.0_{\pm 0.8}$ | $26.2_{\pm 4.2}$ | 25.5 |
| *Expanded* | | | | |
| RandAugment | $29.6_{\pm 0.8}$ | $49.2_{\pm 0.2}$ | $39.7_{\pm 2.5}$ | 39.5 (+14.0) |
| GIF-DALLE | $43.7_{\pm 0.2}$ | $60.0_{\pm 0.6}$ | $47.8_{\pm 0.6}$ | 50.5 (+25.0) |
| GIF-SD | $\mathbf{64.1}_{\pm 1.3}$ | $\mathbf{75.1}_{\pm 0.4}$ | $\mathbf{60.2}_{\pm 1.6}$ | **63.5** (+38.0) |

Table 4: Comparison between our methods and directly fine-tuning CLIP models on three medical image datasets.

| Dataset | PathMNIST | BreastMNIST | OrganSMNIST |
|---|---|---|---|
| *Original* dataset | $72.4_{\pm 0.7}$ | $55.8_{\pm 1.3}$ | $76.3_{\pm 0.4}$ |
| Linear-probing of CLIP | $74.3_{\pm 0.1}$ | $60.0_{\pm 2.9}$ | $64.9_{\pm 0.2}$ |
| fine-tuning of CLIP | $78.4_{\pm 0.9}$ | $67.2_{\pm 2.4}$ | $78.9_{\pm 0.1}$ |
| distillation of CLIP | $77.3_{\pm 1.7}$ | $60.2_{\pm 1.3}$ | $77.4_{\pm 0.8}$ |
| $5\times$-expanded by GIF-SD | $\mathbf{86.9}_{\pm 0.3}$ | $\mathbf{77.4}_{\pm 1.8}$ | $\mathbf{80.7}_{\pm 0.2}$ |

Table 5: Benefits of dataset expansion to CLIP fine-tuning on CIFAR100-S. Moreover, ID indicates in-distribution performance, while OOD indicates out-of-distribution performance on CIFAR100-C.

| Methods | ID Accuracy | OOD Accuracy |
|---|---|---|
| Training from scratch on original dataset | 35.0 | 23.6 |
| Fine-tuning CLIP on original dataset | 75.2 (+40.2) | 55.4 (+31.8) |
| Fine-tuning CLIP on 5x-expanded dataset by GIF-SD | **79.4** (+44.4) | **61.4** (+37.8) |

our method is at least $4\times$ more efficient than them. The limitations of these augmentations lie in their inability to generate new and highly diverse content. In contrast, GIF leverages strong prior models (*e.g.,* SD), guided by our discovered criteria, to perform data imagination. Hence, our method can generate more diversified and informative samples, yielding more significant gains per expansion.

**Benefits to model generalization.** Theorem 4.1 has shown the theoretical benefit of GIF to model generalization. Here, Table 2 demonstrates that GIF significantly boosts model out-of-distribution (OOD) generalization on CIFAR100-C [27], bringing 19.3% accuracy gain on average over 15 types of OOD corruption. This further verifies the empirical benefit of GIF to model generalization.

**Versatility to various architectures.** We also apply the $5\times$-expanded Cars dataset by GIF to train ResNeXt-50, WideResNet-50 and MobileNet V2 from scratch. Table 3 shows that the expanded dataset brings consistent accuracy gains for all architectures. This underscores the versatility of our method: once expanded, these datasets can readily be applied to train various model architectures.

**Comparisons with CLIP.** As our method applies CLIP for dataset expansion, one might question why not directly use CLIP for classifying the target dataset. In fact, our GIF offers two main advantages over CLIP in real-world small-data applications. First, GIF has superior applicability to the scenarios of different image domains. Although CLIP performs well on natural images, its transferability to non-natural domains, such as medical images, is limited (cf. Table 4). In contrast, our GIF is able to create samples of similar nature as the target data for dataset expansion, making it more applicable to real scenarios across diverse image domains. Second, GIF supplies expanded datasets suitable for training various model architectures. In certain scenarios like mobile terminals, hardware constraints may limit the supportable model size, which makes the public CLIP checkpoints (such as ResNet-50, ViT-B/32, or even larger models) unfeasible to use. Also, distilling from these CLIP models can only yield limited performance gains (cf. Table 1). In contrast, the expanded datasets by our method can be directly used to train various architectures (cf. Table 3), making our approach more practical for hardware-limited scenarios. Further comparisons and discussions are provided in Appendix F.4.

**Benefits to model fine-tuning**. In previous experiments, we have demonstrated the advantage of dataset expansion over model fine-tuning on medical image domains. Here, we further evaluate the benefits of dataset expansion to model fine-tuning. Hence, we use the 5x-expanded dataset by GIF-SD to fine-tune the pre-trained CLIP VIT-B/32. Table 5 shows that our dataset expansion significantly improves the fine-tuning performance of CLIP on CIFAR100-S, in terms of both in-distribution and out-of-distribution performance.

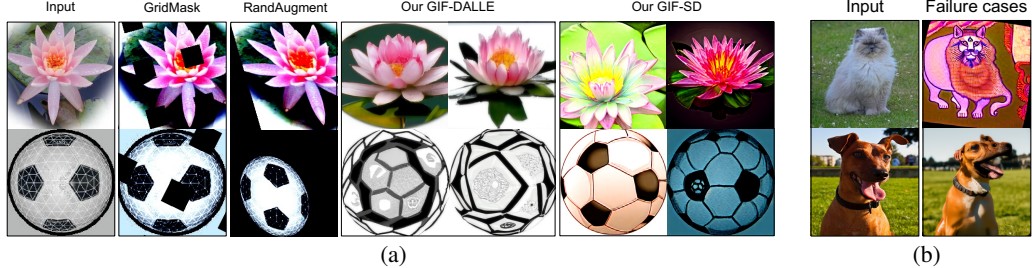

(a)                                                           (b)

Figure 5: Visualization. (a) Examples of the created samples for Caltech101 by augmentation and GIF. Please see Appendix G for the visualization of more datasets. (b) Failure cases by GIF-SD.

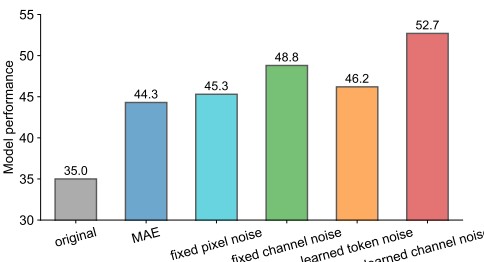

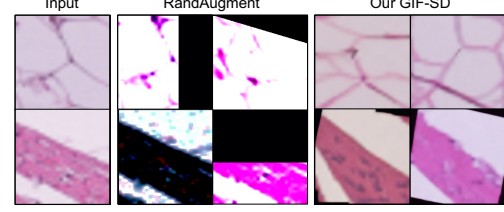

Figure 6: Effects of different perturbation noise on MAE for expanding CIFAR100-Subset by 5×.

Figure 7: Visualization of the created medical images by GIF-SD, where SD is fine-tuned on medical data before dataset expansion (Please see the analysis of fine-tuning in Appendix B.6).

## 5.2 Analyses and discussions

We next empirically analyze GIF. Due to the page limit, we provide more analyses of GIF (*e.g.,* mixup, CLIP, image retrieval, and long-tailed data) in Appendix B and Appendix F.

**Effectiveness of zero-shot CLIP in GIF.** We start with analyzing the role of zero-shot CLIP in GIF. We empirically find (cf. Appendix B.5) that GIF with fine-tuned CLIP performs only comparably to that with zero-shot CLIP on medical datasets. This reflects that zero-shot CLIP is enough to provide sound guidance without the need for fine-tuning. Additionally, a random-initialized ResNet50 performs far inferior to zero-shot CLIP in dataset expansion, further highlighting the significant role of zero-shot CLIP in GIF. Please refer to Appendix B.5 for more detailed analyses.

**Effectiveness of guidance in GIF.** Table 1 shows that our guided expansion obtains consistent performance gains compared to unguided expansion with SD, DALL-E2 or MAE, respectively. For instance, GIF-SD obtains 13.5% higher model accuracy on natural image datasets than unguided expansion with SD. This verifies the effectiveness of our criteria in optimizing the informativeness and diversity of the created samples. More ablation analyses of each criterion are given in Appendix F.5.

**Pixel-wise vs. channel-wise noise.** GIF-SD and GIF-MAE inject perturbation along the channel dimension instead of the spatial dimension. This is attributed to our empirical analysis in Appendix B.2. We empirically find that the generated image based on pixel-level noise variation is analogous to adding pixel-level noise to the original images. This may harm the integrity and smoothness of image content, leading the generated images to be noisy (cf. Figure 10(d) of Appendix B.2). Therefore, we decouple latent features into two dimensions (*i.e.,* token and channel) and particularly conduct channel-level perturbation. As shown in Figure 10(e), optimizing channel-level noise variation can generate more informative data, leading to more effective expansion (cf. Figure 6).

**Visualization.** The samples we created are visualized in Figures 5a and 7. While GridMask obscures some image pixels and RandAugment randomly alters images with pre-set transformations, both fail to generate new image content (cf. Figure 5a). More critically, as shown in Figure 7, RandAugment may crop the lesion location of medical images, leading to less informative and even noisy samples. In contrast, our method can not only generate *samples with novel content* (*e.g.,* varied postures and backgrounds of water lilies) but also *maintains their class semantics*, and thus is a more effective way to expand small-scale datasets than traditional augmentations, as evidenced by Table 1.

Table 6: Consumption costs. Time and costs are calculated from the expansion of 10,000 images. Accuracy improvements are compared to the original, small dataset.

| Method | Expansion speed | Time | Costs | Accuracy gains |
|---|---|---|---|---|
| Human collection | 121.0s per image | 2 weeks | $800 | - |
| Cutout | 0.008s per image | 76 seconds | $0.46 | +12.8 |
| GridMask | 0.007s per image | 72 seconds | $0.43 | +14.4 |
| RandAugment | 0.008s per image | 82 seconds | $0.49 | +20.5 |
| GIF-MAE (ours) | 0.008s per image | 80 seconds | $0.48 | +23.5 |
| GIF-SD (ours) | 6.6s per image | 2 hours | $40 | +36.9 |

Table 7: Relation analysis between the domain gap (FID) and model accuracy.

| Datasets | FID | Accuracy (%) |
|---|---|---|
| CIFAR100-S | - | 35.0 |
| RandAugment | 24.3 | 46.7 |
| Cutout | 104.7 | 44.3 |
| GridMask | 104.8 | 48.2 |
| GIF-MAE | 72.3 | 52.7 |
| GIF-DALLE | 39.5 | 54.5 |
| GIF-SD | 81.7 | 61.1 |

**Failure cases**. Figure 5b visualizes some failure cases by GIF-SD. As we use pre-trained models without fine-tuning on the natural images, the quality of some created samples is limited due to domain shifts. For example, the face of the generated cat in Figure 5b seems like a lion face with a long beard. However, despite seeming less realistic, those samples are created following our guidance, so they can still maintain class consistency and bring new information, thus benefiting model training.

**Computational efficiency and time costs**. Compared to human data collection, our GIF offers substantial savings in time and costs for small dataset expansion. As shown in Table 6, GIF-MAE achieves a $5\times$ expansion per sample in just 0.04 seconds, while GIF-SD accomplishes the same in 33 seconds. To illustrate, according to rates from Masterpiece Group[2], manually annotating 10,000 images takes two weeks and costs around $800. In contrast, GIF-SD generates the same amount of labeled data in a mere two hours, costing roughly $40 for renting 8 V100 GPUs[3]. Moreover, with a slight model performance drop, GIF-MAE can create 10,000 labeled data in just 6 seconds, at a cost of about $0.48$ for renting 8 V100 GPUs for 80 seconds. Specifically, GIF-MAE has a time cost within the same magnitude as data augmentation, but it delivers much better performance gains. The slight time overhead introduced by MAE is offset by GPU acceleration, resulting in competitive time costs. For those prioritizing performance, GIF-SD becomes a more attractive option. Although it involves a longer time due to its iterative diffusion process, it provides more significant performance gains. Note that our method only requires one-time expansion: the resultant dataset can be directly used to train various models (cf. Table 3), without the need for regeneration for each model.

**Relation analysis between the domain gap and model performance**. We further compute the Fréchet Inception Distance (FID) between the synthetic data generated by different methods and the original data of CIFAR100-S. Interestingly, while one might initially assume that a lower FID implies better quality for the expanded data, the actual performance does not consistently follow this notion. As shown in Table 7, even though GIF-SD has a worse FID than GIF-DALLE, it achieves better performance. Likewise, despite having nearly identical FIDs, Cutout and GridMask lead to different performance. These results suggest that the effectiveness of dataset expansion methods depends on how much additional information and class consistency the generated data can provide to the original dataset, rather than the distribution similarity between those samples and the original data. This discussion may spark further research into the relationship between expansion effectiveness and data fidelity (as measured by metrics like FID), potentially guiding the development of even more effective dataset expansion techniques in the future.

## 6 Conclusion

This paper has explored a novel task, dataset expansion, towards resolving the data scarcity issue in DNN training. Inspired by human learning with imagination, we presented a novel guided imagination framework for dataset expansion. Promising results on small-scale natural and medical image datasets have verified its effectiveness. Despite its encouraging results, there is still room to improve. That is, using only the generated samples for model training is still worse than using real samples of equivalent size, suggesting huge potential for algorithmic data generation to improve. We expect that our work can inspire further exploration of dataset expansion so that it can even outperform a human-collected dataset of the same size. Please refer to Appendix D.5 for a more detailed discussion on limitations and broader impact of our work.

---

[2] https://mpg-myanmar.com/annotation
[3] https://cloud.google.com/compute/gpus-pricing#gpu-pricing

**Acknowledgments**

This work was partially supported by the National Research Foundation Singapore under its AI Singapore Programme (Award Number: AISG2-TC-2021-002 and AISG2-PhD-2021-08-008).

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

# A   More Related Studies

**Image synthesis.**   Over the past decade, image synthesis [2, 13, 65, 73, 111] has been extensively explored, with four main approaches leading the way: generative adversarial networks (GANs) [17, 33], auto-regressive models [38, 57], diffusion models [16, 31], and neural radiance fields [20, 47, 93]. Recently, diffusion techniques, such as DALL-E2 [56], Imagen [61], and Stable Diffusion [59], have demonstrated exceptional capabilities in producing photo-realistic images. In practice, these techniques can serve as prior models in our GIF framework for dataset expansion.

Additionally, CLIP [54], thanks to its text-image matching ability, has been used to guide image generation [37, 48, 51, 76]. In these approaches, CLIP matches a generated image with a given text. In contrast, our work uses CLIP to align the latent features of category-agnostic generative models with the label space of the target dataset. This alignment enables GIF to perform guided data expansion, generating informative new samples specific to target classes.

Furthermore, model inversion [83, 86] is another technique that has been investigated for image generation by inverting a trained classification network [77, 92] or a GAN model [112]. Although we currently apply only two advanced generative models (DALL-E2 and Stable Diffusion) and a reconstruction model (MAE) within the GIF framework in this study, model inversion methods could also be incorporated into our framework for dataset expansion. This opens up exciting avenues for future research.

**More discussion on data augmentation.**   Image data augmentation has become a staple in enhancing the generalization of DNNs during model training [69, 91]. Based on technical characteristics, image data augmentation can be categorized into four main types: image manipulation, image erasing, image mix, and auto augmentation.

Image manipulation augments data through image transformations like random flipping, rotation, scaling, cropping, sharpening, and translation [91]. Image erasing, on the other hand, substitutes pixel values in certain image regions with constant or random values, as seen in Cutout [15], Random Erasing [109], GridMask [8], and Fenchmask [43]. Image mix combines two or more images or sub-regions into a single image, as exemplified by Mixup [100], CutMix [96], and AugMix [28]. Lastly, Auto Augmentation utilizes a search algorithm or random selection to determine augmentation operations from a set of random augmentations, such as AutoAugment [11], Fast AutoAugment [45], and RandAugment [12].

While these methods have shown effectiveness in certain applications, they primarily augment data by applying pre-defined transformations on each image. This results in only local variations in pixel values and does not generate images with significantly diversified content. Furthermore, as most methods employ random operations, they cannot ensure that the augmented samples are informative for model training and may even introduce noisy augmented samples. Consequently, the new information brought about is often insufficient for expanding small datasets, leading to low expansion efficiency (cf. Figure 4). In contrast, our proposed GIF framework utilizes powerful generative models (such as DALL-E2 and Stable Diffusion) trained on large-scale image datasets, guiding them to optimize latent features in accordance with our established criteria (*i.e., class-maintained information boosting* and *sample diversity promotion*). This results in the creation of images that are both more informative and diversified than those from simple image augmentation, thereby leading to more efficient and effective dataset expansion.

We note that the work [87] also explores MAE for image augmentation based on its reconstruction capability. It first masks some sub-regions of images and then feeds the masked images into MAE for reconstruction. The recovered images with slightly different sub-regions are then used as augmented samples. Like other random augmentation methods, this approach only varies pixel values locally and cannot ensure that the reconstructed images are informative and useful. In contrast, our GIF-MAE guides MAE to create informative new samples with diverse styles through our guided latent feature optimization strategy. Therefore, GIF-MAE is capable of generating more useful synthetic samples, effectively expanding the dataset.

**Contrasting with dataset distillation.** Dataset distillation, also known as dataset condensation, is a task that seeks to condense a large dataset into a smaller set of synthetic samples that are comparably effective [6, 68, 79, 89, 106, 107, 108, 110]. The goal of this task is to train models to achieve performance comparable to the original dataset while using significantly fewer resources. Such a task is diametrically opposed to our work on dataset expansion, which strives to *expand a smaller dataset into a larger, richer, and more informative one*. We achieve this by intelligently generating new samples that are both informative and diverse. Hence, dataset distillation focuses on large-data applications, whereas our focus lies on expanding dataset diversity and information richness for more effective deep model training in small-data applications.

**Contrasting with transfer learning.** Numerous studies have focused on model transfer learning techniques using publicly available large datasets like ImageNet [14, 58]. These approaches include model fine-tuning [23, 44, 101], knowledge distillation [21, 30], and domain adaptation [19, 46, 53, 74, 95, 104].

Despite effectiveness in certain applications, these model transfer learning paradigms also suffer from key limitations. For instance, the study [55] found that pre-training and fine-tuning schemes do not significantly enhance model performance when the pre-trained datasets differ substantially from the target datasets, such as when transferring from natural images to medical images. Moreover, model domain adaptation often necessitates that the source dataset and the target dataset share the same or highly similar label spaces, a requirement that is often unmet in small-data application scenarios due to the inaccessibility of a large-scale and labeled source domain with a matching label space. In addition, the work [71] found that knowledge distillation does not necessarily work if the issue of model mismatch exists [9], *i.e.,* large discrepancy between the predictive distributions of the teacher model and the student model. The above limitations of model transfer learning underscore the importance of the dataset expansion paradigm: if a small dataset is successfully expanded, it can be directly used to train various model architectures.

We note that some data-free knowledge distillation studies [7, 92, 98] also synthesize images, but their goal is particularly to enable *knowledge distillation* in the setting without data. In contrast, our task is independent of model knowledge distillation. The expanded datasets are not method-dependent or model-dependent, and, thus, can train various model architectures to perform better than the original small ones.

# B  More Preliminary Studies

## B.1   Sample-wise expansion or sample-agnostic expansion?

When we design the selective expansion strategy in Section 3.2, another question appears: should we ensure that each sample is expanded by the same ratio? To determine this, we empirically compare RandAugment expansion with sample-wise selection and sample-agnostic selection according to one expansion criteria, *i.e., class-maintained information boosting*. Figure 8 shows that sample-wise expansion performs much better than sample-agnostic expansion. To find out the reason for this phenomenon, we visualize how many times a sample is expanded by sample-agnostic expansion. As shown in Figure 9, the expansion numbers of different samples by sample-agnostic expansion present a long-tailed distribution [103], with many image samples not expanded at all. The main reason for this is that, due to the randomness of RandAugment and the differences among images, not all created samples are informative and it is easier for some samples to be augmented more frequently than others. Therefore, given a fixed expansion ratio, the sample-agnostic expansion strategy, as it ignores the differences in images, tends to select more expanded samples for more easily augmented images. This property leads sample-agnostic expansion to waste valuable original samples for expansion (*i.e.,* loss of information) and also incurs a class-imbalance problem, thus resulting in worse performance in Figure 8. In contrast, sample-wise expansion can fully take advantage of all the samples in the target dataset and thus is more effective than sample-agnostic expansion, which should be considered when designing dataset expansion approaches.

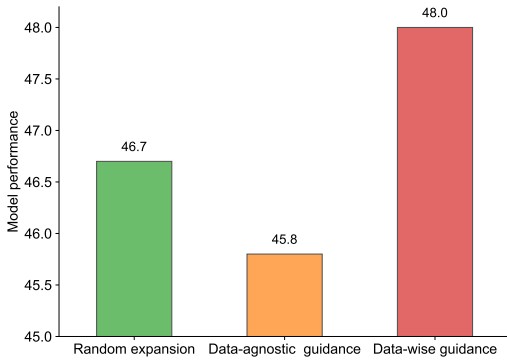

Figure 8: Comparison of model performance between samples-wise selection and sample-agnostic selection for RandAugment expansion on CIFAR100-Subset.

Figure 9: Statistics of the expansion numbers of different data in CIFAR100-Subset by sample-agnostic selective expansion with RandAugment, which presents a long-tailed distribution.

## B.2   Pixel-level noise or channel-level noise?

In our preliminary studies exploring the MAE expansion strategy, we initially used pixel-level noise to modify latent features. However, this approach did not perform well. To understand why, we analyze the reconstructed images. An example of this is presented in Figure 10(d). We find that the generated image based on pixel-level noise variation is analogous to adding pixel-level noise to the original images. This may harm the integrity and smoothness of image content, leading the reconstructed images to be noisy and less informative. In comparison, as shown in Figure 10(b), a more robust augmentation method like RandAugment primarily alters the style and geometric positioning of images but only slightly modifies the content semantics. As a result, it better preserves content consistency. This difference inspires us to factorize the influences on images into two dimensions: image styles and image content. In light of the findings in [32], we know that the channel-level latent features encode more subtle style information, whereas the token-level latent features convey more content information. We thus decouple the latent features of MAE into two dimensions (*i.e.,* a token dimension and a channel dimension), and plot the latent feature distribution change between the generated image and the original image in these two dimensions.

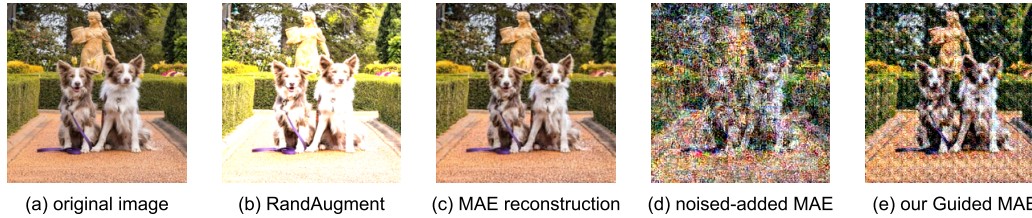

| (a) original image | (b) RandAugment | (c) MAE reconstruction | (d) noised-added MAE | (e) our Guided MAE |

Figure 10: An illustrated visualization of the generated images by (b) RandAugment, (c) MAE reconstruction, (d) random pixel-level variation over latent features, and (e) our guided MAE expansion. We find our guided MAE can generate content-consistent images of diverse styles.

Figure 11 shows the visualization of this latent feature distribution change. The added pixel-level noise changes the token-level latent feature distribution more significantly than RandAugment (cf. Figure 11(a)). However, it only slightly changes the channel-level latent feature distribution (cf. Figure 11(b)). This implies that pixel-level noise mainly alters the content of images but slightly changes their styles, whereas RandAugment mainly influences the style of images while maintaining their content semantics. In light of this observation and the effectiveness of RandAugment, we are motivated to disentangle latent features into the two dimensions, and particularly conduct channel-level noise to optimize the latent features in our method. As shown in Figure 11, the newly explored channel-level noise variation varies the channel-level latent feature more significantly than the token-level latent feature. It thus can diversify the style of images while maintaining the integrity of image content. This innovation enables the explored MAE expansion strategy to generate more informative samples compared to pixel-level noise variation (cf. Figure 10(d) vs. Figure 10(e)), leading to more effective dataset expansion, as shown in Figure 6. In light of this finding, we also conduct channel-level noise variation for GIF-SD.

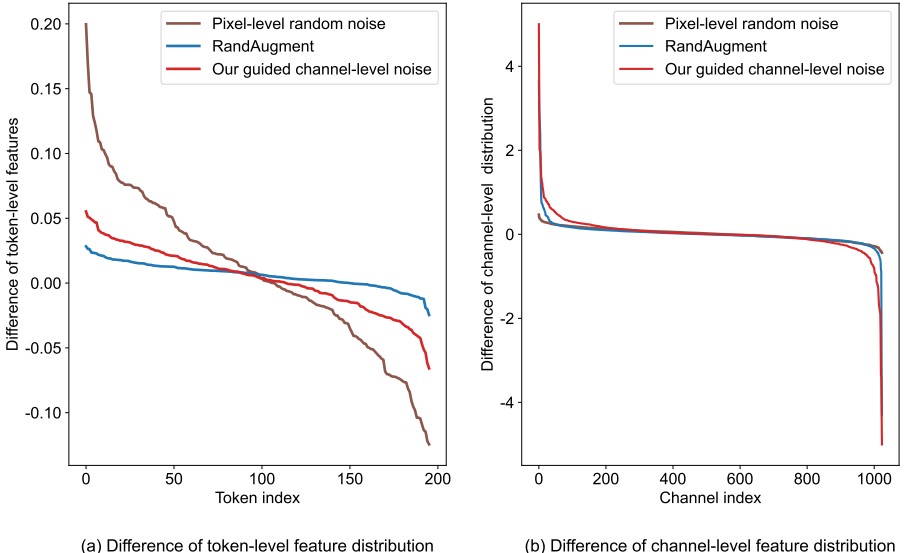

(a) Difference of token-level feature distribution  (b) Difference of channel-level feature distribution

Figure 11: Changes of the latent feature distributions along the token dimension and the channel dimension, between the latent feature of the generated image and that of the original image.

### B.3 How to design prompts for Stable Diffusion?

Text prompts play an important role in image generation of Stable Diffusion. The key goal of prompts in dataset expansion is to further diversify the generated image without changing its class semantics. We find that domain labels, class labels, and adjective words are necessary to make the prompts semantically effective. The class label is straightforward since we need to ensure the created samples have the correct class labels. Here, we show the influence of different domain labels and adjective words on image generation of Stable Diffusion.

**Domain labels**. We first visualize the influence of different domain prompts on image generation. As shown in Figure 12, domain labels help to generate images with different styles. We note that similar domain prompts, like "a sketch of" and "a pencil sketch of", tend to generate images with similar styles. Therefore, it is sufficient to choose just one domain label from a set of similar domain prompts, which does not influence the effectiveness of dataset expansion but helps to reduce the redundancy of domain prompts. In light of this preliminary study, we design the domain label set by ["an image of", "a real-world photo of", "a cartoon image of", "an oil painting of", "a sketch of"].

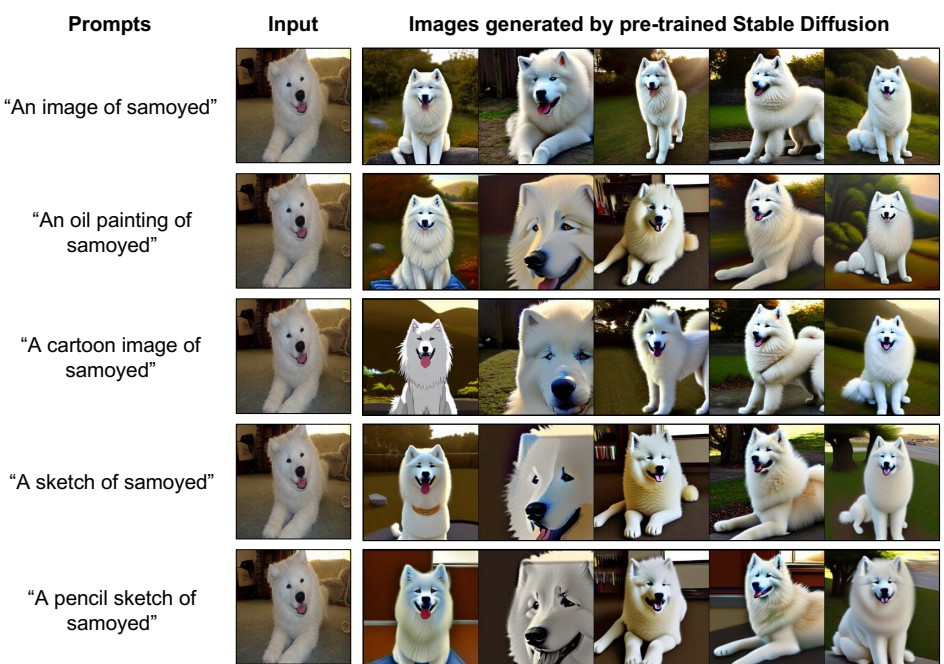

Figure 12: The influence of the domain prompts on image generation of pre-trained Stable Diffusion. The input image is selected from the Pets dataset. Here, the strength hyper-parameter is set to 0.9, and the scale is set to 20.

**Adjective words**. We next show the influence of different adjective words on image generation of Stable Diffusion. As shown in Figure 13, different adjectives help diversify the content of the generated images further, although some adjectives may lead to similar effects on image generation. Based on the visualization exploration, we design the adjective set by [" ", "colorful", "stylized", "high-contrast", "low-contrast", "posterized", "solarized", "sheared", "bright", "dark"].

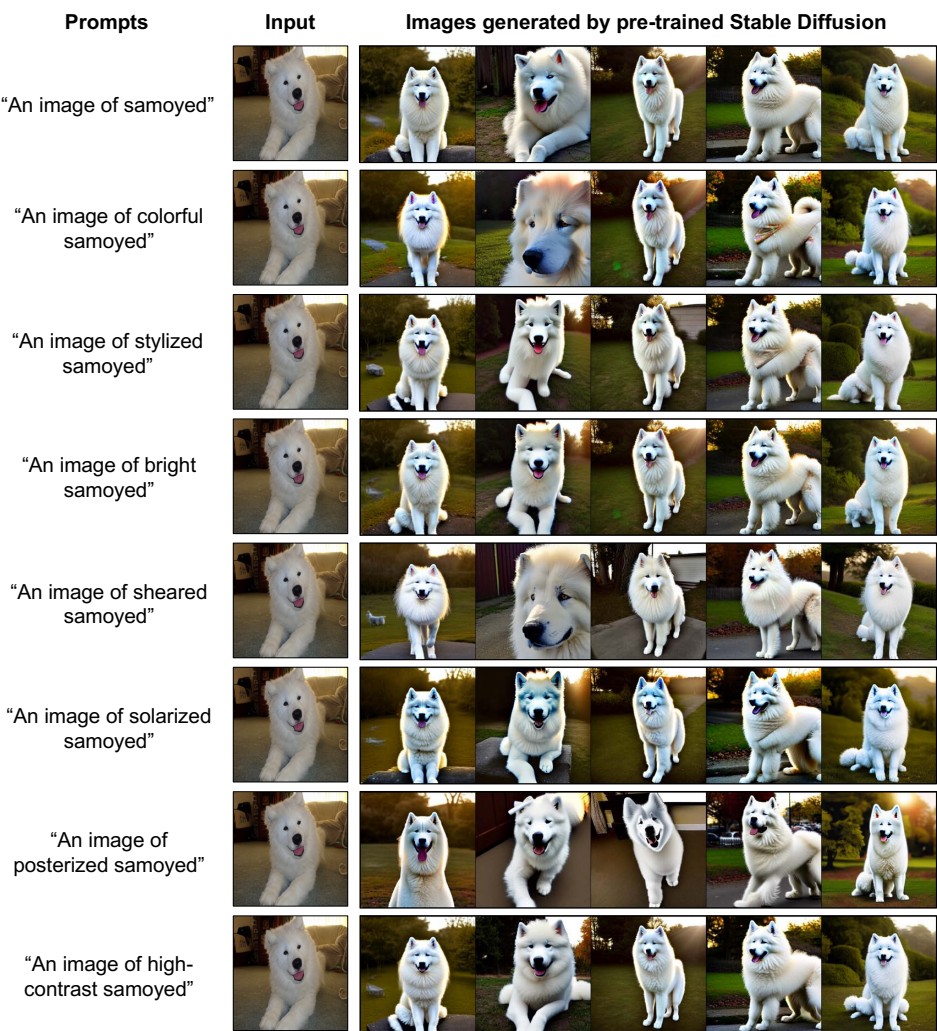

Figure 13: The influence of the adjective prompts on image generation of pre-trained Stable Diffusion. The input image is selected from the Pets dataset. Here, the strength hyper-parameter is set to 0.9, and the scale is set to 20.

### B.4 How to set hyper-parameters for Stable Diffusion?

### B.4.1 Hyper-parameter of strength

The hyper-parameter of the nosing strength controls to what degree the initial image is destructed. Setting strength to 1 corresponds to the full destruction of information in the input image while setting strength to 0 corresponds to no destruction of the input image. The higher the strength value is, the more different the generated images would be from the input image. In dataset expansion, the choice of strength depends on the target dataset, but we empirically find that selecting the strength value from [0.5, 0.9] performs better than other values. A too-small value of strength (like 0.1 or 0.3) brings too little new information into the generated images compared to the seed image. At the same time, a too-large value (like 0.99) may degrade the class consistency between the generated images and the seed image when the hyper-parameter of scale is large.

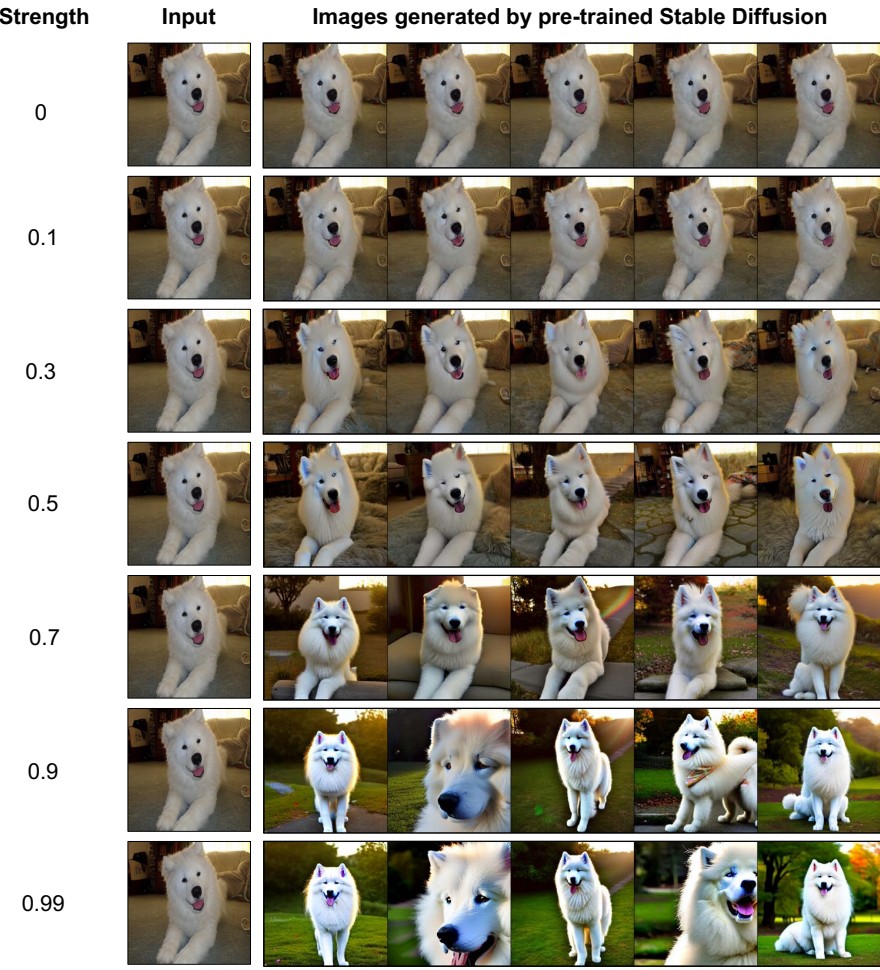

Figure 14: The influence of the "strength" hyper-parameter on image generation of pre-trained Stable Diffusion. The input image is selected from the Pets dataset. The prompt is "an image of colorful samoyed", while the scale is set to 20.

### B.4.2 Hyper-parameter of scale

The hyper-parameter of scale controls the importance of the text prompt guidance on image generation of Stable Diffusion. The higher the scale value, the more influence the text prompt has on the generated images. In dataset expansion, the choice of strength depends on the target dataset, but we empirically find that selecting the strength value from [5, 50] performs better than other values. A too-small value of scale (like 1) brings too little new information into the generated images, while a too-large value (like 100) may degrade the class information of the generated images.

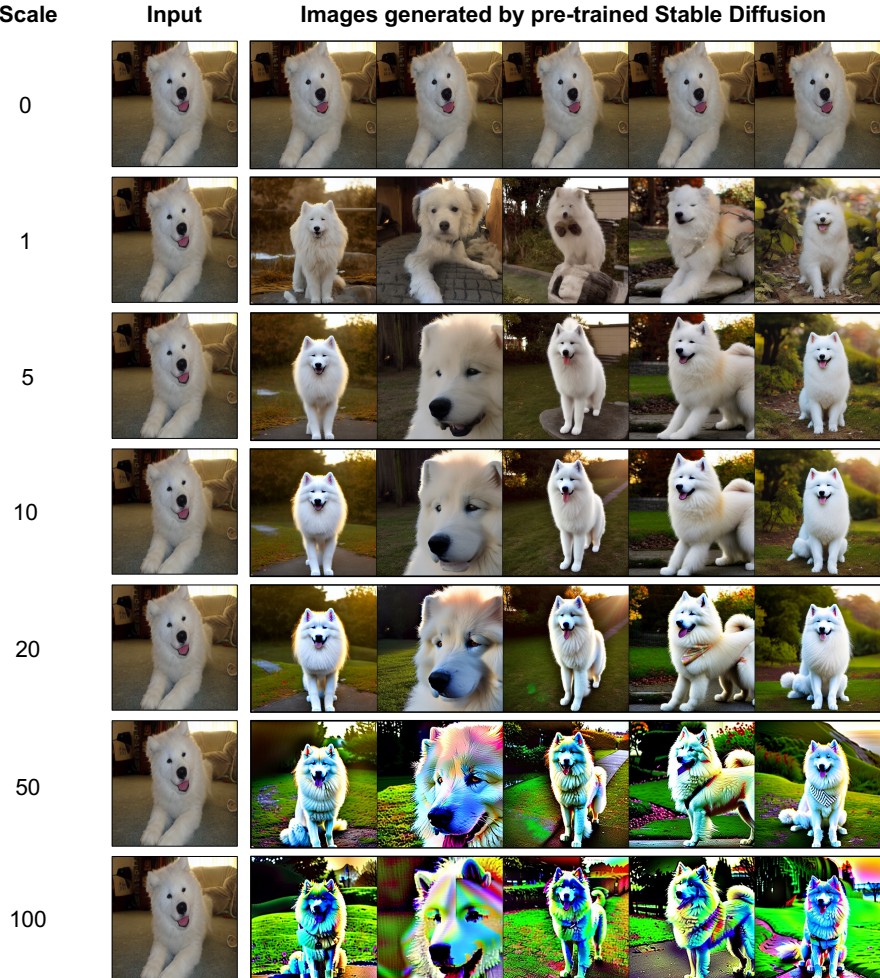

Figure 15: The influence of the "scale" hyper-parameter on image generation of pre-trained Stable Diffusion. The input image is selected from the Pets dataset. The prompt is "an image of colorful samoyed", while the strength is set to 0.9.

## B.5 More discussions on the effectiveness of zero-shot CLIP

In GIF, we exploit the zero-shot discriminability of the pre-trained CLIP to guide dataset expansion. In Table 1, we have found that the zero-shot performance of CLIP is not significantly good, particularly on medical image datasets. It is interesting to know whether further fine-tuning CLIP on the target medical dataset can bring further improvement. To determine this, we further compare the results of GIF-MAE with fine-tuned CLIP and with zero-shot CLIP based on OrganSMNIST. To be specific, we add a linear classifier on the top of the CLIP image encoder and fine-tune the CLIP model.

As shown in Table 8, GIF-MAE with fine-tuned CLIP performs only comparably to that with zero-shot CLIP, which reflects that the CLIP's zero-shot classifier is enough to provide sound guidance. The reason is that, although the zero-shot performance is not that good, **CLIP still plays an important anchor effect in maintaining the class semantics of the generated samples and helps to bring new information**. Let us first recall the class-maintained informativeness score: $\mathcal{S}_{inf} = s'_j + (s \log(s) - s' \log(s'))$. Specifically, no matter whether CLIP zero-shot classifier is accurate or not, maximizing $s'_j$ essentially uses the prediction of **the seed data as an anchor in the CLIP semantic space to regularize the class semantics of the perturbed features**. This ensures the created data maintain the correct class, which is highly important for effective dataset expansion. In addition, maximizing the entropy difference, *i.e.,* $s \log(s) - s' \log(s')$, encourages the perturbed feature to have higher entropy regarding CLIP zero-shot prediction. When CLIP zero-shot classifier is accurate, the entropy increment enables the created data to become more difficult to classify regarding CLIP zero-shot discrimination and thus brings more information for classification model training. When CLIP zero-shot classifier is not that accurate, the entropy increment introduces variations into the created data and makes them different from the seed data. **Under the condition that the true class is maintained, this optimization is beneficial to boosting the diversity of the expanded dataset, which is helpful for model training**. Hence, CLIP's zero-shot abilities are useful for guided imagination in various image domains.

Afterwards, given that zero-shot CLIP can provide valuable guidance despite its limited accuracy, one may wonder whether a random-initialized deep model could serve a similar function. However, as shown in Table 8, using a random-initialized ResNet50 as the guidance model for dataset expansion performs much worse than zero-shot CLIP (*i.e.,* 79.0 vs. 80.6). This could be attributed to the fact that, **although the classifiers of both random ResNet50 and zero-shot CLIP struggle with the target medical classes, the CLIP's pre-training results in a feature space that is more semantically meaningful and representative than a randomly-initialized ResNet50**. This distinction allows zero-shot CLIP to better anchor the class semantics of synthetic samples, thereby leading to more effective dataset expansion. These empirical observations further verify the effectiveness of using zero-shot CLIP in guiding dataset expansion.

Table 8: Comparison between the model performance by GIF-MAE expansion with zero-shot CLIP guidance and fine-tuned CLIP guidance, as well as random-initialized ResNet-50 guidance, based on the OrganSMNIST medical image dataset. All results are averaged over three runs.

| OrganSMNIST | Guidance model | Guidance model accuracy | Model accuracy |
|---|---|---|---|
| Original dataset | - | - | 76.3 |
| | Random-initialized ResNet50 | $7.1_{\pm 0.8}$ | 79.0 (+2.7) |
| 5×-expanded by GIF-MAE | Fine-tuned CLIP | $75.6_{\pm 1.2}$ | 80.7 (+4.4) |
| | Zero-shot CLIP (ours) | $7.7_{\pm 0.0}$ | 80.6 (+4.3) |

### B.6 Do we need to fine-tune generative models on medical image datasets?

Stable Diffusion (SD) and DALL-E2 are trained on large-scale datasets consisting of natural image and text pairs, showing powerful capabilities in natural image generation and variation. However, when we directly apply them to expand medical image datasets, we find the performance improvement is limited, compared to MAE as shown in Table 9.

Table 9: Accuracy of ResNet-50 trained on the $5\times$-expanded medical image datasets by GIF based on SD and DALLE w/o and w/ fine-tuning. All results are averaged over three runs.

| Dataset | PathMNIST | BreastMNIST | OrganSMNIST | Average |
|---|---|---|---|---|
| *Original* | $72.4_{\pm 0.7}$ | $55.8_{\pm 1.3}$ | $76.3_{\pm 0.4}$ | 68.2 |
| GIF-MAE | $82.0_{\pm 0.7}$ | $73.3_{\pm 1.3}$ | $80.6_{\pm 0.5}$ | 78.6 |
| GIF-DALLE (w/o tuning) | $78.4_{\pm 1.0}$ | $59.3_{\pm 2.5}$ | $76.4_{\pm 0.3}$ | 71.4 |
| GIF-DALLE (w/ tuning) | $84.4_{\pm 0.3}$ | $76.6_{\pm 1.4}$ | $80.5_{\pm 0.2}$ | 80.5 |
| GIF-SD (w/o tuning) | $80.8_{\pm 1.6}$ | $59.4_{\pm 2.2}$ | $79.5_{\pm 0.4}$ | 73.2 |
| GIF-SD (w/ tuning) | $86.9_{\pm 0.6}$ | $77.4_{\pm 1.8}$ | $80.7_{\pm 0.2}$ | 81.7 |

To pinpoint the reason, we visualize the generated images by SD on PathMNIST. As shown in Figure 16(top), we find that SD fails to generate photo-realistic medical images, particularly when the hyper-parameter of strength is high. For example, the generated colon pathological images by pre-trained SD look more like a natural sketch and lack medical nidus areas found in the input image. This implies that directly applying SD suffers from significant domain shifts between natural and medical images, preventing the generation of photo-realistic and informative medical samples using its image variation abilities. This issue also happens when applying DALL-E2 for medical dataset expansion. In contrast, MAE is a reconstruction model and does not need to generate new content for the target images, so it has much less negative impact by domain shifts. To address the issue, when applying SD and DALL-E2 to medical domains, we first fine-tune them on target medical datasets, followed by dataset expansion. Specifically, DALL-E2 is fine-tuned based on image reconstruction, while SD is fine-tuned based on Dreambooth [60]. As shown in Figure 16(bottom), the fine-tuned SD is able to generate medical images that are more domain-similar to the input colon pathological image. Thanks to the fine-tuned SD and DALL-E2, GIF is able to bring more significant performance gains over GIF-MAE (cf. Table 9), and thus expands medical image datasets better.

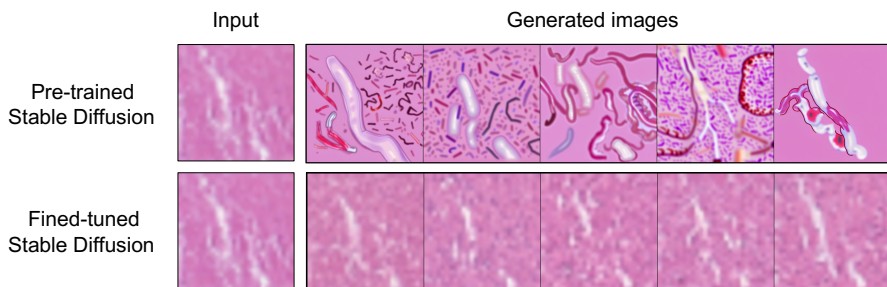

Figure 16: Visualization of the synthetic medical colon pathological images by Stable Diffusion (SD) with or without fine-tuning. Here, the prompt of SD is "a colon pathological sketch of colorful debris", while the strength is set to 0.5. We find that SD suffers from severe domain shifts between natural and medical images and cannot generate photo-realistic and informative medical samples. In contrast, the generated medical images by the fine-tuned SD are more domain-similar to the input colon pathological image.

### B.7 Visualization of created medical images

In the main paper, we visualize the created medical samples by GIF-SD. Here, we further visualize the created medical samples by GIF-MAE and discuss them. As shown in Figure 17, RandAugment randomly varies the medical images based on a set of pre-defined transformations. However, due to its randomness, RandAugment may crop the lesion location of medical images and cannot guarantee the created samples to be informative, even leading to noise samples. In contrast, our GIF-MAE can generate content-consistent images with diverse styles, so it can enrich the medical images while maintaining their lesion location unchanged. Therefore, GIF-MAE is able to expand medical image datasets better than RandAugment, leading to higher model performance improvement (cf. Table 1). However, GIF-MAE is unable to generate images with diverse content, which limits its effectiveness. In comparison, SD, after fine-tuning, is able to generate class-maintained samples with more diverse content and styles, and thus achieves better expansion effectiveness (cf. Table 1). To summarize, our methods can expand medical image datasets more effectively than data augmentation.

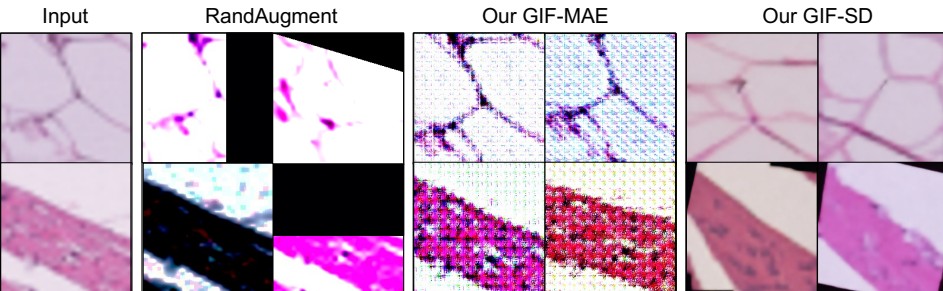

Figure 17: Examples of the created samples for PathMNIST by RandAugment and GIF.

## C Theoretical Analysis

In this appendix, we seek to analyze the benefits of our dataset expansion to model generalization performance. Inspired by [67], we resort to the concept of $\delta$-cover [62, 29] to analyze how data diversity influences the generalization error bound. Specifically, "a dataset $E$ is a $\delta$-cover of a dataset $S$" means a set of balls with radius $\delta$ centered at each sample of the dataset $E$ can cover the entire dataset $S$.

**Definition C.1.** ($\delta$-cover [62]) Let $(\mathcal{M}, \rho)$ be a metric space, let $S \subseteq \mathcal{M}$ and let $\mu > 0$. A set $E \subseteq \mathcal{M}$ is a $\delta$-cover for $S$, if for every $s \in S$, there is an $e \in E$ such that $\rho(s, e) \leq \delta$. The minimal $\delta$ regarding $S$ and $E$ is denoted by $\delta_{min}$.

In this work, we follow the assumptions of the work [67] and extend its Theorem 1 to the version of the generalization error bound. Let $A$ be a learning algorithm that outputs a set of parameters, given a training dataset $\mathcal{D} = \{x_i, y_i\}_{i \in [n]}$ with $n$ i.i.d. samples drawn from the data distribution $\mathcal{P}_{\mathcal{Z}}$. Assume that the hypothesis function is $\lambda^\eta$-Lipschitz continuous, the loss function $\ell(x, y)$ is $\lambda^\ell$-Lipschitz continuous for all $y$ and bounded by $L$, and $\ell(x_i, y_i; A) = 0$ for $\forall i \in [n]$. If the training set $\mathcal{D}$ is a $\delta$-cover of $\mathcal{P}_{\mathcal{Z}}$, with probability at least $1 - \gamma$, the generalization error bound satisfies:

$$|\mathbb{E}_{x,y \sim \mathcal{P}_{\mathcal{Z}}}[\ell(x, y; A)] - \frac{1}{n} \sum_{i \in [n]} \ell(x_i, y_i; A)| \overset{c}{\leq} \delta_{min}(\lambda^\ell + \lambda^\eta LC), \qquad (3)$$

where $C$ is a constant, and the symbol $\overset{c}{\leq}$ indicates "smaller than" up to an additive constant. According to the property of the $\delta$-cover, we then define the dataset diversity, called $\delta$-diversity, by the inverse of the minimal $\delta_{min}$:

**Definition C.2.** ($\delta$-diversity) If a dataset $E$ is a $\delta$-cover of the full dataset $S$, then the $\delta$-diversity of the set $E$ regarding the full set $S$ is $\delta_{div} = \frac{1}{\delta_{min}}$.

The $\delta$-diversity is easy to understand: given a training set $\mathcal{D} = \{x_i, y_i\}_{i \in [n]}$ that is a $\delta$-cover of the data distribution $\mathcal{P}_{\mathcal{Z}}$, if the radius $\delta_{min}$ is high, the diversity of this dataset must be low. Then, we have:

**Theorem C.1.** *Let $A$ denote a learning algorithm that outputs a set of parameters given a dataset $\mathcal{D} = \{x_i, y_i\}_{i \in [n]}$ with $n$ i.i.d. samples drawn from distribution $\mathcal{P}_{\mathcal{Z}}$. Assume the hypothesis function is $\lambda^\eta$-Lipschitz continuous, the loss function $\ell(x, y)$ is $\lambda^\ell$-Lipschitz continuous for all $y$, and is bounded by $L$, with $\ell(x_i, y_i; A) = 0$ for all $i \in [n]$. If $\mathcal{D}$ constitutes a $\delta$-cover of $\mathcal{P}_{\mathcal{Z}}$, then with probability at least $1 - \gamma$, the generalization error bound satisfies:*

$$|\mathbb{E}_{x,y \sim \mathcal{P}_{\mathcal{Z}}}[\ell(x, y; A)] - \frac{1}{n} \sum_{i \in [n]} \ell(x_i, y_i; A)| \overset{c}{\leq} \frac{\lambda^\ell + \lambda^\eta LC}{\delta_{div}}, \qquad (4)$$

*where $C$ is a constant, and the symbol $\overset{c}{\leq}$ indicates "smaller than" up to an additive constant.*

This theorem shows that the generalization error is bounded by the inverse of $\delta$-diversity. That is, the more diverse samples are created by a dataset expansion method, the more improvement of generalization performance would be made in model training. In real small-data applications, the data limitation issue leads the covering radius $\delta$ to be very large and thus the $\delta$-diversity is low, which severely affects the generalization performance of the trained model. More critically, simply increasing the data number (*e.g.,* via data repeating) does not help the generalization since it does not increase $\delta$-diversity. Instead of simply increasing the number of samples, our proposed GIF framework adopts two key imagination criteria (*i.e.,* "class-maintained informativeness boosting" and "sample diversity promotion") to guide advanced generative models (*e.g.,* DALL-E2 and Stable Diffusion) to synthesize informative and diversified new samples. Therefore, the expanded dataset would have higher data diversity than random augmentation, which helps to increase $\delta$-diversity and thus improves model generalization performance.

## D  More Method and Implementation Details

### D.1  Method details of GIF-DALLE

Thanks to strong image generation abilities, GIF-DALLE applies DALL-E2 [56] as its prior model which follows the pipeline described in Section 4. Its pseudo-code is provided in Algorithm 1, where the image embedding obtained by $f_{\text{CLIP-I}}$ serves as diffusion guidance to help the diffusion decoder to generate new images. GIF-DALLE conducts guided imagination on *the CLIP embedding space.*

We further clarify the implementation of the proposed guidance. Specifically, *class-maintained informativeness* $\mathcal{S}_{inf}$ encourages the consistency between the predicted classification scores $s$ and $s'$, and improves the information entropy for the predicted score of the generated sample $s'$:

$$\mathcal{S}_{inf} = s'_j + (s\log(s) - s'\log(s')), \quad \text{s.t.,} \quad j = \arg\max(s). \tag{5}$$

Here, $j = argmax(s)$ is the predicted class label of the original latent feature. Such a criterion helps to keep the class semantics of the optimized feature the same as that of the original one in the CLIP embedding space while encouraging the perturbed feature to have higher information entropy regarding CLIP zero-shot predictions. This enables the generated samples to be more informative for follow-up model training. To promote sample diversity, the *diversity* $\mathcal{S}_{div}$ is computed by the Kullback–Leibler (KL) divergence among all perturbed latent features of a seed sample as follows:

$$\mathcal{S}_{div} = \mathcal{D}_{KL}(f'\|\bar{f}) = \sigma(f')\log(\sigma(f')/\sigma(\bar{f})), \tag{6}$$

where $f'$ denotes the current perturbed latent feature and $\bar{f}$ indicates the mean over the $K$ perturbed latent features of this seed sample. In implementing diversity promotion $\mathcal{S}_{div}$, we measure the dissimilarity of two feature vectors by applying the softmax function $\sigma(\cdot)$ to the latent features, and then measuring the KL divergence between the resulting probability vectors.

---
**Algorithm 1:** GIF-DALLE Algorithm

**Input:** Original small dataset $\mathcal{D}_o$; CLIP image encoder $f_{\text{CLIP-I}}(\cdot)$; DALL-E2 diffusion decoder $G(\cdot)$; CLIP zero-shot classifier $w(\cdot)$; Expansion ratio $K$; Perturbation constraint $\varepsilon$.
**Initialize:** Synthetic data set $\mathcal{D}_s = \emptyset$;
**for** $x \in \mathcal{D}_o$ **do**
    $\mathcal{S}_{inf} = 0$;
    $f = f_{\text{CLIP-I}}(x)$ ;                      // latent feature encoding for seed sample
    $s = w(f)$ ;                          // CLIP zero-shot prediction for seed sample
    **for** *i=1,...,K* **do**
        Initialize noise $z_i \sim \mathcal{U}(0,1)$ and bias $b_i \sim \mathcal{N}(0,1)$;
        $f'_i = \mathcal{P}_{f,\epsilon}((1+z_i)f + b_i)$ ;              // noise perturbation (Eq.(1))
        $s' = w(f'_i)$ ;                      // CLIP zero-shot prediction
        $\mathcal{S}_{inf} \mathrel{+}= s'_j + (s\log(s) - s'\log(s'))$, s.t. $j = \arg\max(s)$ ;    // class-maintained
        informativeness (Eq.(5))
    **end**
    $\bar{f} = mean(\{f'_i\}_{i=1}^K)$;
    $\mathcal{S}_{div} = \sum_i \{\mathcal{D}_{KL}(\sigma(f'_i)\|\sigma(\bar{f}))\}_{i=1}^K = \sum_i \sigma(f'_i)\log(\sigma(f'_i)/\sigma(\bar{f}))$ ;    // diversity (Eq.(6))
    $\{z'_i, b'_i\}_{i=1}^K \leftarrow \arg\max_{z,b} \mathcal{S}_{inf} + \mathcal{S}_{div}$ ;        // guided latent optimization (Eq.(2))
    **for** *i=1,...,K* **do**
        $f''_i = \mathcal{P}_{f,\epsilon}((1+z'_i)f + b'_i)$ ;            // guided noise perturbation (Eq.(1))
        $x''_i = G(f''_i)$ ;                      // sample creation
        Add $x''_i \rightarrow \mathcal{D}_s$.
    **end**
**end**
**Output:** Expanded dataset $\mathcal{D}_o \cup \mathcal{D}_s$.

---

**More implementation details**. In our experiment, DALL-E2 is pre-trained on Laion-400M [66] and then used for dataset expansion. The resolution of the created images by GIF-DALLE is $64\times64$ for model training without further super-resolution. Only when visualizing the created images, we use super-resolution to up-sample the generated images to $256\times256$ for clarification. Moreover, we set $\varepsilon = 0.1$ in the guided latent feature optimization. During the diffusion process, we set the guidance scale as 4 and adopt the DDIM sampler [70] for 100-step diffusion. For expanding medical image datasets, it is necessary to fine-tune the prior model for alleviating domain shifts.

## D.2 Method details of GIF-SD

GIF-SD applies Stable Diffusion (SD) [59] as its prior model. As its encoder differs from the CLIP image encoder, we slightly modify the pipeline of GIF-SD.

**Pipeline**. As shown in Algorithm 2, GIF-SD first generates a latent feature for the seed image via its image encoder. Following that, GIF-SD conducts prompt-based diffusion for the latent feature, where the generation rule of prompts will be elaborated in Eq. (7). Please note that, with a suitable prompt design, the prompt-based diffusion helps to create more diversified samples. Afterward, GIF-SD conducts *channel-wise* noise perturbation. Here, the latent feature of SD has three dimensions: two spatial dimensions and one channel dimension. As discussed in our preliminary (cf. Appendix B.2), the channel-level latent feature encodes more subtle style information, whereas the spatial-level latent features encode more content information. In light of the findings in this preliminary study, we particularly conduct channel-level noise to optimize the latent features in GIF-SD for further diversifying the style of the generated images while maintaining the content semantics of the latent features (after prompt-guided diffusion) unchanged. Based on the randomly perturbed feature, GIF-SD generates an intermediate image via its image decoder and applies CLIP to conduct zero-shot prediction for both the seed and the intermediate image to compute the guidance. With the guidance, GIF-SD optimizes the latent features for creating more style-diverse samples. Here, GIF-SD conducts guided imagination on its own latent space.

---

**Algorithm 2:** GIF-SD Algorithm

---

**Input:** Original small dataset $\mathcal{D}_o$; SD image encoder $f(\cdot)$ and image decoder $G(\cdot)$; SD diffusion module $f_{\text{diff}}(\cdot; [prompt])$; CLIP image encoder $f_{\text{CLIP-I}}(\cdot)$; CLIP zero-shot classifier $w(\cdot)$; Expansion ratio $K$; Perturbation constraint $\varepsilon$.

**Initialize**: Synthetic data set $\mathcal{D}_s = \emptyset$;

**for** $x \in \mathcal{D}_o$ **do**

 $\mathcal{S}_{inf} = 0$;

 $f = f(x)$ ;        // latent feature encoding for seed sample

 Randomly sample a $[prompt]$ ;      // Prompt generation (Eq.(7))

 $f = f_{\text{diff}}(f; [prompt])$ ;        // SD latent diffusion

 $s = w(f_{\text{CLIP-I}}(x))$ ;     // CLIP zero-shot prediction for seed sample

 **for** *i=1,...,K* **do**

  Initialize noise $z_i \sim \mathcal{U}(0, 1)$ and bias $b_i \sim \mathcal{N}(0, 1)$;

  $f'_i = \mathcal{P}_{f,\epsilon}((1 + z_i)f + b_i)$ ;     // noise perturbation (Eq.(1))

  $s' = w(f'_i)$ ;        // CLIP zero-shot prediction

  $\mathcal{S}_{inf} += s'_j + (s \log(s) - s' \log(s'))$, s.t. $j = \arg\max(s)$ ;   // class-maintained informativeness (Eq.(5))

 **end**

 $\bar{f} = mean(\{f'_i\}_{i=1}^K)$;

 $\mathcal{S}_{div} = \sum_i \{\mathcal{D}_{KL}(\sigma(f'_i) \| \sigma(\bar{f}))\}_{i=1}^K = \sum_i \sigma(f'_i) \log(\sigma(f'_i)/\sigma(\bar{f}))$ ;  // diversity (Eq.(6))

 $\{z'_i, b'_i\}_{i=1}^K \leftarrow \arg\max_{z,b} \mathcal{S}_{inf} + \mathcal{S}_{div}$ ;   // guided latent optimization (Eq.(2))

 **for** *i=1,...,K* **do**

  $f''_i = \mathcal{P}_{f,\epsilon}((1 + z'_i)f + b'_i)$ ;     // guided noise perturbation (Eq.(1))

  $x''_i = G(f''_i)$ ;        // sample creation

  Add $x''_i \rightarrow \mathcal{D}_s$.

 **end**

**end**

**Output:** Expanded dataset $\mathcal{D}_o \cup \mathcal{D}_s$.

---

**Rule of prompt design**. In our preliminary studies in Appendix B.3, we find that domain labels, class labels, and adjective words are necessary to make the prompts semantically effective. Therefore, we design the prompts using the following rule:

$$\text{Prompt} := [\text{domain}] \text{ of a(n) } [\text{adj}] [\text{class}]. \tag{7}$$

For example, "an oil painting of a colorful fox". To enable the prompts to be diversified, inspired by our preliminary studies, we design a set of domain labels and adjective words for natural image datasets as follows.

- Domain label set: ["an image of", "a real-world photo of", "a cartoon image of", "an oil painting of", "a sketch of"]

- Adjective word set: [" ", "colorful", "stylized", "high-contrast", "low-contrast", "posterized", "solarized", "sheared", "bright", "dark"]

For a seed sample, we randomly sample a domain label and an adjective word from the above sets to construct a prompt. Note that, for medical image datasets, we cancel the domain label set and replace it as the modality of the medical images, *e.g.,* ["Abdominal CT image of"], ["Colon pathological image of"].

**Implementation details**. In our experiment, we implement GIF-SD based on CLIP VIT-B/32 and Stable Diffusion v1-4, which are pre-trained on large datasets and then used for dataset expansion. Here, we use the official checkpoints of CLIP VIT-B/32 and Stable Diffusion v1-4. The resolution of the created images by GIF-SD is $512{\times}512$ for all datasets. Moreover, for guided latent feature optimization in GIF-SD, we set $\varepsilon = 0.8$ for natural image datasets and $\varepsilon = 0.1$ for medical image datasets. Here, we further adjust $\varepsilon = 4$ for Caltech101 to increase image diversity for better performance. During the diffusion process, we adopt the DDIM sampler [70] for 50-step latent diffusion. Moreover, the hyper-parameters of strength and scale in SD depend on datasets, while more analysis is provided in Appendix B.4. Note that, for expanding medical image datasets, it is necessary to fine-tune the prior model for alleviating domain shifts.

### D.3 Method details of GIF-MAE

Thanks to strong image reconstruction abilities, our GIF-MAE applies the MAE-trained model [24] as its prior model. As its encoder is different from the CLIP image encoder, we slightly modify the pipeline of GIF-MAE.

**Pipeline**. As shown in Algorithm 3, GIF-MAE first generates a latent feature for the seed image via its encoder, and conducts *channel-wise* noise perturbation. Here, the latent feature of MAE has two dimensions: spatial dimension and channel dimension. As discussed in our preliminary (cf. Appendix B.2), the channel-level latent feature encodes more subtle style information, whereas the token-level latent feature encodes more content information. Motivated by the findings in this preliminary study, we particularly conduct channel-level noise to optimize the latent features in our GIF-MAE method for maintaining the content semantics of images unchanged. Based on the perturbed feature, GIF-MAE generates an intermediate image via its decoder and applies CLIP to conduct zero-shot prediction for both the seed and the intermediate image to compute the guidance. With the guidance, GIF-MAE optimizes the latent features for creating content-consistent samples of diverse styles. Here, GIF-MAE conducts guided imagination on its own latent space.

---

**Algorithm 3:** GIF-MAE Algorithm

**Input:** Original small dataset $\mathcal{D}_o$; MAE image encoder $f(\cdot)$ and image decoder $G(\cdot)$; CLIP image encoder
$f_{\text{CLIP-I}}(\cdot)$; CLIP zero-shot classifier $w(\cdot)$; Expansion ratio $K$; Perturbation constraint $\varepsilon$.
**Initialize**: Synthetic data set $\mathcal{D}_s = \emptyset$;
**for** $x \in \mathcal{D}_o$ **do**

    $\mathcal{S}_{inf} = 0$;
    $f = f(x)$ ;                 // latent feature encoding for seed sample
    $s = w(f_{\text{CLIP-I}}(x))$ ;         // CLIP zero-shot prediction for seed sample
    **for** *i=1,...,K* **do**

        Initialize noise $z_i \sim \mathcal{U}(0,1)$ and bias $b_i \sim \mathcal{N}(0,1)$;
        $f_i' = \mathcal{P}_{f,\epsilon}((1+z_i)f + b_i)$ ;     // channel-level noise perturbation (Eq.(1))
        $x_i' = G(f_i')$ ;                // intermediate image generation
        $s' = w(f_{\text{CLIP-I}}(x_i'))$;
        $\mathcal{S}_{inf} \mathrel{+}= s_j' + (s\log(s) - s'\log(s'))$, s.t. $j = \arg\max(s)$ ;    // class-maintained
         informativeness (Eq.(5))
    **end**
    $\bar{f} = mean(\{f_i'\}_{i=1}^K)$;
    $\mathcal{S}_{div} = \sum_i \{\mathcal{D}_{KL}(\sigma(f_i') \| \sigma(\bar{f}))\}_{i=1}^K = \sum_i \sigma(f_i') \log(\sigma(f_i')/\sigma(\bar{f}))$ ;   // diversity (Eq.(6))
    $\{z_i', b_i'\}_{i=1}^K \leftarrow \arg\max_{z,b} \mathcal{S}_{inf} + \mathcal{S}_{div}$ ;      // guided latent optimization (Eq.(2))
    **for** *i=1,...,K* **do**

        $f_i'' = \mathcal{P}_{f,\epsilon}((1+z_i')f + b_i')$ ;  // guided channel-wise noise perturbation (Eq.(1))
        $x_i'' = G(f_i'')$ ;                   // sample creation
        Add $x_i'' \to \mathcal{D}_s$.
    **end**
**end**
**Output:** Expanded dataset $\mathcal{D}_o \cup \mathcal{D}_s$.

---

**Implementation details**. In our experiment, we implement GIF-MAE based on CLIP VIT-B/32 and MAE VIT-L/16, which are pre-trained on large datasets and then fixed for dataset expansion. Here, we use the official checkpoints of CLIP VIT-B/32 and MAE VIT-L/16. The resolution of the created images by GIF-MAE is $224 \times 224$ for all datasets. Moreover, we set $\varepsilon = 5$ for guided latent feature optimization in GIF-MAE.

### D.4 Implementation details of model training

We implement GIF in PyTorch based on CLIP VIT-B/32, DALL-E2, MAE VIT-L/16, and Stable Diffusion (SD) V1-4, which are pre-trained on large datasets and then fixed for dataset expansion. We use the official checkpoints of CLIP VIT-B/32, MAE VIT-L/16, and SD v1-4, and use the DALL-E2 pre-trained on Laion-400M [66]. On medical datasets, since DALL-E2 and SD were initially trained on natural images and suffer from domain shifts to medical domains (please see the discussion in Appendix B.6), we fine-tune them on the target dataset before dataset expansion.

To fairly evaluate the expansion effectiveness of different methods, we use them to expand the original small datasets by the same ratios, followed by training models from scratch on the expanded dataset with the same number of epochs and the same data pre-processing. In this way, the models are trained with the same number of update steps, so that all expansion methods are fairly compared. The expansion ratio depends on the actual demand of real applications. In the experiment of Table 1, CIFAR100-Subset is expanded by $5\times$, Pets is expanded by $30\times$, and all other datasets are expanded by $20\times$. Moreover, all medical image datasets are expanded by $5\times$. In addition, all augmentation baselines expand datasets with the same expansion ratio for fair comparisons.

After expansion, we train ResNet-50 [25] from scratch for 100 epochs based on the expanded datasets. During model training, we process images via random resize to $224\times224$ through bicubic sampling, random rotation, and random flips. If not specified, we use the SGD optimizer with a momentum of 0.9. We set the initial learning rate (LR) to 0.01 with cosine LR decay, except the initial LR of CIFAR100-Subset and OrganSMNIST is 0.1. The model performance is averaged over three runs in terms of micro accuracy on natural image datasets and macro accuracy on medical image datasets.

### D.5 Discussions on limitations and broader impact

**Limitations**. We next discuss the limitations of our method.

1. **Performance of generated samples**. The expanded samples are still less informative than real samples. For example, a ResNet-50 trained from scratch on our 5x-expanded CIFAR100-Subset achieves an accuracy of 61.1%, which lags behind the 71.0% accuracy on the original CIFAR100. This gap signals the potential for advancing algorithmic dataset expansion. Please see Appendix F.6 for detailed discussions. We expect that this pioneering work can inspire more studies to explore dataset expansion so that it can even outperform a human-collected dataset of the same size.

2. **Quality of generated samples**. Some samples might have noise, as exemplified in Figure 5b. Despite seeming less realistic, those samples are created following our guidance (e.g., class-maintained informativeness boosting). This ensures the class consistency of these samples, mitigating potential negative effects on model training. Nonetheless, refining the expansion method to address these noisy cases can further enhance the effectiveness of dataset expansion.

3. **Scope of work**. Our current focus is predominantly on image classification. Exploring the adaptability of our method to other tasks, such as object detection and semantic segmentation, is an intriguing next step.

**Broader impact**. We further summarize our broader impact. Our method can offer a notable reduction in the time and cost associated with manual data collection and annotation for dataset expansion, as discussed in Section 5.1. This can revolutionize how small datasets are expanded, making deep learning more accessible to scenarios with limited data availability (cf. Table 1).

# E Dataset Statistics

**The statistics of natural image datasets.** We evaluate our method on six small-scale natural image datasets, including Caltech-101 [18], CIFAR100-Subset [41], Standard Cars [40], Oxford 102 Flowers [49], Oxford-IIIT Pets [50] and DTD [10]. Here, CIFAR100-Subset is an artificial dataset for simulating small-scale datasets by randomly sampling 100 instances per class from the original CIFAR100 dataset, and the total sample number is 10,000. These datasets cover a wide range of classification tasks, including coarse-grained object classification (*i.e.,* CIFAR100-Subset and Caltech-101), fine-grained object classification (*i.e.,* Cars, Flowers and Pets) and texture classification (*i.e.,* DTD). The data statistics of these natural image datasets are given in Table 10. Note that the higher number of classes or the lower number of average samples per class a dataset has, the more challenging the dataset is.

Table 10: Statistics of small-scale natural image datasets.

| Datasets | Tasks | # Classes | # Samples | # Average samples per class |
|---|---|---|---|---|
| Caltech101 | Coarse-grained object classification | 102 | 3,060 | 30 |
| CIFAR100-Subset | Coarse-grained object classification | 100 | 10,000 | 100 |
| Standard Cars | Fine-grained object classification | 196 | 8,144 | 42 |
| Oxford 102 Flowers | Fine-grained object classification | 102 | 6,552 | 64 |
| Oxford-IIIT Pets | Fine-grained object classification | 37 | 3,842 | 104 |
| Describable Textures (DTD) | Texture classification | 47 | 3,760 | 80 |

**The statistics of medical image datasets.** To evaluate the effect of dataset expansion on medical images, we conduct experiments on three small-scale medical image datasets. These datasets cover a wide range of medical image modalities, including breast ultrasound (*i.e.,* BreastMNIST [1]), colon pathology (*i.e.,* PathMNIST [36]), and Abdominal CT (*i.e.,* OrganSMNIST [88]). We provide detailed statistics for these datasets in Table 11.

Table 11: Statistics of small-scale medical image datasets. To better simulate the scenario of small medical datasets, we use the validation sets of BreastMNIST and PathMNIST for experiments instead of training sets, whereas OrganSMNIST is based on its training set.

| Datasets | Data Modality | # Classes | # Samples | # Average samples per class |
|---|---|---|---|---|
| BreastMNIST [1, 90] | Breast Ultrasound | 2 | 78 | 39 |
| PathMNIST [36, 90] | Colon Pathology | 9 | 10,004 | 1,112 |
| OrganSMNIST [88, 90] | Abdominal CT | 11 | 13,940 | 1,267 |

# F   More Experimental Results and Discussion

## F.1   More comparisons to expansion with augmentations

### F.1.1   More results on expansion efficiency

In Figure 4, we have demonstrated the expansion efficiency of our proposed GIF over Cutout, GridMask and RandAugment on the Cars, DTD and Pets datasets. Here, we further report the results on Caltech101, Flowers, and CIFAR100-Subset datasets. As shown in Figure 18, 5× expansion by GIF-SD and GIF-DALLE has already performed comparably to 20× expansion of these augmentation methods, while 10× expansion by GIF-SD and GIF-DALLE outperforms 20× expansion by these data augmentation methods a lot. This result further demonstrates the effectiveness and efficiency of our GIF, and also reflects the importance of automatically creating informative synthetic samples for model training.

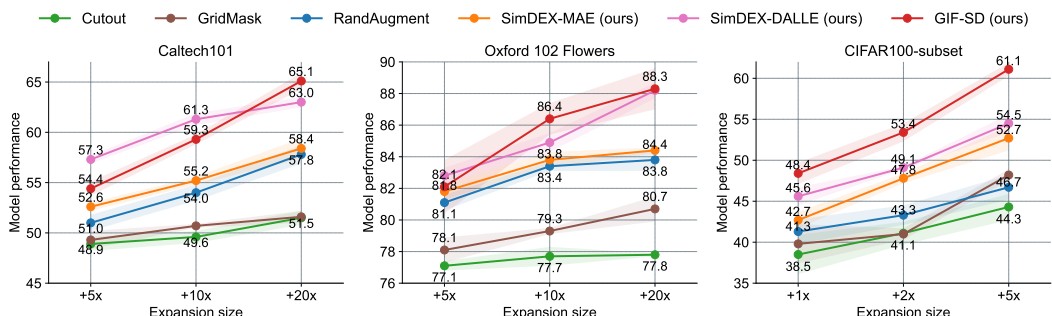

Figure 18: Accuracy of ResNet-50 trained from scratch on the expanded datasets with different expansion ratios based on Caltech101, Flowers, and CIFAR100-Subset datasets.

### F.1.2   Comparison to Mixup and CutMix

We further compare our method to more advanced augmentation methods. Specifically, we apply Mixup-based methods, *i.e.,* Mixup [99] and CutMix [96], to expand CIFAR100-Subset by 5 × and use the expanded dataset to train the model from scratch. As shown in Table 12, GIF-SD performs much better than Mixup and CutMix, further demonstrating the superiority of our method over augmentation-based expansion methods.

Table 12: Comparison between GIF and Mixup methods for expanding CIFAR100-Subset by 5×.

| CIFAR100-Subset | Accuracy |
|---|---|
| *Original dataset* | $35.0_{\pm 1.7}$ |
| *Expanded dataset* | |
| *5×-expanded* by Mixup [99] | $45.6_{\pm 1.2}$ |
| *5×-expanded* by CutMix [96] | $50.7_{\pm 0.2}$ |
| *5×-expanded* by GIF-SD | $\mathbf{61.1}_{\pm 0.8}$ |

### F.1.3 Comparison with an advanced generative method

We further compare our method with an advanced generative method [26] for dataset expansion. This method includes strategies like language enhancement (LE) [26] and CLIP Filter (CF) [26]. We use this method to expand the CIFAR100-S dataset based on Stable Diffusion (SD). As shown in the following table, SD combined with the method [26] is still noticeably inferior to our GIF-SD for both training from scratch and CLIP tuning. This further demonstrates the superiority of our method.

Table 13: Comparison between GIF and the method [26] for expanding CIFAR100-Subset by 5×.

| CIFAR100-S | Training from scratch | CLIP fine-tuning |
|---|---|---|
| Original dataset | 35.0 | 75.2 |
| 5x-expanded dataset by SD+method [26] | 55.1 (+20.1) | 77.0 (+1.8) |
| 5x-expanded dataset by GIF-SD (ours) | 61.1 (+26.1) | 79.4 (+4.2) |

### F.1.4 Comparison to infinite data augmentation

The training time varies based on the specific datasets. However, it is pivotal to note that all dataset expansion methods were compared based on the same expansion ratio, thus ensuring consistent training time/cost and fair comparisons. We acknowledge that training on an expanded dataset will inevitably take longer than training on the original dataset. However, as shown in Table 1 (cf. Section 5.1), the significant improvement in model performance (*i.e.,* by 36.9% on average over six natural image datasets and by 13.5% on average over three medical datasets) makes the increased investment in training time worthwhile.

Despite this, one may wonder how the explored dataset expansion would perform compared to training with infinite data augmentation. Therefore, in this appendix, we further evaluate the performance of infinite data augmentation on the CIFAR100-Subset. Specifically, based on RandAugment, we train ResNet-50 using infinite online augmentation for varying numbers of epochs from 100 to 700. As shown in Table 14, using RandAugment to train models for more epochs leads to better performance, but gradually converges (around 51% accuracy at 500 epochs) and keeps fluctuating afterward. By contrast, our proposed method proves advantageous with the same training consumption costs: training the model on the original CIFAR100-S dataset for 5x more epochs performs much worse than the model trained on our 5x-expanded dataset. This comparison further underscores the effectiveness of our method in achieving higher accuracy without inflating training costs.

Table 14: Comparison between GIF-SD and infinite data augmentation on CIFAR100-Subset. Here, consumption costs equal data number × training epoch.

| Methods | Epochs | Consumption | Accuracy |
|---|---|---|---|
| *Original* | | | |
| Standard training | 100 | 1 million | $35.0_{\pm 1.7}$ |
| Training with RandAugment | 100 | 1 million | $39.6_{\pm 2.5}$ |
| Training with RandAugment | 200 | 2 million | $46.9_{\pm 0.9}$ |
| Training with RandAugment | 300 | 3 million | $48.1_{\pm 0.6}$ |
| Training with RandAugment | 400 | 4 million | $49.6_{\pm 0.4}$ |
| Training with RandAugment | 500 | 5 million | $51.3_{\pm 0.3}$ |
| Training with RandAugment | 600 | 6 million | $51.1_{\pm 0.3}$ |
| Training with RandAugment | 700 | 7 million | $50.6_{\pm 1.1}$ |
| *Expanded* | | | |
| *5×-expanded* by GIF-SD | 100 | 6 million | $\mathbf{61.1}_{\pm 0.8}$ |

### F.1.5 Discussion of picking related samples from larger datasets

Picking and labeling data from larger image datasets with CLIP is an interesting idea for dataset expansion. However, such a solution is limited in real applications, since a large-scale related dataset may be unavailable in many image domains (*e.g.,* medical image domains). Moreover, selecting data from different image domains (*e.g.,* from natural images to medical images) is unhelpful for dataset expansion. Despite the above limitations in real applications, we also evaluate this idea on CIFAR100-Subset and investigate whether it helps dataset expansion when there is a larger dataset of the same image nature, *e.g.,* ImageNet. Here, we use CLIP to select and annotate related images from ImageNet to expand CIFAR100-Subset. Specifically, we scan over all ImageNet images and use CLIP to predict them to the class of CIFAR100-Subset. We select the samples with the highest prediction probability higher than 0.1 and expand each class by 5×. As shown in Table 15, the idea of picking related images from ImageNet makes sense, but performs worse than our proposed method. This result further demonstrates the effectiveness and superiority of our method. In addition, how to better transfer large-scale datasets to expand small datasets is an interesting open question, and we expect to explore it in the future.

Table 15: Comparison between GIF and picking related data from ImageNet for expanding CIFAR100-Subset by 5×.

| CIFAR100-Subset | Accuracy |
|---|---|
| *Original dataset* | $35.0_{\pm 1.7}$ |
| *Expanded dataset* | |
| *5×-expanded* by picking data from ImageNet with CLIP | $50.9_{\pm 1.1}$ |
| *5×-expanded* by GIF-SD | $\mathbf{61.1}_{\pm 0.8}$ |

## F.2 More results of benefits to model generalization

In CIFAR100-C [27], there are 15 types of OOD corruption (as shown in Table 16), *i.e.,* Gaussian noise, shot noise, impulse noise, defocus blur, glass blue, motion blur, zoom blur, snow, frost, fog, brightness, contrast, elastic transformation, pixelation, and JPEG compression. Each corruption type has 5 different severity levels: the larger severity level means more severe distribution shifts between CIFAR100 and CIFAR100-C. In Table 2 of the main paper, we have shown the empirical benefit of our method to model out-of-distribution (OOD) generalization based on CIFAR100-C with the severity level 3. Here, we further report its performance on CIFAR100-C with other severity levels. As shown in Table 16, our method is able to achieve consistent performance gains across all severity levels, which further verifies the benefits of GIF to model OOD generalization.

Table 16: Corruption Accuracy of ResNet-50 trained from scratch on CIFAR100-S and our 5× expanded dataset, under 15 types of corruption in CIFAR100-C with various severity levels.

(a) CIFAR100-C with the severity level 1

| | Noise | | | Blur | | | | Weather | | | | Digital | | | | |
| Dataset | Gauss. | Shot | Impul. | Defoc. | Glass | Motion | Zoom | Snow | Frost | Fog | Brit. | Contr. | Elastic | Pixel | JPEG | Average |
|---|---|---|---|---|---|---|---|---|---|---|---|---|---|---|---|---|
| *Original* | 25.6 | 29.3 | 25.0 | 34.2 | **32.2** | 31.7 | 30.9 | 32.3 | 28.3 | 31.8 | 33.7 | 29.2 | 31.7 | 34.1 | 30.9 | 30.7 |
| *5×-expanded* by GIF-SD | 50.3 | 54.6 | 50.8 | 59.2 | 29.4 | 53.7 | 51.9 | 53.1 | 54.0 | 58.7 | 59.5 | 57.1 | 52.5 | 57.9 | **54.7** | 53.2 (+22.5) |
| *20×-expanded* by GIF-SD | **55.0** | **60.5** | **54.8** | **66.1** | 30.2 | **56.0** | **58.0** | **61.1** | **62.2** | **65.1** | **66.2** | **64.3** | **59.2** | **63.8** | 60.8 | **58.9** (+27.2) |

(b) CIFAR100-C with the severity level 2

| | Noise | | | Blur | | | | Weather | | | | Digital | | | | |
| Dataset | Gauss. | Shot | Impul. | Defoc. | Glass | Motion | Zoom | Snow | Frost | Fog | Brit. | Contr. | Elastic | Pixel | JPEG | Average |
|---|---|---|---|---|---|---|---|---|---|---|---|---|---|---|---|---|
| *Original* | 18.6 | 24.4 | 17.4 | 32.5 | **31.9** | 28.3 | 29.8 | 28.4 | 22.9 | 23.6 | 31.1 | 16.3 | 30.8 | 33.7 | 29.2 | 26.6 |
| *5×-expanded* by GIF-SD | 39.5 | 48.8 | 41.7 | 56.3 | 29.6 | 46.4 | 49.7 | 45.2 | 46.4 | 52.8 | 57.6 | 45.5 | 52.1 | 54.2 | 51.1 | 47.8 (+21.2) |
| *20×-expanded* by GIF-SD | **42.7** | **53.7** | **43.9** | **63.1** | 31.2 | **51.8** | **56.1** | **52.0** | **54.9** | **60.4** | **65.2** | **54.3** | **59.2** | **60.0** | **55.6** | **52.3** (+25.7) |

(c) CIFAR100-C with the severity level 3

| | Noise | | | Blur | | | | Weather | | | | Digital | | | | |
| Dataset | Gauss. | Shot | Impul. | Defoc. | Glass | Motion | Zoom | Snow | Frost | Fog | Brit. | Contr. | Elastic | Pixel | JPEG | Average |
|---|---|---|---|---|---|---|---|---|---|---|---|---|---|---|---|---|
| *Original* | 12.8 | 17.0 | 12.5 | 30.5 | 31.7 | 25.2 | 28.6 | 26.5 | 19.0 | 18.6 | 28.3 | 11.5 | 29.5 | 33.6 | 28.8 | 23.6 |
| *5×-expanded* by GIF-SD | 29.7 | 36.4 | 32.7 | 51.9 | 32.4 | 39.2 | 46.0 | 45.3 | 38.1 | 47.1 | 55.7 | 37.3 | 48.6 | 53.2 | 49.4 | 43.3 (+19.3) |
| *20×-expanded* by GIF-SD | **31.8** | **39.2** | **34.7** | **58.4** | 33.4 | **43.1** | **51.9** | **51.7** | **47.4** | **55.0** | **63.3** | **46.5** | **54.9** | **58.0** | **53.6** | **48.2** (+24.6) |

(d) CIFAR100-C with the severity level 4

| | Noise | | | Blur | | | | Weather | | | | Digital | | | | |
| Dataset | Gauss. | Shot | Impul. | Defoc. | Glass | Motion | Zoom | Snow | Frost | Fog | Brit. | Contr. | Elastic | Pixel | JPEG | Average |
|---|---|---|---|---|---|---|---|---|---|---|---|---|---|---|---|---|
| *Original* | 10.8 | 14.3 | 7.7 | 28.5 | **29.3** | 25.2 | 27.8 | 23.3 | 19.5 | 14.1 | 24.9 | 7.4 | 29.0 | 33.0 | 28.1 | 21.5 |
| *5×-expanded* by GIF-SD | 25.3 | 31.2 | 18.0 | 45.1 | 21.4 | 39.6 | 42.5 | 41.7 | 37.7 | 40.2 | 52.1 | 26.1 | 44.2 | 47.8 | 48.2 | 37.4 (+15.9) |
| *20×-expanded* by GIF-SD | **27.4** | **33.7** | **20.2** | **50.7** | 21.7 | **43.9** | **47.8** | **48.8** | **46.7** | **47.6** | **60.7** | **35.3** | **47.9** | **49.3** | **51.2** | **42.2** (+20.7) |

(e) CIFAR100-C with the severity level 5

| | Noise | | | Blur | | | | Weather | | | | Digital | | | | |
| Dataset | Gauss. | Shot | Impul. | Defoc. | Glass | Motion | Zoom | Snow | Frost | Fog | Brit. | Contr. | Elastic | Pixel | JPEG | Average |
|---|---|---|---|---|---|---|---|---|---|---|---|---|---|---|---|---|
| *Original* | 9.4 | 10.7 | 5.5 | 24.9 | **28.9** | 22.3 | 25.9 | 19.4 | 16.6 | 8.2 | 18.3 | 2.7 | 29.0 | 31.8 | 27.3 | 18.7 |
| *5×-expanded* by GIF-SD | 21.4 | 23.8 | 10.8 | 31.8 | 22.8 | 33.1 | 37.6 | 38.1 | 31.1 | 24.7 | 43.7 | 8.6 | 38.6 | **36.0** | 45.6 | 29.8 (+11.1) |
| *20×-expanded* by GIF-SD | **22.9** | **25.5** | **11.1** | **33.5** | 24.1 | **36.2** | **41.8** | **46.4** | **38.4** | **32.1** | **53.5** | **13.9** | **40.4** | 32.0 | **48.8** | **33.4** (+14.7) |

### F.3 More results of applicability to various model architectures

In Table 3, we have demonstrated the generalizability of our expanded Cars dataset to various model architectures. Here, we further apply the expanded Caltech101, Flowers, DTD, CIFAR100-S, and Pets datasets ($5\times$ expansion ratio) by GIF-SD and GIF-DALLE to train ResNeXt-50 [85], WideResNet-50 [97] and MobileNet V2 [64] from scratch. Table 17 shows that our expanded datasets bring consistent performance gains for all the architectures on all datasets. This further affirms the versatility of our expanded datasets, which, once expanded, are readily suited for training various model architectures.

Table 17: Model performance of various model architectures trained on $5\times$ expanded natural image datasets by GIF.

| Dataset | Caltech101 [18] | | | | |
| --- | --- | --- | --- | --- | --- |
| | ResNet-50 | ResNeXt-50 | WideResNet-50 | MobilteNet-v2 | Avg. |
| *Original dataset* | $26.3_{\pm1.0}$ | $32.6_{\pm0.5}$ | $34.7_{\pm0.8}$ | $33.8_{\pm1.1}$ | 31.9 |
| *5×-expanded* by GIF-DALLE | $\mathbf{57.3}_{\pm0.4}$ | $\mathbf{55.2}_{\pm0.1}$ | $\mathbf{61.8}_{\pm0.5}$ | $\mathbf{59.4}_{\pm0.7}$ | $\mathbf{58.4}$ (+26.5) |
| *5×-expanded* by GIF-SD | $54.4_{\pm0.7}$ | $52.8_{\pm1.1}$ | $60.7_{\pm0.3}$ | $55.6_{\pm0.5}$ | 55.9 (+24.0) |

| Dataset | Cars [40] | | | | |
| --- | --- | --- | --- | --- | --- |
| | ResNet-50 | ResNeXt-50 | WideResNet-50 | MobilteNet-v2 | Avg. |
| *Original dataset* | $19.8_{\pm0.9}$ | $18.4_{\pm0.5}$ | $32.0_{\pm0.8}$ | $26.2_{\pm4.2}$ | 24.1 |
| *5×-expanded* by GIF-DALLE | $53.1_{\pm0.2}$ | $43.7_{\pm0.2}$ | $60.0_{\pm0.6}$ | $47.8_{\pm0.6}$ | 51.2 (+27.1) |
| *5×-expanded* by GIF-SD | $\mathbf{60.6}_{\pm1.9}$ | $\mathbf{64.1}_{\pm1.3}$ | $\mathbf{75.1}_{\pm0.4}$ | $\mathbf{60.2}_{\pm1.6}$ | $\mathbf{65.0}$ (+40.9) |

| Dataset | Flowers [49] | | | | |
| --- | --- | --- | --- | --- | --- |
| | ResNet-50 | ResNeXt-50 | WideResNet-50 | MobilteNet-v2 | Avg. |
| *Original dataset* | $74.1_{\pm0.1}$ | $75.8_{\pm1.2}$ | $79.3_{\pm1.6}$ | $85.5_{\pm1.0}$ | 78.7 |
| *5×-expanded* by GIF-DALLE | $\mathbf{82.8}_{\pm0.5}$ | $81.6_{\pm0.4}$ | $84.6_{\pm0.2}$ | $88.8_{\pm0.5}$ | 84.4 (+5.7) |
| *5×-expanded* by GIF-SD | $82.1_{\pm1.7}$ | $\mathbf{82.0}_{\pm1.2}$ | $\mathbf{85.0}_{\pm0.6}$ | $\mathbf{89.0}_{\pm0.1}$ | $\mathbf{84.5}$ (+5.8) |

| Dataset | DTD [10] | | | | |
| --- | --- | --- | --- | --- | --- |
| | ResNet-50 | ResNeXt-50 | WideResNet-50 | MobilteNet-v2 | Avg. |
| *Original dataset* | $23.1_{\pm0.2}$ | $25.4_{\pm0.6}$ | $26.1_{\pm0.6}$ | $28.1_{\pm0.9}$ | 25.7 |
| *5×-expanded* by GIF-DALLE | $31.2_{\pm0.9}$ | $30.6_{\pm0.1}$ | $35.3_{\pm0.9}$ | $37.4_{\pm0.8}$ | 33.6 (+7.9) |
| *5×-expanded* by GIF-SD | $\mathbf{33.9}_{\pm0.9}$ | $\mathbf{33.3}_{\pm1.6}$ | $\mathbf{40.6}_{\pm1.7}$ | $\mathbf{40.8}_{\pm1.1}$ | $\mathbf{37.2}$ (+11.5) |

| Dataset | CIFAR100-S [41] | | | | |
| --- | --- | --- | --- | --- | --- |
| | ResNet-50 | ResNeXt-50 | WideResNet-50 | MobilteNet-v2 | Avg. |
| *Original dataset* | $35.0_{\pm3.2}$ | $36.3_{\pm2.1}$ | $42.0_{\pm0.3}$ | $50.9_{\pm0.2}$ | 41.1 |
| *5×-expanded* by GIF-DALLE | $54.5_{\pm1.1}$ | $52.4_{\pm0.7}$ | $55.3_{\pm0.3}$ | $56.2_{\pm0.2}$ | 54.6 (+13.5) |
| *5×-expanded* by GIF-SD | $\mathbf{61.1}_{\pm0.8}$ | $\mathbf{59.0}_{\pm0.7}$ | $\mathbf{64.4}_{\pm0.2}$ | $\mathbf{62.4}_{\pm0.1}$ | $\mathbf{61.4}$ (+20.3) |

| Dataset | Pets [50] | | | | |
| --- | --- | --- | --- | --- | --- |
| | ResNet-50 | ResNeXt-50 | WideResNet-50 | MobilteNet-v2 | Avg. |
| *Original dataset* | $6.8_{\pm1.8}$ | $19.0_{\pm1.6}$ | $22.1_{\pm0.5}$ | $37.5_{\pm0.4}$ | 21.4 |
| *5×-expanded* by GIF-DALLE | $46.2_{\pm0.1}$ | $52.3_{\pm1.5}$ | $66.2_{\pm0.1}$ | $60.3_{\pm0.3}$ | 56.3 (+34.9) |
| *5×-expanded* by GIF-SD | $\mathbf{65.8}_{\pm0.6}$ | $\mathbf{56.5}_{\pm0.6}$ | $\mathbf{70.9}_{\pm0.4}$ | $\mathbf{60.6}_{\pm0.5}$ | $\mathbf{63.5}$ (+42.1) |

## F.4 More discussions on CLIP

In the following subsections, we provide more discussions on the comparisons with CLIP.

### F.4.1 Why not directly transfer CLIP models to target datasets?

In our proposed GIF framework, we leverage the pre-trained CLIP model to guide dataset expansion. An inevitable question might be: why not directly use or transfer the CLIP model to the target dataset, especially given its proven effectiveness on many natural image datasets? Before delving into that, it is important to note that we aim to tackle small-data scenarios, where only a limited-size dataset is available and there are no large-scale external datasets with a similar nature to the target dataset. Consequently, training a new CLIP model on the target dataset (*e.g.,* in the medical image domains) is not feasible. Therefore, we rely on *publicly available CLIP models* for dataset expansion. Compared to directly using or transferring CLIP models, our dataset expansion introduces a necessary new paradigm for two primary reasons as follows.

First, our GIF method has better applicability to scenarios across various image domains. While CLIP demonstrates good transfer performance on certain natural image datasets, it struggles to achieve this performance on other domains, such as medical image datasets. To illustrate this, we test the linear-probing and fine-tuning performance of the CLIP-trained ResNet-50 model on three medical datasets. As shown in Table 18, directly employing or transferring the CLIP model yielded unsatisfactory results or only marginally improved performance—significantly underperforming compared to our dataset expansion approach. The limited transfer performance is attributed to the fact that, when the pre-trained datasets are highly different from the target datasets, the pre-training weights do not significantly bolster performance compared to training from scratch [55]. Such an issue cannot be resolved by conducting CLIP pre-training on these domains, since there is no large-scale dataset of similar data nature to the target dataset in real scenarios. In contrast, our GIF framework is capable of generating images of similar nature as the target data for dataset expansion, enhancing its applicability to real-world scenarios across diverse image domains.

Second, our dataset expansion can provide expanded datasets suitable for training various network architectures. In certain practical scenarios, such as mobile terminals, the permissible model size is severely limited due to hardware constraints. Nonetheless, the publicly available CLIP checkpoints are restricted to ResNet-50, ViT-B/32, or even larger models, which may not be viable in these constrained settings. In contrast, the expanded dataset by our method can be readily employed to train a various range of model architectures (cf. Table 3), making it more applicable to scenarios with hardware limitations. One might suggest using CLIP in these situations by conducting knowledge distillation from large CLIP models to facilitate the training of smaller model architectures. However, as indicated in Table 1 and Table 18, although knowledge distillation of CLIP does enhance model performance on most datasets, the gains are limited. This arises from two key limitations of CLIP knowledge distillation. First, distillation can only yield marginal improvements when the performance of CLIP on the target dataset (*e.g.,* medical image domains) is not good. Second, distillation tends to be ineffective when there is a mismatch between the architectures of student and teacher models [9, 71]. This comparison further underscores the advantages of our method for training various network architectures, while the CLIP model architectures are fixed and not editable.

Table 18: Comparison between our methods and directly fine-tuning CLIP models on three medical image datasets. All results are averaged over three runs.

| Dataset | PathMNIST | BreastMNIST | OrganSMNIST |
|---|---|---|---|
| *Original* dataset | $72.4_{\pm 0.7}$ | $55.8_{\pm 1.3}$ | $76.3_{\pm 0.4}$ |
| CLIP linear probing | $74.3_{\pm 0.1}$ | $60.0_{\pm 2.9}$ | $64.9_{\pm 0.2}$ |
| CLIP fine-tuning | $78.4_{\pm 0.9}$ | $67.2_{\pm 2.4}$ | $78.9_{\pm 0.1}$ |
| CLIP knowledge distillation | $77.3_{\pm 1.7}$ | $60.2_{\pm 1.3}$ | $77.4_{\pm 0.8}$ |
| $5\times$-expanded by GIF-SD | $\mathbf{86.9}_{\pm 0.3}$ | $\mathbf{77.4}_{\pm 1.8}$ | $\mathbf{80.7}_{\pm 0.2}$ |

### F.4.2 Discussion on when to use GIF over zero-shot CLIP models

In Table 1, it is noted that while zero-shot CLIP performs well on datasets like Caltech 101 and Pets, it struggles with medical image datasets. This poses the question: when should we prefer GIF over pre-trained CLIP models? Although zero-shot CLIP outperforms our GIF-SD on the Caltech 101 and Pets datasets, our method demonstrates superior overall performance across six natural image datasets, as well as medical image datasets (see Table 1). Thus, *we recommend using our method as the primary option.*

Meanwhile, if the target dataset has a high distributional similarity with the CLIP training dataset, it may also be beneficial to consider CLIP as an alternative and see whether it can achieve better performance. Nevertheless, it is important to note that, as discussed in Section 5.1 and Appendix F.4.1, CLIP is less effective in some specific application scenarios. For instance, its performance on non-natural image domains like medical images is limited (as shown in Table 4). Additionally, publicly available CLIP checkpoints are restricted to larger models like ResNet-50 and ViT-B/32, making them unsuitable for scenarios with hardware constraints (*e.g.,* mobile terminals) where smaller model sizes are necessary. In these scenarios, our proposed method exhibits promising performance (as shown in Tables 3 and 4), offering a more versatile solution.

## F.5 More ablation studies

### F.5.1 The effectiveness of guidance in GIF-DALLE

GIF optimizes data latent features for informative sample creation by maximizing the designed objective functions of guidance (*i.e.,* class-maintained informativeness $\mathcal{S}_{inf}$ and sample diversity $\mathcal{S}_{div}$), which are essential for effective dataset expansion. With these essential guidance criteria, as shown in Table 19, our guided expansion framework obtains consistent performance gains compared to unguided expansion with SD, DALL-E2, or MAE, respectively. This verifies the effectiveness of our criteria in optimizing the informativeness and diversity of the created samples.

Table 19: Accuracy of ResNet-50 trained from scratch on small datasets and their expanded datasets by various methods. Here, CIFAR100-Subset is expanded by $5\times$, Pets is expanded by $30\times$, and all other natural image datasets are expanded by $20\times$. All medical image datasets are expanded by $5\times$. Moreover, MAE, DALL-E2 and SD (Stable Diffusion) are the baselines of directly using them to expand datasets without our GIF. All results are averaged over three runs.

| Dataset | Natural image datasets | | | | | | | Medical image datasets | | | |
|---|---|---|---|---|---|---|---|---|---|---|---|
| | Caltech101 | Cars | Flowers | DTD | CIFAR100-S | Pets | Average | PathMNIST | BreastMNIST | OrganSMNIST | Average |
| *Original* | 26.3 | 19.8 | 74.1 | 23.1 | 35.0 | 6.8 | 30.9 | 72.4 | 55.8 | 76.3 | 68.2 |
| *Expanded* by MAE | 50.6 | 25.9 | 76.3 | 27.6 | 44.3 | 39.9 | 44.1 (+13.2) | 81.7 | 63.4 | 78.6 | 74.6 (+6.4) |
| *Expanded* by GIF-MAE (ours) | 58.4 | 44.5 | 84.4 | 34.2 | 52.7 | 52.4 | 54.4 (+23.5) | 82.0 | 73.3 | 80.6 | 78.6 (+10.4) |
| *Expanded* by DALL-E2 | 61.3 | 48.3 | 84.1 | 34.5 | 52.1 | 61.7 | 57.0 (+26.1) | 82.8 | 70.8 | 79.3 | 77.6 (+9.4) |
| *Expanded* by GIF-DALLE (ours) | 63.0 | 53.1 | 88.2 | 39.5 | 54.5 | 66.4 | 60.8 (+29.9) | 84.4 | 76.6 | 80.5 | 80.5 (+12.3) |
| *Expanded* by SD | 51.1 | 51.7 | 78.8 | 33.2 | 52.9 | 57.9 | 54.3 (+23.4) | 85.1 | 73.8 | 78.9 | 79.3 (+11.1) |
| *Expanded* by GIF-SD (ours) | **65.1** | **75.7** | **88.3** | **43.4** | **61.1** | **73.4** | **67.8 (+36.9)** | **86.9** | **77.4** | **80.7** | **81.7 (+13.5)** |

In this appendix, we further explore the individual influence of these criteria on GIF-DALLE. Specifically, as mentioned in Appendix D.1, GIF-DALLE conducts guided imagination on *the CLIP embedding space*, which directly determines the content of the created samples. With the aforementioned essential criteria, as shown in Figure 2b, our GIF-DALLE is able to create motorbike images with more diverse angles of view and even a new driver compared to unguided DALLE expansion. Here, we further dig into how different criteria influence the expansion effectiveness of GIF-DALLE. As shown in Table 20, boosting the class-maintained informativeness $\mathcal{S}_{inf}$ is the foundation of effective expansion, since it makes sure that the created samples have correct labels and bring new information. Without it, only $\mathcal{S}_{div}$ cannot guarantee the created samples to be meaningful, although the sample diversity is improved, even leading to worse performance. In contrast, with $\mathcal{S}_{inf}$, diversity promotion $\mathcal{S}_{div}$ can further bring more diverse information to boost data informativeness and thus achieve better performance (cf. Table 20). Note that contrastive entropy increment $s\log(s) - s'\log(s')$ in class-maintained informativeness plays different roles from diversity promotion $\mathcal{S}_{div}$. Contrastive entropy increment promotes the informativeness of each generated image by increasing the prediction difficulty over the corresponding seed image, but this guidance cannot diversify different latent features obtained from the same image. By contrast, the guidance of diversity promotion encourages the diversity of various latent features of the same seed image, but it cannot increase the informativeness of generated samples regarding prediction difficulty. Therefore, using the two guidance together leads the generated images to be more informative and diversified, thus bringing higher performance improvement (cf. Table 20). As a result, as shown in Table 19, with these two essential criteria as guidance, the model accuracy by GIF-DALLE is 3.3% accuracy higher than unguided data generation with DALL-E2.

Table 20: Ablation of guidance in GIF-DALLE for expanding CIFAR100-Subset by $5\times$.

| Method | $\mathcal{S}_{inf}$ | $\mathcal{S}_{div}$ | CIFAR100-Subset |
|---|---|---|---|
| GIF-DALLE | | | $52.1_{\pm0.9}$ |
| | ✓ | | $53.1_{\pm0.3}$ |
| | | ✓ | $51.8_{\pm1.3}$ |
| | ✓ | ✓ | $54.5_{\pm1.1}$ |

### F.5.2 The effectiveness of guidance in GIF-SD

We next analyze GIF-SD. As mentioned in Appendix D.2, we conduct *channel-wise* noise perturbation for latent optimization in GIF-SD. As analyzed in Appendix B.2, the channel-level latent feature encodes more subtle style information, and conducting channel-level noise perturbation diversifies the style of images while maintaining its content integrity. Therefore, our guided optimization particularly diversifies the style of the created images, without changing the content semantics of the latent features after diffusion (cf. Figure 19). Moreover, the prompt-guided diffusion with our explored prompts helps to enrich image styles further (*e.g.,* cartoon or oil painting). Hence, combining both of them enables GIF-SD to create new samples with much higher diversity (cf. Figure 19).

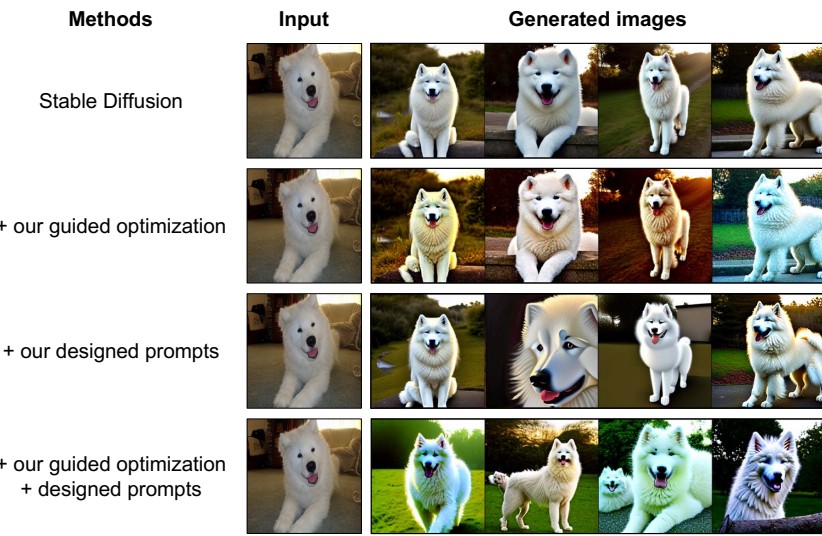

Figure 19: Visualization of the generated images by SD with our explored guided optimization and designed prompts.

We then investigate the individual influence of our guidance criteria on GIF-SD on the basis of our prompt-guided diffusion. As shown in Table 21, both the class-maintained informativeness guidance $\mathcal{S}_{inf}$ and the diversity promotion guidance $\mathcal{S}_{div}$ contribute to model performance. One interesting thing is that, unlike GIF-DALLE that does not work without $\mathcal{S}_{inf}$, GIF-SD can work well using only the diversity promotion guidance $\mathcal{S}_{div}$. The key reason is that GIF-SD conducts channel-level noise perturbation over latent features and particularly diversifies the style of the created images without changing the content semantics of the latent features after diffusion. Therefore, the class semantics can be maintained well when only promoting sample diversity. Moreover, combining both guidance criteria enables GIF-SD to achieve the best expansion effectiveness (cf. Table 21), leading to promising performance gains (*i.e.,* 13.5% accuracy improvement on average over six natural image datasets) compared to unguided expansion with SD (cf. Table 19).

Table 21: Ablation of guidance and prompts in GIF-SD for expanding CIFAR100-Subset by $5\times$.

| Method | Designed prompts | $\mathcal{S}_{inf}$ | $\mathcal{S}_{div}$ | CIFAR100-Subset |
|---|---|---|---|---|
| | | | | $52.9_{\pm 0.8}$ |
| | ✓ | | | $56.2_{\pm 1.0}$ |
| GIF-SD | ✓ | ✓ | | $59.6_{\pm 1.1}$ |
| | ✓ | | ✓ | $59.4_{\pm 1.2}$ |
| | ✓ | ✓ | ✓ | $61.1_{\pm 0.8}$ |

### F.5.3 Discussions on the constraint of the perturbed feature in GIF

The hyper-parameter $\varepsilon$ is used to ensure that the perturbed feature does not deviate from the input feature significantly, and its value depends on the prior model and target dataset. As described in Appendix D, for GIF-SD, we set $\varepsilon = 0.8$ for most natural image datasets and further adjust $\varepsilon = 4$ for Caltech101 to increase its dataset diversity for better performance. Once determined for a given prior model and target dataset, $\varepsilon$ remains fixed for various expansion ratios. As shown in the following table, there is no need to increase when the expansion ratio becomes larger.

Table 22: Ablation of hyper-parameter $\varepsilon$ on Caltech101 for GIF-SD.

| $\varepsilon$ on Caltech101 | 2 | 4 | 8 |
|---|---|---|---|
| 5×-expanded by GIF-SD | 53.0 | **54.4** | 53.6 |
| 10×-expanded by GIF-SD | 59.2 | **59.3** | 58.2 |

### F.6 Discussion of training models with only expanded images

It is interesting to know how the model performs when trained with only the created images by our method. To this end, we train ResNet-50 from scratch using only images generated by GIF-DALLE on the CIFAR100-Subset and compare the result with a model trained on the real images of the CIFAR100-Subset.

We report the results regarding $1\times$ expansion in Table 23. We find that the model trained with only $1\times$ synthetic images performs worse than the model trained with the original dataset, indicating that the quality of synthetic data still lags behind that of real images. Please note that this does not degrade our contribution, since our work aims to expand small datasets rather than replace them entirely. Moreover, mixing the original images with the created images to the same size as the original dataset can lead to better performance than using only the original dataset. This suggests that the created images are not a simple repetition of the original dataset but offer new information that is useful for model training. Lastly, the model trained on the complete 1x-expanded dataset significantly outperforms the models trained either only on the original dataset or solely on the generated images, underscoring the potential of synthetic images in expanding small-scale datasets for model training.

Table 23: Performance of the model trained with only the expanded data of the $5\times$-expanded CIFAR100-Subset dataset by GIF-DALLE.

| CIFAR100-Subset | Data amount | Accuracy |
|---|---|---|
| *Training with real images in original dataset* | 10,000 | $35.0_{\pm1.7}$ |
| *Training with only the $1\times$-created data* by GIF-DALLE | 10,000 | $21.0_{\pm0.7}$ |
| *Training with mixing original data and $1\times$-created data* by GIF-DALLE | 10,000 | $37.2_{\pm0.8}$ |
| *Training with $1\times$-expanded dataset* by GIF-DALLE | 20,000 | $45.6_{\pm1.1}$ |

We next report the results regarding $5\times$ expansion in Table 24. The model trained with $5\times$ synthetic images has already performed comparably to the model trained with real images. This result further verifies the effectiveness of our explored dataset expansion method. Moreover, the model trained with the full $5\times$-expanded dataset performs much better than that trained with only the original dataset or with only the generated images. This further shows that using synthetic images for model training is a promising direction. We expect that our innovative work on dataset expansion can inspire more studies to explore this direction in the future.

Table 24: Performance of the model trained with only the expanded data of the $5\times$-expanded CIFAR100-Subset dataset by GIF-DALLE.

| CIFAR100-Subset | Data amount | Accuracy |
|---|---|---|
| *Training with real images in original dataset* | 10,000 | $35.0_{\pm1.7}$ |
| *Training with only the $5\times$-created data* by GIF-DALLE | 50,000 | $35.2_{\pm1.3}$ |
| *Training with $5\times$-expanded dataset* by GIF-DALLE | 60,000 | $54.5_{\pm1.1}$ |

## F.7 Effectiveness on long-tailed classification dataset

In previous experiments, we have demonstrated the effectiveness of our proposed method on relatively balanced small-scale datasets. However, real-world classification datasets are usually class imbalanced and even follow a long-tailed class distribution. Therefore, we further apply GIF-SD to expand a long-tailed dataset, *i.e.,* CIFAR100-LT [5] (with the imbalance ratio of 100), to see whether it is also beneficial to long-tailed learning. Here, we train ResNet-50 from scratch with the cross-entropy loss or the balanced softmax loss [35] for 200 epochs, where Balanced Softmax [35, 102] is a class re-balancing loss designed for long-tailed learning.

As shown in Table 25, compared to training with cross-entropy directly on the original CIFAR100-LT dataset, $20\times$ expansion by our GIF-SD leads to a 13.5% model accuracy gain. This demonstrates the effectiveness of our proposed method in long-tailed learning. More encouragingly, our GIF expansion boosts the performance of few-shot classes more than many-shot classes, which means that GIF helps to address the issue of class imbalance.

Besides the cross-entropy loss, our dataset expansion is also beneficial to model training with long-tailed losses, such as Balanced Softmax. As shown in Table 25, $20 \times$ expansion by GIF-SD boosts the accuracy of the Balanced Softmax trained model by 14.8%, and significantly improves its tail-class performance by 26.8%. These results further demonstrate the applicability of our GIF to long-tailed learning applications. We expect that this work can inspire more long-tailed learning studies to explore dataset expansion since information lacking is an important challenge in long-tailed learning [103].

Table 25: Effectiveness of GIF-SD for expanding CIFAR100-LT (imbalance ratio 100) by $10\times$, where all models are trained for 200 epochs. Here, Balanced Softmax [35, 102] is a class re-balancing losses designed for long-tailed learning.

| CIFAR100-LT | Training losses | Many-shot classes | Medium-shot classes | Few-shot classes | Overall |
|---|---|---|---|---|---|
| *Original* | Cross-entropy | 70.5 | 41.1 | 8.1 | 41.4 |
| *20×-expanded* by GIF-SD | Cross-entropy | 79.5 (+9.0) | 54.9 (+13.8) | 26.4 (+18.3) | 54.9 (+13.5) |
| *Original* | Balanced Softmax | 67.9 | 45.8 | 17.7 | 45.1 |
| *20×-expanded* by GIF-SD | Balanced Softmax | 73.7 (+5.8) | 59.2 (+13.4) | 44.5 (+26.8) | 59.9 (+14.8) |

## F.8 Effectiveness on larger-scale dataset

In previous experiments, we have demonstrated the effectiveness of our proposed method on small-scale natural and medical image datasets. In addition to that, one may also wonder whether our method can be applied to larger-scale datasets. Although expanding larger-scale datasets is not the goal of this paper, we also explore our method to expand the full CIFAR100 by $5\times$ for model training. As shown in Table 26, compared to direct training on the original CIFAR100 dataset, our GIF-SD leads to a 9.4% accuracy gain and GIF-DALLE leads to an 8.7% accuracy gain. Such encouraging results verify the effectiveness of our methods on larger-scale datasets.

Table 26: Effectiveness of GIF for expanding the full CIFAR100.

| Dataset | CIFAR100 |
|---|---|
| *Original* | $70.9_{\pm 0.6}$ |
| *Expanded* | |
| *5×-expanded* by GIF-DALLE | $79.6_{\pm 0.3}$ |
| *5×-expanded* by GIF-SD | $\mathbf{80.3}_{\pm 0.3}$ |

## F.9 Safety check

Ethical considerations, especially in AI research and data generation, are indeed paramount. Our approach is constructed with care to avoid negative implications, as evidenced in the following points:

- **Controlled generation**: In our approach, the generation of synthetic data is driven by our expansion guidances, which ensure that new data is derived directly and meaningfully from the original dataset. This controlled mechanism minimizes the risks of creating unrelated or potentially harmful images.

- **No personal or sensitive data**: It is also worth noting that our method primarily focuses on publicly available datasets like CIFAR, Stanford Cars, and similar, which do not contain personal or sensitive information. As such, the risks related to privacy breaches or misrepresentations are substantially diminished.

Following this, we further employ the Google Cloud Vision API[4] to perform a safety check on the 50,000 images generated during 5x-expansion of CIFAR100-S by GIF-SD. The Google Cloud Vision API is a tool from Google that uses deep learning to analyze and categorize content in images, commonly used for safety checks. It evaluates the likelihood of the image containing **adult** themes such as nudity or sexual activities, alterations made for humor or offensiveness (**spoof**), **medical** relevance, **violent** content, and **racy** elements which could include suggestive clothing or poses. This assessment aids in ensuring that images adhere to content standards and are appropriate for their target audiences.

As evidenced by Table 27, the synthetic images by our method are safe and harmless. To be specific, the majority of our generated images are categorized as either "Very unlikely" or "Unlikely" across all five metrics. Moreover, for categories like "Adult" and "Medical", the likelihood is almost negligible. Moreover, the visualized images in Appendix G also highlight the benign nature of the images produced by our method.

Table 27: Safety check of the generated images of CIFAR100-S by our GIF-SD, in terms of different metrics of Google Cloud Vision API.

| Metrics | Very unlikely | Unlikely | Neutral | Likely | Very likely |
| --- | --- | --- | --- | --- | --- |
| Adult | 96% | 4% | 0% | 0% | 0% |
| Spoof | 82% | 15% | 3% | 0% | 0% |
| Medical | 86% | 14% | 0% | 0% | 0% |
| Violence | 69% | 31% | 0% | 0% | 0% |
| Racy | 66% | 25% | 9% | 0% | 0% |

---

[4]https://cloud.google.com/vision/docs/detecting-safe-search

# G    More Visualization Results

This appendix provides more visualized results for the created samples by our methods on various natural image datasets. Specifically, we report the synthetic images by GIF-SD on Caltech101 in Figure 20, those by GIF-DALLE in Figure 21 and those by GIF-MAE in Figure 22. The visualized results show that our GIF-SD and GIF-DALLE can create semantic-consistent yet content-diversified images well, while GIF-MAE can generate content-consistent yet highly style-diversified images. The visualization of GIF-SD and GIF-DALLE on other natural image datasets are shown in Figures 23-32.

## G.1    Visualization of the expanded images on Caltech101

### G.1.1    Visualization of the expanded images by GIF-SD on Caltech101

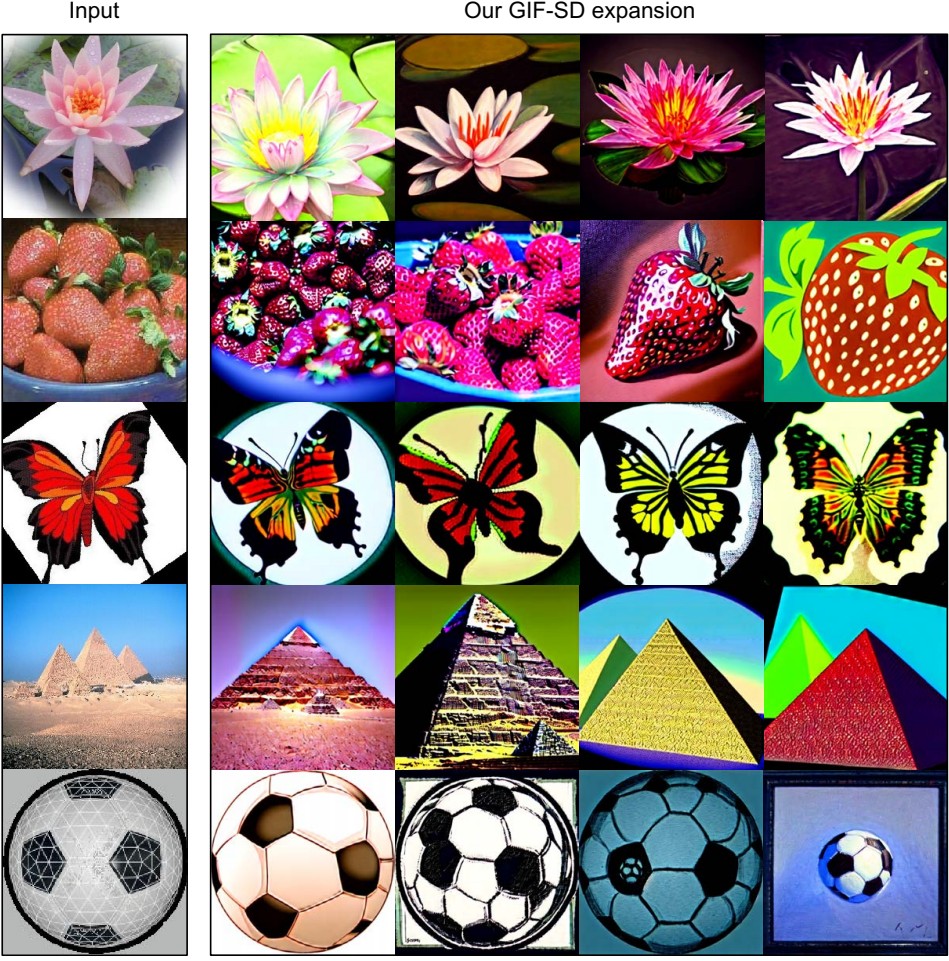

Figure 20: Visualization of the created samples on Caltech101 by GIF-SD.

### G.1.2 Visualization of the expanded images by GIF-DALLE on Caltech101

Input            Our GIF-DALLE expansion

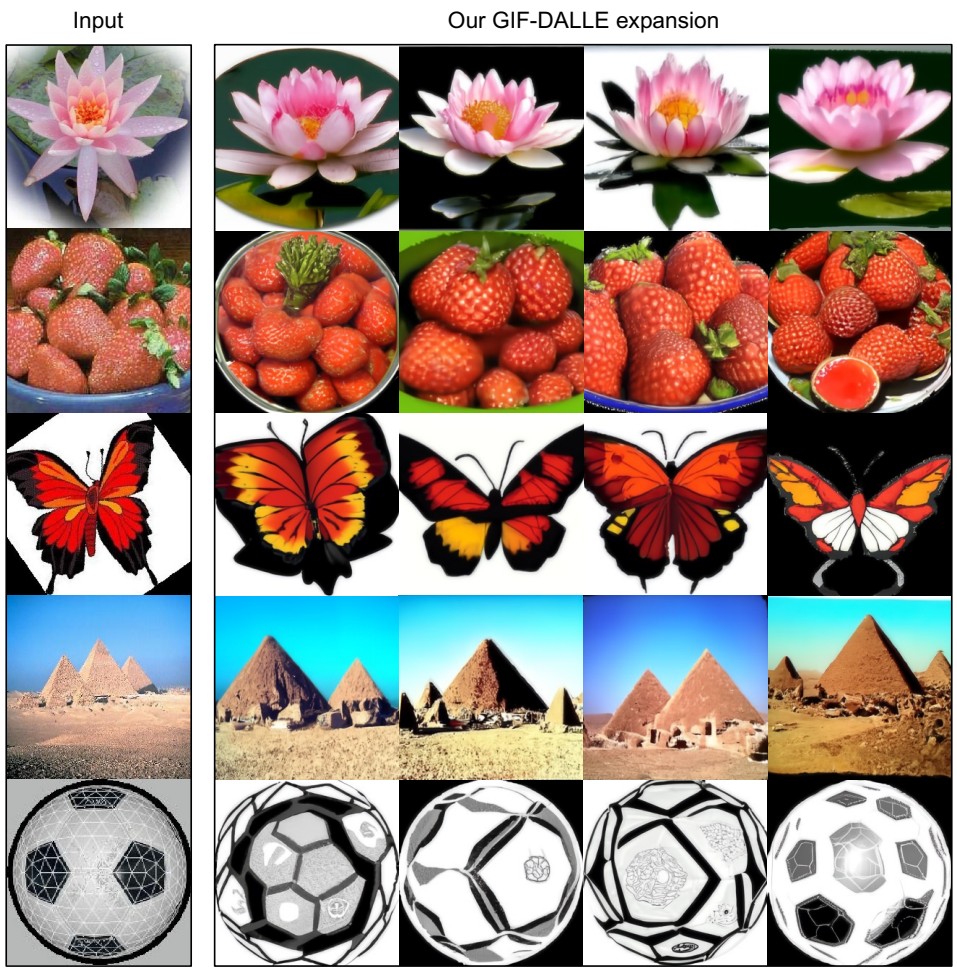

Figure 21: Visualization of the created samples on Caltech101 by GIF-DALLE.

### G.1.3 Visualization of the expanded images by GIF-MAE on Caltech101

Input             Our GIF-MAE expansion

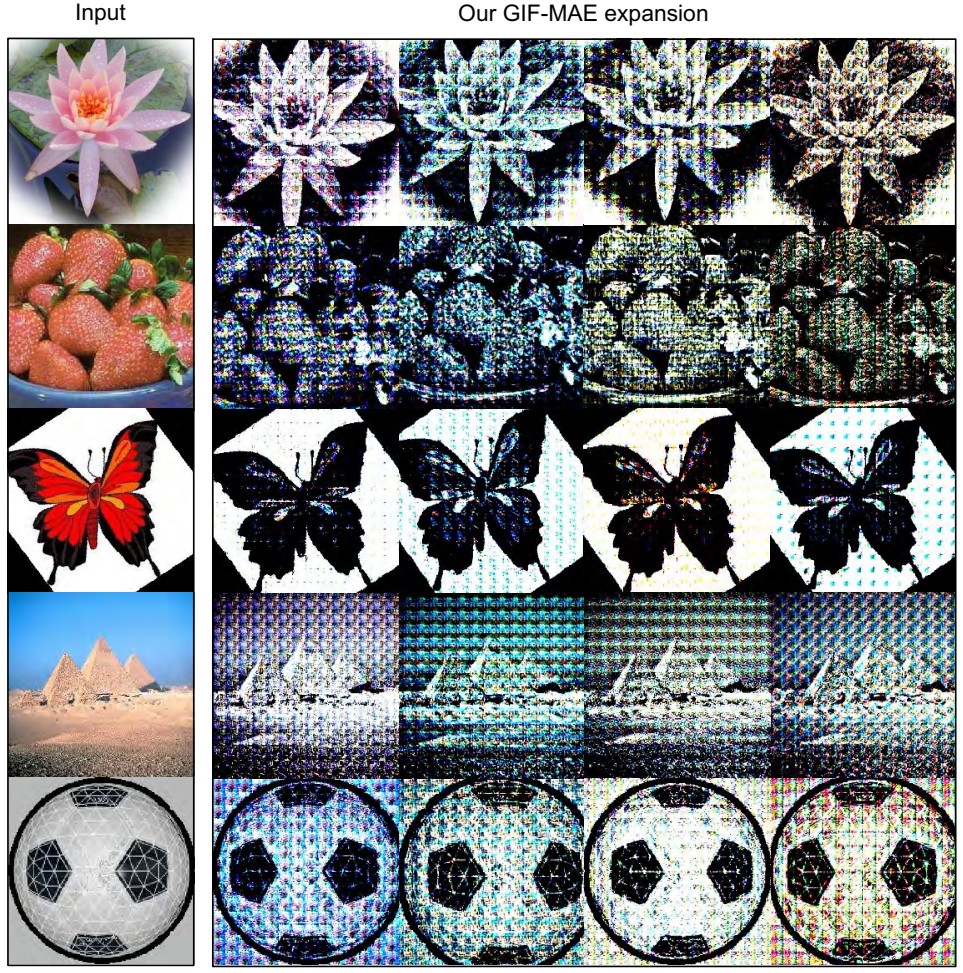

Figure 22: Visualization of the created samples on Caltech101 by GIF-MAE.

## G.2 Visualization of the expanded images on Cars

### G.2.1 Visualization of the expanded images by GIF-SD on Cars

Input                                    Our GIF-SD expansion

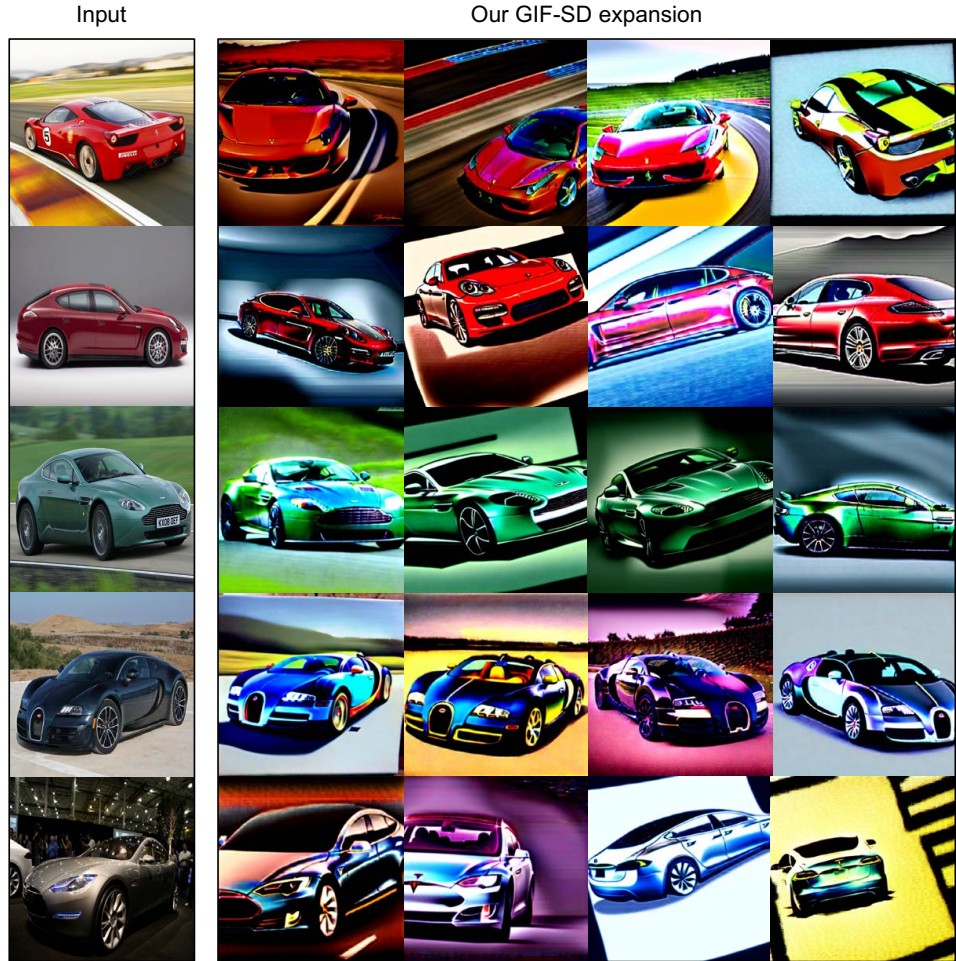

Figure 23: More visualization of the synthetic samples on Cars by GIF-SD.

## G.2.2 Visualization of the expanded images by GIF-DALLE on Cars

Input          Our GIF-DALLE expansion

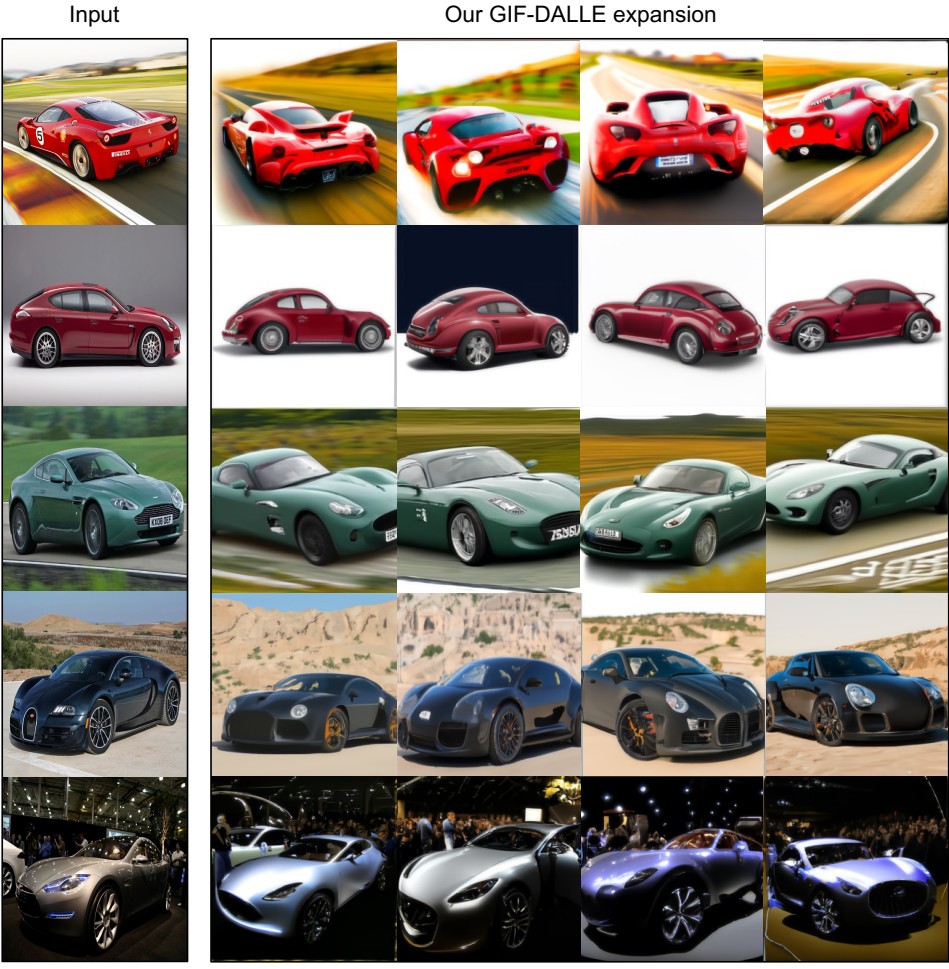

Figure 24: More visualization of the synthetic samples on Cars by GIF-DALLE.

## G.3 Visualization of the expanded images on Flowers

### G.3.1 Visualization of the expanded images by GIF-SD on Flowers

Input                           Our GIF-SD expansion

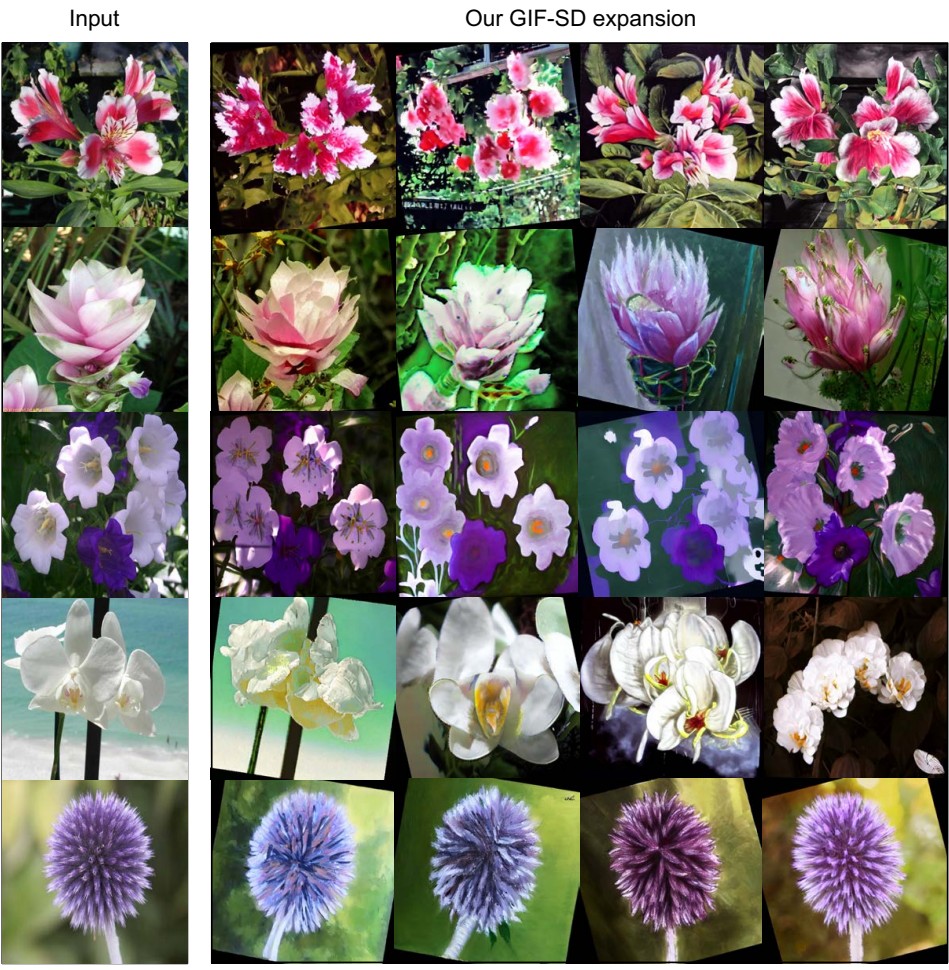

Figure 25: More visualization of the synthetic samples on Flowers by GIF-SD.

## G.3.2 Visualization of the expanded images by GIF-DALLE on Flowers

Input                                  Our GIF-DALLE expansion

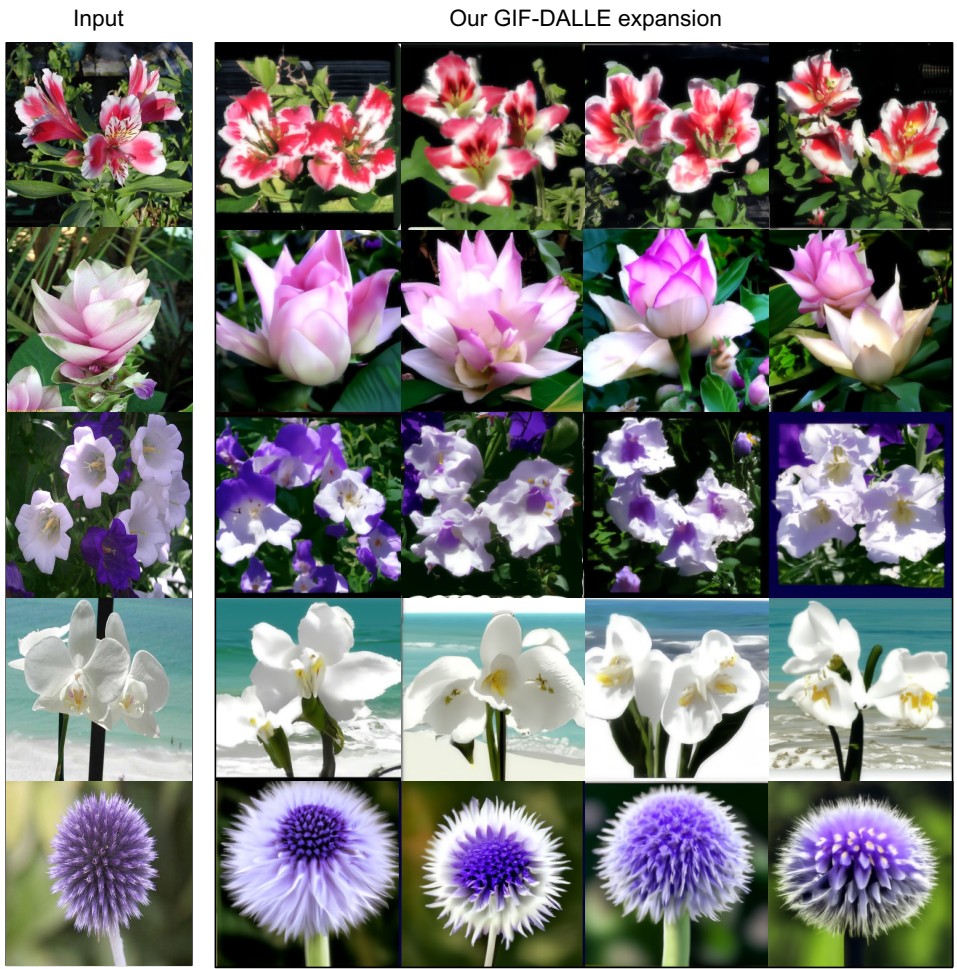

Figure 26: More visualization of the synthetic samples on Flowers by GIF-DALLE.

## G.4 Visualization of the expanded images on Pets

### G.4.1 Visualization of the expanded images by GIF-SD on Pets

Input                                    Our GIF-SD expansion

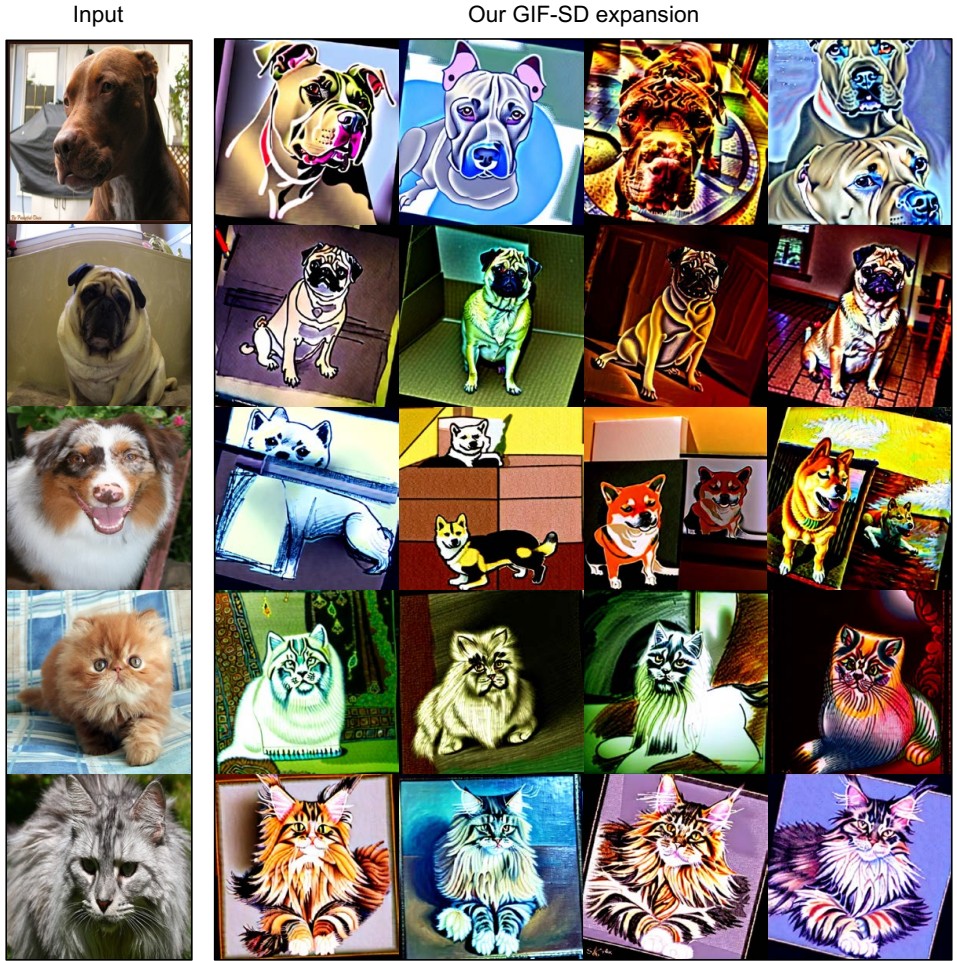

Figure 27: More visualization of the synthetic samples on Pets by GIF-SD.

## G.4.2 Visualization of the expanded images by GIF-DALLE on Pets

Input         Our GIF-DALLE expansion

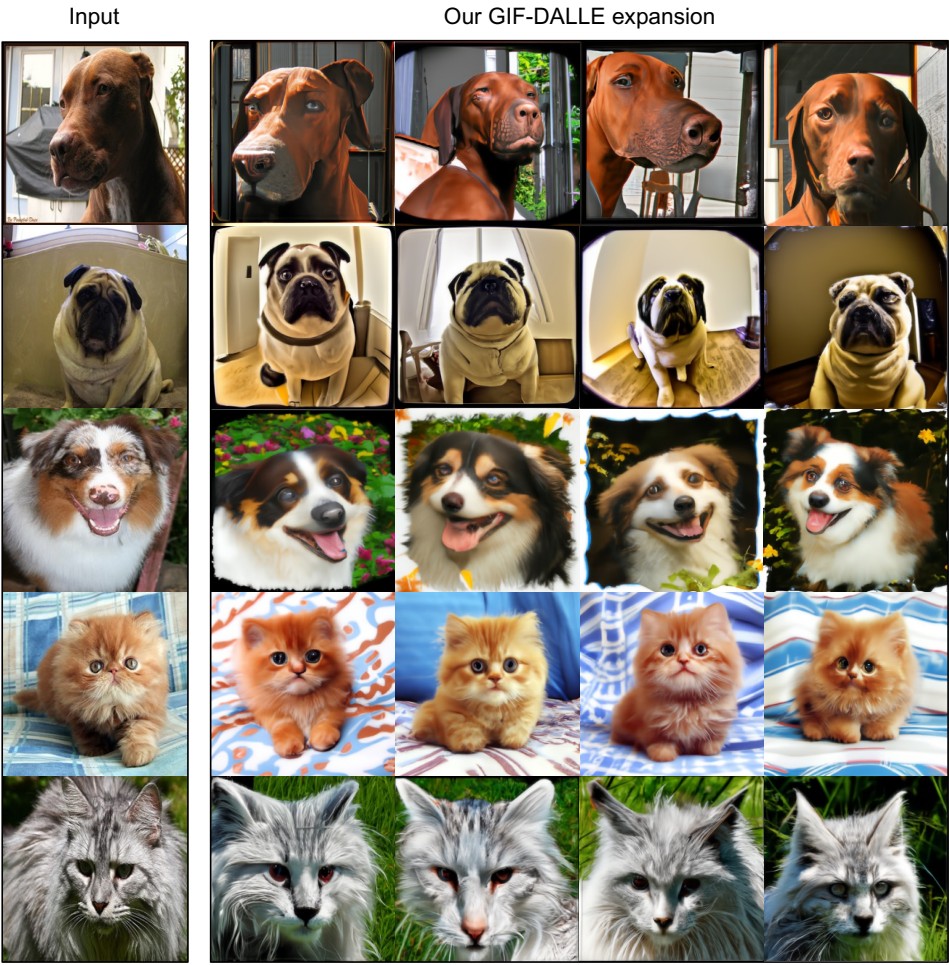

Figure 28: More visualization of the synthetic samples on Pets by GIF-DALLE.

## G.5 Visualization of the expanded images on CIFAR100-Subset

### G.5.1 Visualization of the expanded images by GIF-SD on CIFAR100-Subset

Input                                          Our GIF-SD expansion

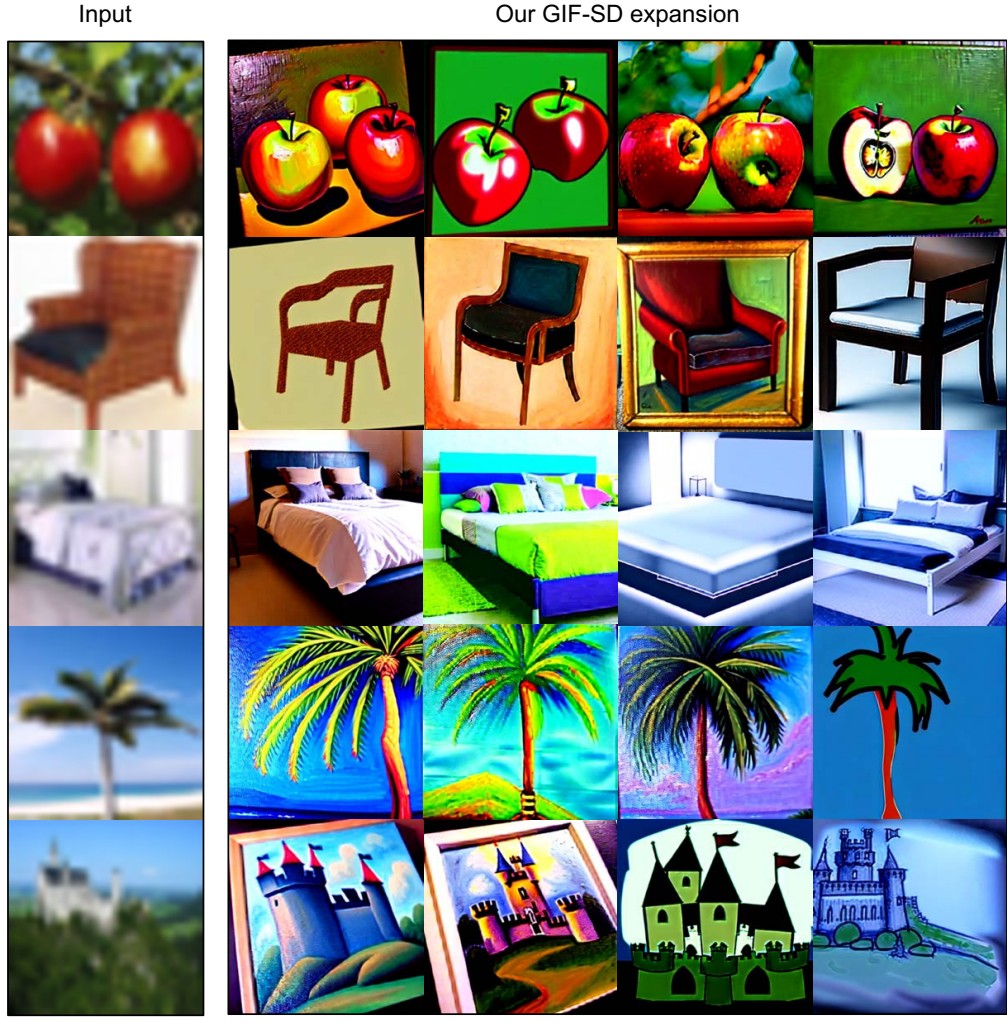

Figure 29: More visualization of the synthetic samples on CIFAR100-Subset by GIF-SD.

### G.5.2 Visualization of the expanded images by GIF-DALLE on CIFAR100-Subset

Input                                            Our GIF-DALLE expansion

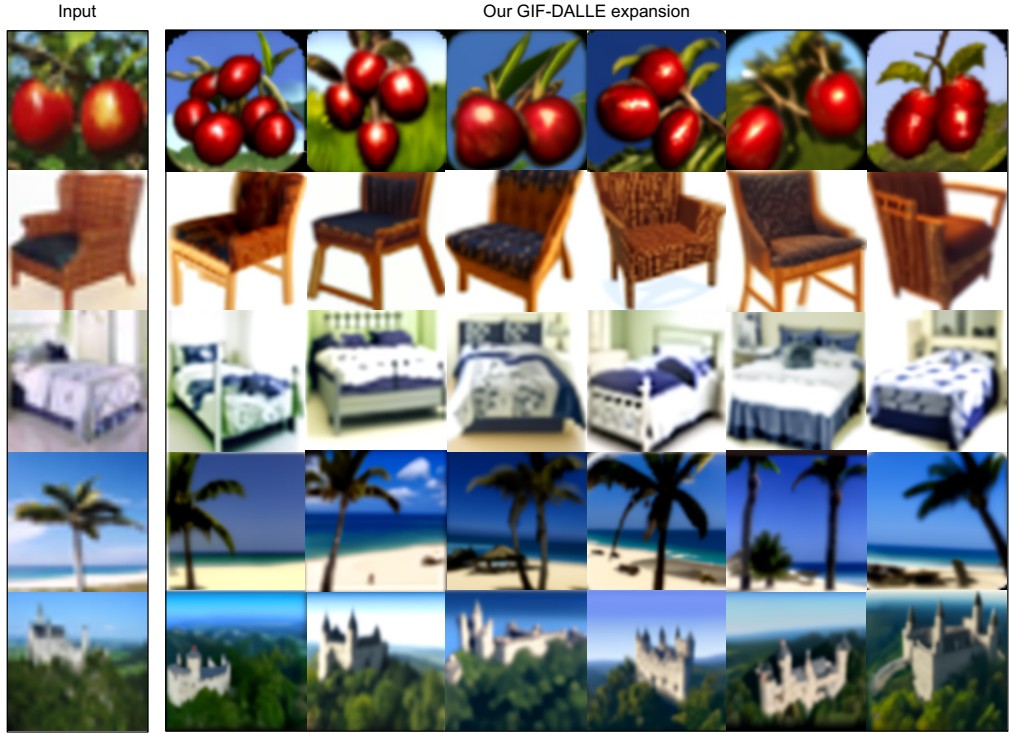

Figure 30: More visualization of the synthetic samples on CIFAR100-Subset by GIF-DALLE. Note that the resolution of the input CIFAR100 images is small (*i.e.,* 32×32), so their visualization is a little unclear.

## G.6 Visualization of the expanded images on DTD

### G.6.1 Visualization of the expanded images by GIF-SD on DTD

Input                                    Our GIF-SD expansion

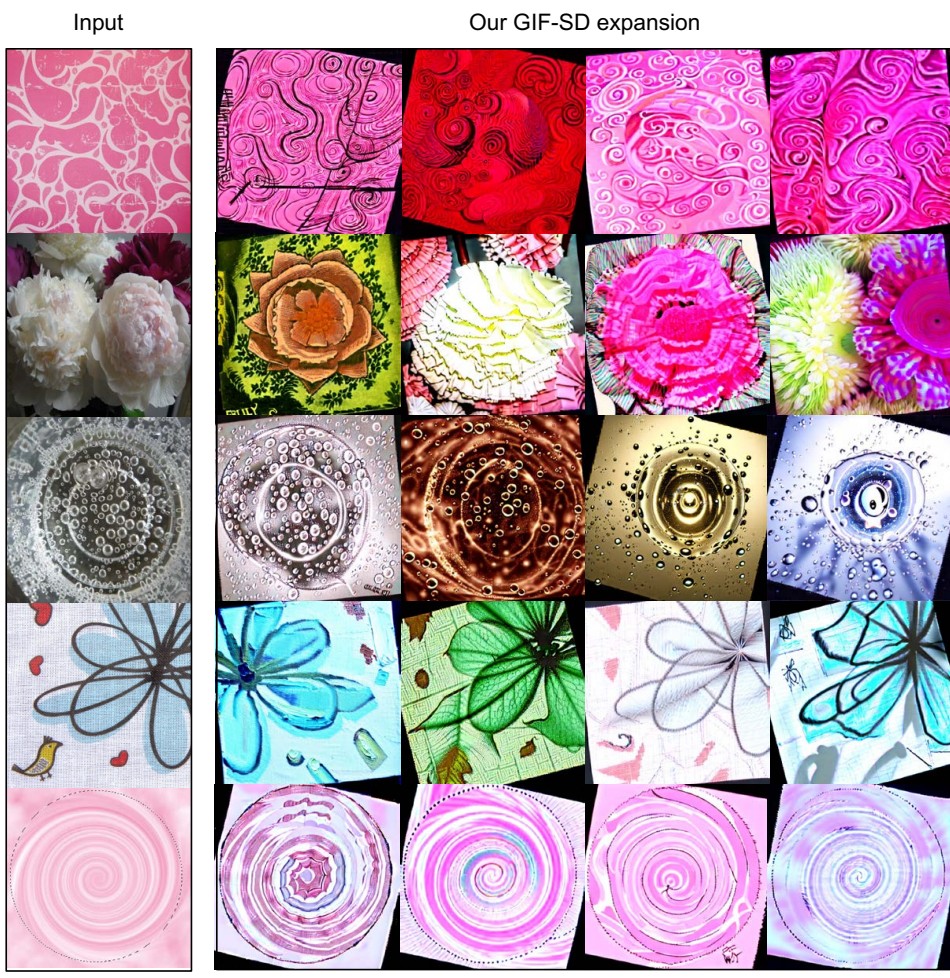

Figure 31: More visualization of the synthetic samples on DTD by GIF-SD.

## G.6.2 Visualization of the expanded images by GIF-DALLE on DTD

Input                    Our GIF-DALLE expansion

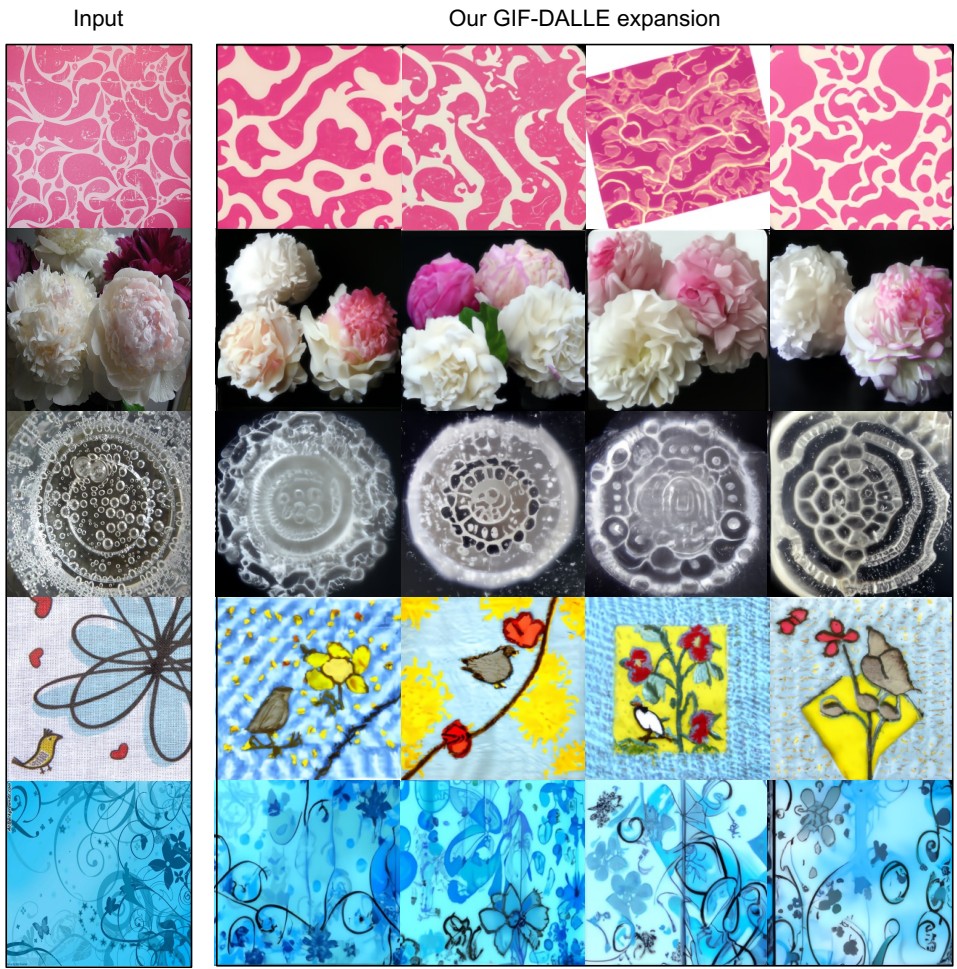

Figure 32: More visualization of the synthetic samples on DTD by GIF-DALLE.

