# Expanding Small-Scale Datasets with Guided Imagination

**Yifan Zhang**[1*]   **Daquan Zhou**[2*]   **Bryan Hooi**[1]   **Kai Wang**[1]   **Jiashi Feng**[2]
[1]National University of Singapore     [2]ByteDance

We organize the supplementary materials as follows:

- Appendix A: More related studies.

- Appendix B: More preliminary studies.

- Appendix C: Theoretical analysis.

- Appendix D: More method and implementation details.

- Appendix E: Dataset statistics.

- Appendix F: More experimental results and discussions.

- Appendix G: More visualization results.