# OpenReview forum: "Expanding Small-Scale Datasets with Guided Imagination"
_NeurIPS.cc/2023/Conference — NeurIPS 2023 poster_

### Official Review · Reviewer_6GPi · 2023-06-27

**Soundness:** 3 good
**Presentation:** 3 good
**Contribution:** 3 good
**Rating:** 7
**Confidence:** 4

**Summary:**

The paper proposes an image generation framework for expanding small-scale datasets. The proposed Guided Imagination Framework (GIF) leverages large language-vision models (i.e., CLIP) and generative models (i.e., DALL-E2, Stable Diffusion, and MAE) based on two criteria that help to generate informative new images with (i) class-consistent semantics but (ii) higher content diversity. Experimental analyses and ablation studies on several natural and medical image datasets show that GIF is effective and efficient in boosting the accuracy and generalization of the models trained on artificially expanded datasets.

**Strengths:**

### Significance
Combining large language-and-vision models with generative models such as stable diffusion to generate synthetic images has become a popular topic that can support various downstream tasks. To this end, the paper has the potential to attract wide attention.

### Originality
The proposed guided imagination framework for dataset expansion combines CLIP with several SOTA generative models in a unique way to optimize the variation over the latent space so that they can generate images with more diverse content but still within the same semantic concept/category.

### Clarity
The two core criteria of class-maintained informativeness boosting and sample diversity promotion are defined and illustrated with clear examples. The paper is written well in general although there is just too much material that could not be fit into the main paper and hence moved to supplementary material (but this leaves holes in the paper).

### Quality
The experimental evaluations and ablation studies are quite extensive and informative. Main findings are summarized adequately in the main text and detailed discussion is presented in the supplementary material due to page constraints. The contribution of the proposed framework is supported with quantitative evidence from various perspectives.


**Weaknesses:**

### Experiments
Some of the important technical and algorithmic details are not included in the paper, which leaves the reader a bit hanging in the air and pushes them to read the supplementary material (which becomes rather like a "mandatory material"). Not sure if there is an easy way to fix this but it would be good to re-organize the material to remedy this concern.

How are the baseline DALL-E2, SD, and MAE implemented? Are they configured the same way as the proposed methods but without the actual GIF components (i.e., ablated)?

While fine-tuning CLIP, how much effort was put into to its proper optimization, for example, was something similar to robust fine-tuning of CLIP as presented in [A] followed?

Also it would be interesting to see how much more is there to be gained if GIF is applied to large-scale datasets?

### Literature Review & Citations
Literature review is too brief. It would be good to squeeze in a bit more of the other relevant text-driven dataset generation papers (e.g., [33, 44, 48, 69, B-D]). At the end of the day, GIF-SD is also driven by text prompts. To this end, some of the redundancy across sections (e.g., Sections 1, 3, and 4) with repeated statements can be eliminated.

The references contain papers that are not cited in the main paper. Actually, only 43 (see the list below) of the 98 papers are referenced in the main paper and the others are probably mentioned (didn't fully check) in the supplementary material. The bibliography must include only the references that are cited in the main paper. Supplementary material can have its own references.

Citations that appear in the main paper: 2, 6, 8, 9, 10, 11, 12, 14, 15, 20, 22, 23, 24, 29, 30, 35, 36, 37, 45, 46, 47, 49, 50, 52, 55, 59, 60, 61, 62, 63, 66, 68, 72, 73, 74, 76, 77, 81, 82, 86, 89, 93, 97

There are some additional recent work that might be good to include in the paper or in the supplementary [B-E].

### Minor Concerns
76: Better to include references to Cutout, GridMask, RandAugment, Cars and DTD here because they are mentioned for the first time in the paper.

208: s' should be f'

328: conducts <== conduct

329: There is no Figure 10(e) in the main paper.

### Suggested References
[A] Wortsman, Mitchell, et al. "Robust fine-tuning of zero-shot models." Proceedings of the IEEE/CVF Conference on Computer Vision and Pattern Recognition. 2022.

[B] Sarıyıldız, Mert Bülent, et al. "Fake it till you make it: Learning transferable representations from synthetic ImageNet clones." Proceedings of the IEEE/CVF Conference on Computer Vision and Pattern Recognition. 2023.

[C] Zhou, Yongchao, Hshmat Sahak, and Jimmy Ba. "Training on Thin Air: Improve Image Classification with Generated Data." arXiv preprint arXiv:2305.15316 (2023).

[D] Tian, Yonglong, et al. "StableRep: Synthetic Images from Text-to-Image Models Make Strong Visual Representation Learners." arXiv preprint arXiv:2306.00984 (2023).

[E] Azizi, Shekoofeh, et al. "Synthetic data from diffusion models improves imagenet classification." arXiv preprint arXiv:2304.08466 (2023).


**Questions:**

- How are the prior generative models (DALL-E2, SD, and MAE) trained as baselines? Are they configured exactly the same way as the proposed methods but simply without the GIF components?
- While fine-tuning CLIP, how much effort was put into to its proper optimization, for example, was something similar to robust fine-tuning of CLIP as presented in [A] followed?
- How much more is there to be gained if GIF is applied to large-scale datasets? Not asking for more experiments here but looking for some intuition/speculation/expert opinion on this.
- Would it be possible to expand the literature review with a few more relevant papers on text-driven dataset generation?
- Would it be possible to reorganize the paper to include some of the technical details about the method/algorithm in the main paper?
- References must be updated to include only those cited in the main paper.

**Limitations:**

There is no discussion on limitations and broader impact of the work. Error analysis is a bit superficial. A more detailed error analysis would be helpful. Is there a risk of mode collapse? Is there a risk of exacerbating biases while generating images? Cutting down the required human effort and cost of data collection can be highlighted as broader impact.

---

> ### Author Rebuttal · Authors · 2023-08-09
>
> Thanks a lot for the highly constructive comments. We are glad to see that the task significance and the method originality are appreciated. We answer all questions point by point as follows.
>
> ---
>
>
> >**Q1. Would it be possible to put more technical details in the main paper?**
>
> We highly appreciate the constructive suggestion on the organization of technical details. Due to page constraints, some method details were placed in the supplementary material (cf. Appendix D). However, per the NeurIPS guidelines (https://nips.cc/Conferences/2023/CallForPapers), we are allowed an additional page in the camera-ready version if accepted. Given this opportunity, we plan to incorporate more essential method details into the main text to enhance its clarity and accessibility. We hope this adjustment can address the concern.
>
> ---
>
> >**Q2. Are the baselines (DALL-E2, SD, and MAE) configured the same way as the proposed methods but without the actual GIF components?**
>
>
> Yes, these baselines were configured identically to their GIF counterparts for ablated evaluation, with the only exception being the absence of the guided imagination optimization (cf. Section 4). Without the crucial GIF components, these baselines cannot ensure that the synthesized data bring sufficient new information, content diversity, and accurate labels for small-scale dataset expansion. This is evident in Table 1 (cf. Ssction 5.1), where their performance is markedly inferior compared to our GIF methods.
>
> ---
>
> >**Q3. The implementation of CLIP fine-tuning**
>
> In this work, we did not use any sophisticated fine-tuning strategies like the robust fine-tuning of CLIP as presented in [A]. Instead, we kept it straightforward and used cross-entropy for standard fine-tuning. It is worth noting that this cross-entropy training is consistent with other training-from-scratch baselines of dataset expansion methods. By doing so, we ensure a fair comparison across different methods and baselines.
>
> ---
>
> >**Q4. Can GIF be applied to large-scale datasets?**
>
> Thanks for raising this question. In fact, we have discussed this in Appendix F.8, where our GIF method is adaptable to expanding larger-scale datasets, e.g., the original CIFAR dataset. The results in Table 23 (cf. Appendix F.8) further demonstrate the effectiveness of our approach.
>
> ---
>
> >**Q5. Literature review is too brief. Please review more text-driven dataset generation methods and the studies [B-E]**
>
> Thank you for the suggestions. Due to page limitations, we put a detailed review of related work in Appendix A. Should our work be accepted, the extra page granted for the camera-ready will allow us to expand the literature review in the main text. We will then incorporate a more comprehensive discussion on text-driven dataset generation papers and include the related studies [B-E] mentioned by the reviewer.
>
>
> ---
>
> >**Q6. References must be updated to include only those cited in the main paper**
>
> Thanks for pointing this out. We will revise the reference in the main paper to include only the ones cited, and ensure the supplementary has its own reference list.
>
> ---
>
> >**Q7. Discussion on limitations and broader impact**
>
> In light of the suggestion, we will enrich our supplementary with the following discussions:
>
> **Limitation**:
>
> - **(1) Performance of generated samples**: The expanded samples are still less informative than real samples. For example, a ResNet-50 trained from scratch on our 5x-expanded CIFAR100-Subset achieves an accuracy of 61.1%, which lags behind the 71.0% accuracy on the original CIFAR100. This gap signals the potential for advancing algorithmic dataset expansion. We expect that this pioneering work can inspire more studies to explore dataset expansion so that it can even outperform a human-collected dataset of the same size.
>
> - **(2) Quality of generated samples**: Some samples might have noise, as exemplified in Figure 5b. Despite seeming less realistic, those samples are created following our guidance (e.g., class-maintained informativeness boosting). This ensures the class consistency of these samples, mitigating potential negative effects on model training. Nonetheless, refining the expansion method to address these noisy cases can further enhance the effectiveness of dataset expansion.
>
> - **(3) Scope of work**. Our current focus is predominantly on image classification. Exploring the adaptability of our method to other tasks, such as object detection, is an intriguing next step.
>
> **Broader impact**. Our method can offer a notable reduction in the time and cost associated with manual data collection and annotation for dataset expansion, as discussed in Section 5.2 (cf. Lines 343-350). This can revolutionize how small datasets are expanded, making deep learning more accessible to scenarios with limited data availability (cf. Table 1).
>
> ---
>
> >**Q8. Other minor issues**
>
> Thanks for noting these details. We appreciate the meticulous review. We will make the necessary corrections following the suggestions.
>
> ---
>
> We deeply appreciate the insightful feedback from the reviewer. Thanks to these constructive suggestions, our paper's quality has been greatly elevated to meet the conference's high standards. We kindly ask the reviewer to take into account the refinements and enhancements we have incorporated. The reviewer's endorsement is influential and could potentially influence other reviewers in recognizing our paper's merits.

---

> > ### Comment · Reviewer_6GPi · 2023-08-14
> > **Response to the rebuttal**
> >
> > Thanks for the detailed rebuttal! I have noted your responses to my concerns and do not have any further queries.

---

> > > ### Author Response · Authors · 2023-08-16
> > >
> > > Thanks again for your insightful and meticulous review. We genuinely appreciate your continued engagement with our work. With the invaluable feedback, our manuscript has seen significant improvement. We hope that your understanding of our method's value might convey the significance of our contributions more clearly to the entire review panel. Once again, thank the reviewer for the dedication to enhancing the quality of our paper.

---

### Official Review · Reviewer_J4xT · 2023-06-28

**Soundness:** 3 good
**Presentation:** 2 fair
**Contribution:** 3 good
**Rating:** 5
**Confidence:** 5

**Summary:**

This paper introduces a novel task that addresses the expansion of a limited dataset into a larger one by utilizing a Guided Imagination Framework. The proposed approach harnesses latent features to generate new data instances while placing emphasis on preserving class-specific information and enhancing sample diversity. By employing these criteria, the generated data effectively facilitates the improvement of the model's learning process. The efficacy of the proposed method is demonstrated through rigorous validation on multiple datasets, showcasing its superior performance.

**Strengths:**

- This paper presents a promising solution to the challenge of generating reliable supplementary training data, distinguishing itself from conventional data augmentation methods by generating unique content distinct from the existing dataset.
- The paper introduces Guided Imagination Framework that leverages prior knowledge of latent features to generate additional data while maintaining alignment with class labels, ensuring consistency throughout the generated samples.
- Experimental results show the remarkable effectiveness of this proposed approach in significantly enhancing the accuracy of image classification tasks, substantiating its superiority over alternative methods.

**Weaknesses:**

- A major concern arises from the results presented in Table 1, particularly regarding the Stanford Cars dataset (https://ai.stanford.edu/~jkrause/papers/fgvc13.pdf). The comparison reveals that certain alternative methods achieve superior results (>90) without the need for additional data, as demonstrated in (https://arxiv.org/pdf/2102.05918.pdf). This raises skepticism regarding the efficacy of the augmented data in genuinely improving the model's performance, as the reported results fall below the mentioned benchmark (<80).

- Another point of consideration is the authors' exclusive focus on employing ResNet-50 as the sole backbone architecture for their experiments. This singular choice could introduce bias, as it remains plausible that only ResNet performs well within the proposed framework. To establish the robustness and generalizability of the proposed method, it would be valuable for the authors to explore and validate its performance across multiple alternative backbone architectures, thereby mitigating potential biases.

- It is important to acknowledge that the proposed method significantly increases the training time due to the inclusion of a generative model and the utilization of substantially larger amounts of data (at least 5 times more) for training the classification model. This extensive time requirement may pose considerable challenges, particularly in classification tasks where efficiency is a crucial aspect to consider and optimize.

**Questions:**

- It would be valuable to receive further insights into the time required to complete the generating-training procedure. Understanding the time constraints associated with the proposed method is essential for assessing its practicality and scalability. Insights into the time requirements would enable a more comprehensive evaluation of the proposed approach in terms of efficiency and resource allocation.
- A comprehensive discussion on the training process and underlying structures of the listed methods, including GIF-MAE and GIF-SD, would greatly enhance the readers' understanding of these approaches. Detailed explanations of the methodologies employed, such as the specific techniques utilized for data generation and the architectures of the models, would provide valuable insights into the novelty and effectiveness of these methods. Expanding on these aspects would facilitate a more thorough comparative analysis of the proposed approach against other techniques.
- Gaining clarity on whether pretrained weights were utilized in all of the proposed methods would be beneficial for assessing the fairness of the comparison. Understanding the extent to which prior knowledge is leveraged across different approaches is crucial for interpreting and contextualizing the results. Specifically, differentiating between methods that start from scratch without any pretrained weights and those that employ pretrained models, such as CLIP, would provide a more accurate understanding of the experimental setup and potential biases.
- To further validate the proposed approach, it would be beneficial to compare it with other methods that also generate synthesized data, as referenced in the provided papers (https://arxiv.org/pdf/2301.06043.pdf, https://ieeexplore.ieee.org/document/8629301, https://proceedings.mlr.press/v156/bao21a/bao21a.pdf). Conducting such comparisons would offer valuable insights into the relative strengths and weaknesses of the proposed method and provide a broader perspective on its performance compared to existing state-of-the-art techniques.

**Limitations:**

- The generation method employed to expand dataset volumes is a time-consuming process that necessitates various preprocessing steps. Furthermore, training with a substantially larger amount of data (at least 5 times more) adds to the complexity. This undertaking should be approached with careful consideration, acknowledging the resource and time requirements involved.

- In order to provide a comprehensive evaluation, it would be advantageous for the authors to conduct fair comparisons with existing models and methods, taking into account factors such as memory footprint. Considering these aspects would enable a more thorough assessment of the proposed approach in relation to its counterparts, providing a clearer understanding of its advantages and limitations.

- Given that the proposed method involves the generation of new data, it is highly recommended that an ethical review be conducted to ensure compliance with ethical guidelines and considerations. The potential impact and implications of generating synthetic data should be carefully evaluated to ensure that it aligns with ethical principles, privacy regulations, and legal requirements. Such a review would demonstrate a responsible approach towards the development and application of the proposed method.

---

> ### Author Rebuttal · Authors · 2023-08-09
>
> Thanks for the comprehensive and constructive comments, particularly for recognizing that our solution is promising and remarkably effective. We address all the concerns as follows.
>
> ---
>
> >**Q1. Concern on the results in Table 1, particularly regarding Cars**
>
> The experimental results are indeed reasonable based on the following reasons:
>
> - **Different training methodology**: Our focus is to expand small-scale datasets, and as such, **all the models are trained from scratch for evaluation**. This allows us to fairly compare the effectiveness of various expansion methods, without biases from pre-trained models. Please note that our task is orthogonal and complementary to the model transfer task in Align (https://arxiv.org/pdf/2102.05918.pdf), where the transfer performance inevitably surpasses training from scratch due to its extensive pre-training on large datasets.
>
> - **Additional fine-tuning results**: To address this concern, we further fine-tune an ImageNet pre-trained ResNet50 on Cars. Using the original dataset, the fine-tuned ResNet-50 achieves a performance of 87.6. Remarkably, with the expanded dataset by GIF-SD, the performance further increases to 88.9, highlighting the effectiveness of our method in enhancing model fine-tuning. The additional results further substantiate the reasonableness of this paper.
>
> ---
>
> >**Q2. Concern on the exclusive focus on using ResNet-50**
>
> Thanks for the comment, but it seems that there might have been a misunderstanding. Our study did indeed consider the robustness and generalizability of our method across various model architectures, not solely focusing on ResNet-50. Specifically, Table 3 (cf. Section 5.1) and Table 13 (cf. Appendix F.3) demonstrate our method's performance on various architectures, such as ResNext, WideResNet, and MobileNet. These results highlight an important advantage of our method: the expanded dataset can be used with various network structures without needing to regenerate, thereby affirming its broad applicability.
>
> ---
>
> >**Q3. Question about the time and costs**
>
> Thanks for the comment. Please refer to [General Response G1](https://openreview.net/forum?id=82HeVCqsfh&noteId=e7B6qvrwYr) for the detailed response.
>
> ---
>
> >**Q4. Details on GIF-MAE and GIF-SD**
>
> Thanks for the feedback. Due to the page limitations, we had to put the method details, implementation details, and pseudo-code of all proposed methods in Appendix D. We understand that the supplementary is not mandatory reading, but it is intended to provide a comprehensive understanding of what the reviewer suggests. This concern might be addressed by delving into this appendix. If the reviewer has any further questions after referring to these details, we would be more than happy to provide additional clarification.
>
> Furthermore, according to the NeurIPS guidelines, we are permitted to add one more page to the camera-ready. In such a case, we will move more crucial method details to the main text to enhance its clarity and comprehension. We hope these measures can sufficiently address the concern.
>
> ---
>
> >**Q5. Clarification on whether pre-trained weights were utilized during model training**
>
> During the model training phase, the models are trained from scratch for all dataset expansion methods (**including ours**) in all tables (including Table 4), ensuring no pre-trained weights are used for maintaining a fair comparison. The only exceptions are **CLIP-related baselines** (in Tables 1,4, and 15), where the pre-trained checkpoint of CLIP was used. We believe this clarification should offer a more accurate understanding of our experimental setup.
>
> ---
>
> >**Q6. Further comparisons to advanced generative methods**
>
> Thanks for suggesting a comparison with the mentioned generative methods. We recognize its importance, but there are two primary challenges in executing this comparison: (1) The mentioned GAN-based generative methods have not made their model checkpoints publicly available. (2) Training GANs from scratch, particularly when data is limited, frequently results in non-convergence or yields unmeaningful outcomes, making it unsuitable for small dataset expansion.
>
> Meanwhile, recent research [A] has indicated that diffusion models, thanks to iterative denoising, outperform GANs in image generation. As such, we opted for diffusion models in our work, as their checkpoints are readily available.
>
> Even so, we agree with the value of comparisons with more advanced generative methods. Therefore, we further compare our method with a recent ICLR work [B], as recommended by Reviewer 1CUd. Specifically, the method [B] proposes strategies like language enhancement (LE) and CLIP filters (CF) to enhance generative models for generating training data. For a fair comparison, we use this method with SD and our GIF-SD to expand the CIFAR100-S dataset by 5x. The results, as shown in the following table, further affirm the superiority of our method.
>
> | CIFAR100-S | Accuracy |
> | ---------- |:---------:|
> | Original | 35.0 |
> | 5x-expanded by SD | 52.9 (+17.9)|
> | 5x-expanded by SD + method [B] | 56.0 (+21.0)|
> | 5x-expanded by GIF-SD (ours) | 61.1 (+26.1)|
>
> [A] Diffusion models beat gans on image synthesis. In NeurIPS, 2021
> [B] Is synthetic data from generative models ready for image recognition? In ICLR, 2023
>
> ---
>
> >**Q7. Question on potential influences of synthetic data**
>
> We appreciate the question on the ethical implications. Please refer to [General Response G2](https://openreview.net/forum?id=82HeVCqsfh&noteId=e7B6qvrwYr) for the detailed response.
>
> ---
>
> Thanks for the valuable feedback. Through our rebuttal, we have clarified ambiguities, provided additional results, and deepened our discussions to address all concerns. We believe our revised work aligns more closely with the conference's standards. A positive reconsideration of the initial assessment would be greatly beneficial to our efforts. Should any further questions, we are glad to answer them.

---

> > ### Comment · Reviewer_J4xT · 2023-08-14
> > **Response to Authors' rebuttal**
> >
> > Dear Authors:
> >
> > Thank you for your responses to the questions I raised. Some of the response addressed my concerns. Below I'd like to summarize my thoughts:
> >
> > 1. The novelty of the work is still limited, even though the authors have made clear explanations. Reviewer 7xaH has raised a similar idea by citing numerous papers dealing with the same dataset expansion.
> > 2. Regarding the cost of the proposed method. It is shown by the authors that the proposed method is more effective than manual annotation for sure. However, the authors have not included the time cost for other models for comparison. The review 7xaH has also mentioned that a sufficient comparison should be made with related works.
> > 3. The authors mentioned that they trained all the models from scratch for fair comparison without biases from pre-trained models. However they also mentioned CLIP-related baselines are using pretrained weights. There’s conflict within these two claims and thus would lead to biased comparison. Reviewer 7xaH also mentioned that additional experiments with pre-trained models are necessary.
> > 4. The authors mentioned in general response that controlled mechanism minimizes the risks of creating unrelated or potentially harmful images. However this is questionable since no evidence nor experimental assessment was made in this aspect. The reviewer UbXT has raised similar concern that the quality and accuracy of the generated samples with perturbed features was not evaluated.
> >
> > In summation, while the authors have addressed some concerns raised by reviewers, I will not have access to the revised document prior to the final decision. Based on my assessment, I believe the manuscript requires substantial modifications before it is suitable for publication. Given these considerations, I maintain my initial score.

---

> > > ### Author Response · Authors · 2023-08-16
> > > **Follow-up Response (1/3)**
> > >
> > > We greatly appreciate the time the reviewer has dedicated to reviewing our response and providing further comments. Below, we address the outstanding concerns.
> > >
> > > ---
> > >
> > > >**Q1. Concern on the novelty of the work**
> > >
> > > Thanks for the feedback. We totally understand the reviewer's concern, if we merely focus on the idea of using generative models for training data synthesis. However, the novelty of this work does not rest on the freshness of this idea, but is deeply rooted in the task importance and the method novelty.
> > >
> > > - **Task significance**: Automatic dataset expansion, especially in small-data scenarios, holds immense importance. While the concept might not be brand-new, our work provides a unique perspective on formally defining and tackling this task. The proposed dataset expansion significantly reduces human efforts and expenses associated with manual data collection, and markedly improves the model performance in small-data scenarios (cf. Table 1 in Section 5.1). These benefits are highly important for real-world applications since manual data collection is highly expensive in small-data scenarios  (e.g., medical image domains). The importance of this task has been highly recognized by Reviewer 1CUd "*the task is meaningful*" and Reviewer UbXT "*the task could contribute to the development of academia and industry*".
> > >
> > > - **Method novelty**: Although using generative models to create training data is not a new idea, our method introduces a distinct design. The key of our innovation lies in the concept of guided imagination (cf. Section 3.1), coupled with two critical expansion criteria (cf. Section 3.2). These insights are underpinned by both empirical observations and theoretical analysis (cf. Section 3, Section 5.2, Appendix B, Theorem 4.1). Based on these criteria, our approach can guide generative models to create informative new samples with novel content and correct class labels for expanding datasets. In contrast, while the methods mentioned by the reviewer and Reviewer 7xaH also introduce new training data, they cannot ensure the synthesized data bring sufficient new information and accurate labels for the target small datasets. Moreover, training GANs from scratch, especially with very limited data, often fails to converge or produce meaningful results [A,B], making the mentioned GAN-based methods less effective in small-data scenarios. As such, our method emerges as a more effective way to expand small datasets. The contribution of our method has been recognized by Reviewer 1CUd "*the proposed framework is intuitive, and the criteria are well-motivated*". Please note that **after reading our rebuttal, Reviewer 7xaH also concurs with the novelty of our method.**
> > >
> > > We hope this clarification can further illustrate the unique contributions of our work. To make it clearer, we will further clarify our main contributions at the end of Section 1, and add one more paragraph in Section 2 to highlight the differences between our work and the related studies mentioned by reviewers.
> > >
> > > [A] Towards Principled Methods for Training Generative Adversarial Networks. In ICLR, 2017
> > >
> > > [B] Training Generative Adversarial Networks with Limited Data. In NeurIPS, 2020
> > >
> > > ---
> > >
> > > >**Q2. The authors have not included the time cost for other models for comparison**
> > >
> > > Thanks for the constructive suggestions. We're pleased to delve deeper into our comparative analysis on the time cost of data expansion and performance gains between different methods.
> > >
> > > As shown in the table below, our GIF offers a more favorable trade-off compared to other methods. Specifically, GIF-MAE has a time cost within the same magnitude as data augmentation, but it delivers much better performance gains. The slight time overhead introduced by MAE is offset by GPU acceleration, resulting in competitive time costs. This further verifies the superiority of our method. For those prioritizing performance, GIF-SD becomes a more attractive option. Although it involves a longer time due to its iterative diffusion process, it provides more significant performance gains.
> > >
> > >
> > > | Methods | Expansion speed (per image) | Time (10,000 images) | Accuracy gains over natural image datasets|
> > > | -- |:--:| :--:| :--:|
> > > | Cutout | 0.008s | 76s | +12.8 |
> > > | GridMask | 0.007s  | 72s | +14.4 |
> > > | RandAugment | 0.008s | 82s | +20.5 |
> > > | GIF-MAE | 0.008s  | 80s | +23.5 |
> > > | GIF-SD | 6.6s   | 2h (8 GPUs) | +36.9 |
> > >
> > > Upon dissecting the time costs and performance gains, our observations can be summarized as:
> > >
> > > - **Time costs**: Diffusion-based expansion (e.g., GIF-SD) > MAE-based expansion (e.g., GIF-MAE, while GANs have similar time cost)   ≈ Augmentation-based expansion
> > > - **Performance gains**: Diffusion-based expansion > MAE-based  expansion > Augmentation-based expansion
> > >
> > > We hope this comparison further clarifies the merit of our approach. Following the constructive comment, we will enrich the cost analysis in Lines 343-350 with this discussion.

---

> > > > ### Author Response · Authors · 2023-08-16
> > > > **Follow-up Response (2/3)**
> > > >
> > > > >**Q3. Clarification on the training setups of dataset expansion methods and CLIP-related baselines**
> > > >
> > > > Thanks for pointing out the need for clarity, and we apologize for the confusion arising from our previous explanation. For a clearer understanding, let's look at our training setup table-by-table.
> > > >
> > > > - **Tables 1,4, and 14**: All dataset expansion methods (including Cutout, GridMask, RandAugment, MAE, DALL-E2, SD and our GIF methods) are trained from scratch, which ensures **a fair comparison among these dataset expansion methods** (cf. Line 274-280). In contrast, the CLIP baselines (e.g., zero-shot CLIP, CLIP Distillation, CLIP tuning) were based on CLIP checkpoints but **they did not use dataset expansion**. This distinction is crucial as it highlights the inherent differences between CLIP-based baselines and dataset expansion methods. The inclusion of CLIP-related baselines is intentional, since it enables us to demonstrate the advantages of dataset expansion over CLIP transfer in small-data applications (cf. Lines 296-308 and Appendix F.4.1).
> > > > - **Tables 2-3, 5-6, 9-13, 16-23**: All methods are trained from scratch to ensure a fair comparison.
> > > > - **Table 15**: This table aims to evaluate the effectiveness of our expanded datasets in model fine-tuning. Therefore, all the baselines in this table use pre-trained checkpoints of CLIP, including ours. Note that, except for this table, all the results of our methods are obtained through training from scratch.
> > > >
> > > > In sum, these CLIP-related baselines in Tables 1,4&14 are used for different goals, so they would not lead to biased comparisons among dataset expansion methods. We hope our detailed breakdown of the training setups addresses the concern adequately.
> > > >
> > > > ---
> > > >
> > > > >**Q4. Additional experiments with pre-trained models are necessary**
> > > >
> > > > We appreciate the suggestion of providing more comparisons in the setting of model fine-tuning. In the following table, we further report the results of expanding CIFAR100-S for fine-tuning CLIP models. Notably, our GIF-SD not only achieves a remarkable +4.2% improvement over the original dataset, but also outperforms other dataset expansion methods (e.g., RandAugment and the method [C] suggested by Reviewer 1CUd).  This result further verifies the superiority of our approach in the fine-tuning setting, which was also recognized by Reviewer 7xaH after reading our rebuttal. Thanks again for the constructive suggestion, following which we will incorporate this result and discussion into Section 5.1.
> > > >
> > > > | CIFAR100-S | CLIP fine-tuning |
> > > > | -- |:--:|
> > > > | Original dataset | 75.2 |
> > > > | 5x-expanded by RandAugment   | 77.7 (+2.5) |
> > > > | 5x-expanded by SD+the method [C]   | 77.0 (+1.8) |
> > > > | 5x-expanded by GIF-SD (ours)  | 79.4 (+4.2) |
> > > >
> > > >
> > > > [C] Is synthetic data from generative models ready for image recognition? In ICLR, 2023.
> > > >
> > > > ---
> > > >
> > > > >**Q5. No experimental assessment was made for the safety of the synthetic images**
> > > >
> > > > We highly appreciate the valuable suggestion for the safety check of the synthetic images. Following this, we further employ the [Google Cloud Vision API](https://cloud.google.com/vision/docs/detecting-safe-search) to perform a safety check on the 50,000 images generated during 5x-expansion of CIFAR100-S by GIF-SD. The Google Cloud Vision API is a tool from Google that uses deep learning to analyze and categorize content in images, commonly used for safety checks.
> > > >
> > > > As evidenced by the table below, the synthetic images by our method are safe and harmless. To be specific, the majority of our generated images are categorized as either "Very unlikely" or "Unlikely" across all five metrics. Moreover, for categories like "Adult" and "Medical", the likelihood is almost negligible. We also encourage the reviewer to refer to the visualized images in Appendix G, which further highlight the benign nature of the images produced by our method.
> > > >
> > > >
> > > > | Metrics | Very unlikely | Unlikely |  Neutral |  Likely |  Very likely |
> > > > | -- |:--:|:--:|:--:|:--:|:--:|
> > > > | Adult | 96% |   4%| 0%| 0%| 0%|
> > > > | Spoof | 82% |  15%| 3%| 0%| 0%|
> > > > | Medical |86% |  14%| 0%| 0%| 0%|
> > > > | Violence | 69% |  31%| 0%| 0%| 0%|
> > > > | Racy |66% |  25%| 9%| 0%| 0%|
> > > >
> > > >
> > > > Once again, we highly appreciate the reviewer's valuable suggestion.  We will incorporate this experimental discussion and evidence into Appendix F. By adopting reputable tools like the Google Cloud Vision API, we ensure our method that is not just technically sound, but also socially responsible, producing content free from potential harm or inappropriateness.

---

> > > > > ### Author Response · Authors · 2023-08-16
> > > > > **Follow-up Response (3/3)**
> > > > >
> > > > > >**Q6. Clarification on the modification in the main text**
> > > > >
> > > > > Thanks for the comment. We understand the reviewer's concern since we are not allowed to upload the revised paper at the moment. However, it is important to note that our key contributions – including our primary findings, novel methods, theoretical analysis, and empirical insights – have been comprehensively delineated in the main text. In this context, the necessary modifications are minimal compared to the initial main text. Specifically, in light of the reviewers' suggestions, we have made the following enhancements to the main text.
> > > > >
> > > > > - A succinct clarification of our key contributions at the end of Section 1.  [7xaH,J4xT]
> > > > > - Discussions of the mentioned related studies in Section 2.  [7xaH,J4xT,6GPi]
> > > > > - Move the pseudo-code of GIF-DALLE, GIF-SD, and GIF-MAE in Appendix D to Section 4 to further clarify their differences and details. [J4xT,6GPi]
> > > > > - Comparison results in the fine-tuning setting in Section 5.1. [7xaH,1CUd,J4xT]
> > > > > - Comparison results in terms of time cost and performance gain among various methods in Section 5.2 (cf. Lines 343-350). [7xaH,J4xT]
> > > > > - Update the reference section. [6GPi]
> > > > > - Due to the page limitation, we move the analysis of "Pixel-wise vs. channel-wise noise" in Section 5.2 (cf. Lines 323-330 and Figure 6) to Appendix B.
> > > > >
> > > > > Given these added analyses and empirical results, we believe that our contributions have been further enhanced without substantial revisions of the main text.
> > > > >
> > > > > In addition, we wish to highlight that modifications to the supplementary material do not encroach upon the main text's space. We have incorporated additional supportive content in the supplementary:
> > > > >
> > > > > - More detailed discussion on the time and cost of dataset expansion in Appendix D.5. [7xaH,J4xT]
> > > > > - Discussion on limitations and broader impact in Appendix D.6. [1CUd,6GPi]
> > > > > - Safety check of the synthetic images with empirical assessments in Appendix F. [J4xT]
> > > > > - Out-of-distribution results of CLIP model fine-tuning on CIFAR100-C in Appendix F. [7xaH]
> > > > > - Relation analysis between the domain gap (in terms of FID) and model performance in Appendix F. [1CUd]
> > > > > - Comparison results with a more advanced generative method in Appendix F. [7xaH,1CUd,J4xT]
> > > > >
> > > > > We hope these clarifications can further underscore the value of our work and address the concern adequately.
> > > > >
> > > > > ---
> > > > >
> > > > > Thanks again for the continuous engagement and invaluable suggestions, which have significantly enhanced the quality of our work. We hope that with these improvements, the reviewer can re-evaluate our submission in a new light. Should there be any further concerns, we stand ready to address them.

---

> > > > > > ### Comment · Reviewer_J4xT · 2023-08-21
> > > > > > **Response to the rebuttal**
> > > > > >
> > > > > > Thank you for the clarifications. It addresses most of the concerns. With respect to your answers, I propose to increase by one my score to 5.

---

> > > > > > > ### Author Response · Authors · 2023-08-21
> > > > > > >
> > > > > > > Thank the reviewer very much for the updated score and positive feedback. We deeply appreciate the constructive comments, which have significantly improved the quality of our paper. We firmly believe that, by incorporating the valuable suggestions and the enhanced experimental results the reviewer pointed out, our work will make meaningful contributions to the community.

---

### Official Review · Reviewer_UbXT · 2023-07-23

**Soundness:** 3 good
**Presentation:** 2 fair
**Contribution:** 2 fair
**Rating:** 4
**Confidence:** 2

**Summary:**

This paper proposes to expand a small dataset by automatically creating new labeled samples with pre-trained generative models, such as DALL-E2 and Stable Diffusion. The proposed framework, namely Guided Imagination Framework, contains two key parts, i.e., class-maintained information boosting and sample diversity promotion. The experiments of GIF-SD obtain improvements on multiple datasets.

**Strengths:**

1.	The key idea is interesting and dataset expansion task could contribute to the development of academia and industry.
2.	Higher accuracy can be achieved over multiple image datasets.


**Weaknesses:**

The proposed solution to dataset expansion is too easy. The class-maintaining informativeness strategy aims to generate perturbed features with seed sample class consistency. However, how to evaluate the quality and accuracy of generated samples with perturbed features?

**Questions:**

Please see the weaknesses

**Limitations:**

Please see the weaknesses

---

> ### Author Rebuttal · Authors · 2023-08-09
>
> Thanks for the effort in reviewing our paper. We are glad that the interesting idea and the important task are appreciated. We address the concerns point by point as follows.
>
> ---
>
> >**Q1. The proposed solution to dataset expansion is too easy**
>
> Thanks for the feedback on the simplicity of our solution. We believe that simplicity, when paired with effectiveness, is a strength, not a limitation.
>
> - **Simplicity as a virtue**: Simple solutions are generally more intuitive to understand, so they are usually more adaptable and more likely to be used by the community. Although our method might seem straightforward, achieving such simplicity with effectiveness requires substantial underlying effort and insights, as depicted in Section 3, Section 5.2, Appendix B, Appendix F and Theorem 4.1. These empirical observations and theoretical foundations not only justify our approach but also provide invaluable insights for future research.
>
> - **Significance of automatic dataset expansion**: Our straightforward method holds significant value. It showcases notable advancements in various small-data scenarios, such as in-distribution generalization (Table 1), out-of-distribution robustness (Table 2), and long-tailed problems (Table 22 in Appendix F.7). More importantly, it can drastically reduce the cost and time associated with human data collection/annotation for expanding small datasets (cf. the discussion in Section 5.2, Lines 343-350).
>
> In summary, while our method might appear straightforward at first glance, the depth of thought, experiments, and insights behind it attest to its uniqueness and efficacy. Hence, our method offers an important contribution to addressing the small-data challenges. Its value was also recognized by Reviewer 7xaH "*The proposed methodology is simple and easy to utilize*", Reviewer 1CUd "*the proposed framework is intuitive, and the criteria are well-motivated*", and Reviewer J4xT "This paper presents a promising solution".
>
> ---
>
> >**Q2. The class-maintaining informativeness strategy aims to generate perturbed features with seed sample class consistency. However, how to evaluate the quality and accuracy of generated samples with perturbed features?**
>
> The goal of our method is to create informative new samples that, when used in conjunction with the original data, improve the model performance in small-data scenarios. Directly assessing the perturbed features might not provide an accurate reflection of how beneficial the generated samples are when it comes to training a model. Therefore, a more effective way to judge the quality of the generated samples is to directly observe their enhancement in the final model performance. In light of this, this work evaluated various dataset expansion methods by directly measuring the performance of models trained on their expanded datasets.
>
> In addition, we would like to highlight that the class-maintaining property of our method is pivotal. This consistency is reinforced by the objective function outlined in Lines 207-208, ensuring that perturbed features remain aligned with the seed sample class. By guaranteeing this class consistency, our method can bring novel information to model training without the need to concern class misalignment of the generated samples. The ablation results of GIF-DALLE (cf. Table 17 in Appendix F.5.1) further demonstrate the effectiveness and importance of our class-maintaining strategy. To clarify, we provide the related result below (please refer to Appendix F.5.1 for more detailed analyses of the result).
>
> | Methods | Class-maintained strategy | Diversity promotion strategy| Accuracy of 5x-expanded CIFAR100-S |
> | ---------- |:---------------:|:---------------:|:---------------:|
> | GIF-DALLE | ✖ | ✖ | 52.1 |
> | GIF-DALLE | ✔ | ✖ | 53.1 |
> | GIF-DALLE | ✖ | ✔ | 51.8 |
> | GIF-DALLE | ✔ | ✔ | 54.5 |
>
> ---
>
> We genuinely appreciate the effort the reviewer has dedicated to reviewing our work. We have diligently addressed the concerns in the rebuttal, emphasizing the novel contributions and the potential value of our work. We respectfully request the reviewer to reconsider our paper with these clarifications in mind. A positive re-evaluation would be immensely beneficial to our efforts. We remain dedicated to addressing any further questions.

---

### Official Review · Reviewer_1CUd · 2023-07-25

**Soundness:** 3 good
**Presentation:** 3 good
**Contribution:** 3 good
**Rating:** 6
**Confidence:** 4

**Summary:**

This paper describes a new task called dataset expansion, which aims at expanding the size of small datasets to boost the performance of data-driven AI models on tasks like object classification. The paper proposes a framework called Guided Imagination Framework (GIF) to achieve it by utilizing pre-trained large-scale generative models to synthesize new informative samples according to the images in the dataset. Specifically, the method perturbs the latent feature of the examplar image in the dataset and designs two criteria, i.e., *class-maintained information boosting* and *sample diversity promotion*, to optimize the noise added to the latent feature of the exemplar image. The paper conducts extensive experiments and verifies the effectiveness of the proposed method in boosting the classification performance on small datasets of natural images and medical images.

**Strengths:**

This paper is well-organized and may be one of the first research to explore the effectiveness of synthetic images in boosting classification performance. The task is meaningful, the proposed framework is intuitive, and the criteria to maintain the class information and to encourage diversity are well-motivated. The experiments are systematically conducted and show that the proposed framework helps improve performance significantly on various backbones. The ablation study in the appendix shows the effectiveness of the proposed two criteria.

**Weaknesses:**

1. Though the authors give two reasons for not using CLIP for classifying the target dataset: (1) the transferability: Line#298-300; (2) GIF can benefit various model architectures. However, as one of the state-of-the-art backbones for classification, the proposed method's effectiveness in boosting the performance of CLIP on natural images needs to be justified.
2.  I think the comparison with the few-shot setting in [22] is required to prove the superiority of the method, as the few-shot setting in [22] shares similarities with the paper's task.

**Questions:**

1. I have some questions about GIF-SD. Does GIF-SD first sample several Gaussian noises $z$ and then perform the inverse diffusion process with the text prompt condition to obtain latent features $f$ which are perturbed by Eq.1 to obtain $f^{'}$ and then fed into the image decoder? In that case, since $f^{'}$ is not the direct perturbation of the feature of the real image sample in the dataset, I think GIF-SD is quite different from GIF-DALLE and GIF-MAE and may fail to guarantee the resemblance to the image samples in the dataset. Or does GIF-SD use a strategy similar to the real guidance (RG) strategy in [22]?

2. The compared methods include DALL-E2, SD, and MAE, but how exactly these methods are used to generate samples are not clear. Have the authors tried their best to optimize the way these models generate samples, e.g., as does in [22] using language enhancement (LE) and CLIP Filter (CF)? This is important since it can faithfully reflect the proposed method's superiority.

3. I suggest the authors report the domain gap between the real image samples and the synthetic image samples, e.g., using FID or LPIPS. And the analysis of how the domain gap affects the performance is favored.

**Limitations:**

The authors are encouraged to discuss the limitations of the paper.

---

> ### Author Rebuttal · Authors · 2023-08-09
>
> Thanks for the insightful feedback, particularly for recognizing the significance of the studied task and our proposed method. We next address the concerns as follows.
>
> ---
>
> >**Q1. Effectiveness in boosting CLIP fine-tuning**
>
> Thanks for pointing this out. In fact, we have evaluated the effectiveness of our method in boosting CLIP fine-tuning. As shown in Table 15 (cf. Appendix F.4.3), our method significantly enhances the fine-tuning performance of CLIP VIT-B/32 on CIFAR100-S, elevating the accuracy from 75.2% to 79.4%.
>
> During the rebuttal, we further find that dataset expansion is also beneficial to the out-of-distribution robustness of the fine-tuned CLIP model, as shown in the following table.
>
> | CIFAR100-S dataset | Accuracy on CIFAR100-C |
> | ---------- |:---------------:|
> | Training from scratch on original dataset | 23.6 |
> | CLIP fine-tuning on original dataset | 55.4 (+31.8) |
> | CLIP fine-tuning on 5x-expanded dataset by GIF-SD | 61.4 (+37.8) |
>
> These results further underscore the potency of our method.
>
> ---
>
> >**Q2. Does GIF-SD use a strategy similar to the real guidance (RG) strategy in [22]?**
>
> Yes, GIF-SD is implemented with a strategy similar to the real guidance (RG) strategy in [22], which has been detailed in Appendix D.2, including the pseudo-code and other implementation details. To offer further clarity, we summarize the overall pipeline here. GIF-SD first conducts text-guided latent diffusion based on the latent feature of the seed data as the starting point (instead of random noise). After text-guided diffusion, GIF-SD conducts guided imagination based on Eqs. (1-2) and the image decoder. This ensures that the generated samples are class-maintained (cf. visualization results in Appendix G) and bring sufficient new information for boosting model performance (cf. Table 1 in Section 5.1).
>
> ---
>
> >**Q3. Comparison with the few-shot setting in [22]？**
>
> Following the constructive suggestion, we adopt the advanced few-shot strategies in [22] (i.e., language enhancement (LE) and CLIP Filter (CF)) to expand the CIFAR100-S dataset based on Stable Diffusion (SD) and real guidance (RG). As shown in the following table, SD combined with these strategies [22] is still noticeably inferior to our GIF-SD for both training from scratch and CLIP tuning. This further demonstrates the superiority of our method.
>
> | CIFAR100-S | Training from scratch | CLIP fine-tuning |
> | ---------- |:---------------:| :---------------:|
> | Original dataset | 35.0 | 75.2 |
> | 5x-expanded dataset by SD+LE+CF [22] | 55.1 (+20.1)| 77.0 (+1.8)
> | 5x-expanded dataset by GIF-SD (ours) | 61.1 (+26.1)| 79.4 (+4.2) |
>
> ---
>
> >**Q4. Implementation of DALL-E2,SD and MAE baselines**
>
> These baselines were set up identically to their GIF counterparts for ablated evaluation, with the only exception being the absence of the guided imagination optimization (cf. Section 4). To ensure a consistent comparison, we did not leverage other strategies, like LE and CF [22], for both the baselines and our methods.
>
> We agree with the reviewer that integrating these strategies might potentially enhance the quality of data generation. Nevertheless, as shown in the table responding to **the above Q3**, SD using these strategies still performs worse than our GIF-SD, which further verifies our superiority.
>
> ---
>
> >**Q5. Analyzing the relations between the domain gap and model performance**
>
> Thanks for the insightful suggestion. In response, we further compute the Fréchet Inception Distance (FID) between the synthetic data generated by different methods and the original data of CIFAR100-S. The results are summarized in the table below:
>
> | Datasets | FID | Accuracy |
> | ---------- |:---------:| :--------:|
> | Original CIFAR100-S dataset | - | 35.0 |
> | 5x-expanded dataset by RandAugment | 24.3 | 46.7 |
> | 5x-expanded dataset by Cutout | 104.7 | 44.3 |
> | 5x-expanded dataset by Gridmask | 104.8 | 48.2 |
> | 5x-expanded dataset by GIF-MAE | 72.3 | 52.7 |
> | 5x-expanded dataset by GIF-DALLE | 39.5 | 54.5 |
> | 5x-expanded dataset by GIF-SD | 81.7 | 61.1 |
>
> Interestingly, while one might initially assume that a lower FID implies better quality for the expanded data, the actual performance does not consistently follow this notion. For instance, even though GIF-SD has a worse FID than GIF-DALLE, it achieves better performance. Likewise, despite having nearly identical FIDs, Cutout and Gridmask lead to different performance. These results suggest that the effectiveness of dataset expansion methods depends on how much additional information and class consistency the generated data can provide to the original dataset, rather than the distribution similarity between those samples and the original data.
>
> In summary, we are grateful for the thought-provoking question. We believe this discussion will spark further research into the relationship between expansion effectiveness and data fidelity (as measured by metrics like FID), potentially guiding the development of even more effective dataset expansion techniques in the future.
>
> ---
>
> >**Q6. The authors are encouraged to discuss the limitations of the paper**
>
> Thanks for the valuable suggestion. We will enrich our supplementary with a limitation discussion. Due to the word constraints of this rebuttal, we kindly direct the reviewer to our detailed response to [Reviewer 6GPi's Q7](https://openreview.net/forum?id=82HeVCqsfh&noteId=CerYJBcKIX), where we delve into this discussion.
>
> ---
>
> Thanks again for the thoughtful feedback. We have diligently addressed each concern in our rebuttal, incorporating fresh experimental results and deeper discussions. The reviewer's insights have been pivotal in enriching our work, which aligns more closely with the conference's standards. We hope our updates address the concerns and might encourage a positive reconsideration of the initial assessment within the review panel.

---

> > ### Comment · Reviewer_1CUd · 2023-08-11
> > **Thanks for the rebuttal**
> >
> > After carefully reading the rebuttal, I think the authors have properly addressed my concerns. Consequently, I decided to raise my score accordingly.

---

> > > ### Author Response · Authors · 2023-08-11
> > >
> > > We deeply appreciate the thoughtful re-evaluation and recognition of our work's merit. The revised score and strong support boost our confidence and play a pivotal role in the review discussions. We believe that a champion like the reviewer, with a clear understanding of our research's strengths, can encourage the wider review panel to fully recognize and appreciate our contributions.

---

### Official Review · Reviewer_7xaH · 2023-07-26

**Soundness:** 2 fair
**Presentation:** 3 good
**Contribution:** 2 fair
**Rating:** 5
**Confidence:** 3

**Summary:**

This paper explores new dataset expansion task that solves data scarcity of small datasets while minimizing costs.
They leveraged generative models to create an automatic data generation pipeline.

**Strengths:**

- Dataset expansion is interesting and useful research topic for small-scale domain.
- The proposed methodology is simple and easy to utilize.
- Extensive experiments demonstrate the proposed method helps performance improvement in small benchmark as well as model generalization.

**Weaknesses:**

- My major concern is the novelty of the task. I agree that new sample generation can help classifier to be more robust. However, this task (dataset expansion) is not a novel task to me, and I don't think the experimental results are surprising or interesting. Recent works [A,B,C,D,E] for various computer vision tasks have explored generative modeling to solve the limited labeled data.


  - [A] W. Wu et al., "DiffuMask: Synthesizing Images with Pixel-level Annotations for Semantic Segmentation Using Diffusion Models"
  - [B] S. Azizi et al, "Synthetic Data from Diffusion Models Improves ImageNet Classification"
  - [C] G. Gu et al., "CompoDiff: Versatile Composed Image Retrieval With Latent Diffusion"
  - [D] Q. Kong et al., "Active Generative Adversarial Network for Image Classification"
  - [E] V. Sandfort et al., "Data augmentation using generative adversarial networks (CycleGAN) to improve generalizability in CT segmentation tasks"

- Although authors explain expansion efficiency in Fig.4 and Sec.5.1, generation time of diffusion model and conversion speed using GIF in training may be slow. So, I wonder the training efficiency in terms of cost and time.


**Questions:**

- I wonder the impact/synergy of the proposed method for pretrained imagenet models (like ViT), compared to the results in Table 1. Previous methods for data augmentation also explains their superiority under finetuning protocol.

- I guess the linear-probing or finetuning of CLIP will show comparable results for Table 2.


**Limitations:**

No.

---

> ### Author Rebuttal · Authors · 2023-08-09
>
> Thanks a lot for the comments, particularly for recognizing that the task is useful and the proposed method is simple and easy to use. We address the concerns as follows.
>
> ---
>
> >**Q1. Concern about the novelty of our work**
>
> Although expanding datasets is not a completely new concept, our work still presents valuable contributions to the community in the following aspects:
>
> - **Task importance**: Automatic dataset expansion, especially for small-data scenarios, bears great significance. While the concept might not be brand-new, our work provides a unique perspective on formally defining and tackling this task. The proposed dataset expansion can substantially reduce the time and costs involved in human data collection/annotation (cf. Section 5.2, Lines 343-350), and significantly enhances the model performance in small-data scenarios (cf. Table 1 in Section 5.1). The importance of this task has been highly recognized by other reviewers, such as Reviewer 1CUd "*the task is meaningful*", Reviewer UbXT "*the dataset expansion task could contribute to the development of academia and industry*" and Reviewer 6GPi "*the work has the potential to attract wide attention*".
>
> - **Method novelty**: Although using generative models to create training data is not a new idea, our method introduces a distinct design. The crux of our innovation lies in the concept of guided imagination (cf. Section 3.1), coupled with two critical expansion criteria (cf. Section 3.2). These insights are underpinned by both empirical observations and theoretical analysis (cf. Section 3, Section 5.2, Appendix B, Theorem 4.1). Based on these criteria, our approach can guide generative models to create informative new samples with novel content and correct class labels for expanding datasets. While the mentioned methods [A-E] also introduce new training data, they cannot ensure the synthesized data bring sufficient new information and accurate labels for the target small datasets. Moreover, training GANs from scratch, especially with very limited data, often fails to converge or produce meaningful results, making the method like [D,E] less effective in small-data scenarios. As such, our method emerges as a more effective way to expand small datasets. The contribution of our method has been highly recognized by Reviewer 1CUd "*the proposed framework is intuitive, and the criteria are well-motivated*", and Reviewer J4xT "This paper presents a promising solution".
>
> We hope that this clarification can illustrate the unique contributions of our work and address the concern adequately.
>
>
> [A] DiffuMask. Arxiv, 2023-03-21
> [B] Synthetic Data from Diffusion Models Improves ImageNet Classification. Arxiv, 2023-04-17
> [C] CompoDiff. Arxiv, 2023-03-21
> [D] Active Generative Adversarial Network for Image Classification. In AAAI, 2019
> [E] Data augmentation using CycleGAN. Scientific Reports, 2019
>
> ---
>
> >**Q2. Question about the cost and time**
>
> Thanks for the comment. Please refer to [General Response G1](https://openreview.net/forum?id=82HeVCqsfh&noteId=e7B6qvrwYr) for the detailed response.
>
> ---
>
> >**Q3. The effectiveness in model fine-tuning**
>
> Thanks for mentioning the potential synergy between dataset expansion and model fine-tuning. They indeed complement each other. As discussed in Appendix F.4.3 (cf. Table 15), our dataset expansion also improves the fine-tuning performance of pre-trained CLIP VIT-B/32 on CIFAR100-S. For clarity, we provide the related results below.
>
> | Methods | CIFAR100-S |
> | ---------- |:---------------:|
> | Training from scratch | 35.0 |
> | Zero-shot CLIP VIT-B/32 | 41.6 (+6.6) |
> | Fine-tuning CLIP VIT-B/32 on original dataset | 75.2 (+40.2) |
> | Fine-tuning CLIP VIT-B/32 on 5x-expanded dataset by GIF-SD | 79.4 (+44.4) |
>
> During the rebuttal, we further find that our dataset expansion also boosts the fine-tuning performance of models like ImageNet pre-trained ResNet-50 on Cars, as evident in the following table.
>
> | Methods | Cars |
> | ---------- |:---------------:|
> | Fine-tuning ImageNet pre-trained ResNet-50 on original dataset | 87.6 |
> | Fine-tuning ImageNet pre-trained ResNet-50 on 5x-expanded dataset by GIF-SD | 88.9 |
>
>
> In summary, our dataset expansion works effectively with model fine-tuning across various datasets and architectures, which further verifies our effectiveness.
>
> ---
>
> >**Q4. I guess the finetuning of CLIP will show comparable results for Table 2**
>
> Table 2 aims to fairly compare dataset expansion methods in the out-of-distribution (OOD) setting, so we train all models from scratch. We agree with the reviewer that fine-tuning a pre-trained CLIP, benefiting from its extensive pre-training on large datasets, would undoubtedly yield superior OOD performance. However, as discussed in the response to **the above Q3**, dataset expansion and model transfer are distinct yet complementary paradigms. They are not in competition, but rather can be synergistically combined. Therefore, our method can also enhance the OOD performance of fine-tuned CLIP models on CIFAR100-C, as demonstrated in the following table. Thanks again for the insightful feedback, and we will incorporate the new result into the revised manuscript.
>
> | CIFAR100-S dataset | OOD Accuracy on CIFAR100-C |
> | ---------- |:---------------:|
> | Training from scratch on original dataset | 23.6 |
> | Training from scratch on 5x-expanded dataset by GIF-SD | 43.3 (+19.7)|
> | CLIP fine-tuning on original dataset | 55.4 (+31.8) |
> | CLIP fine-tuning on 5x-expanded dataset by GIF-SD | 61.4 (+37.8) |
>
> ---
>
> We deeply value the reviewer's dedication to assessing our paper. Based on the constructive feedback, we have endeavored to address every concern and have incorporated new results and discussions to that end. We kindly ask the reviewer to re-evaluate our paper in light of these enhancements, which enable our work to better align with the conference's standards. Should additional concerns arise, we are glad to resolve them.

---

> > ### Comment · Reviewer_7xaH · 2023-08-14
> > **Thanks to the authors for rebuttal.**
> >
> > Thanks to the authors for responding to my questions.
> >
> > - I do not refute the importance of this task, but I want to emphasize that this task is not a novel task presented for the first time in this paper, and should be presented through sufficient comparison with missing related work.
> > - I agree the novelty of proposed method for guided imagination.
> >
> > I have additional questions for additional experiments with pre-trained model.
> >
> > - How does it compare to using other augmentation methods on pre-trained networks? The performance gap between training with original datasets and with GIF-SD seems to be very marginal compared to the scratch model.
> > - In practice, these days, most small data, zero-shot, or few-shot settings adopt a method of recycling pre-trained model parameters, and I think it would be good to add experimental results on this part to the main text.

---

> > > ### Author Response · Authors · 2023-08-14
> > >
> > > > **Q1: I do not refute the importance of this task, but I want to emphasize this task is not a novel task presented for the first time, and should be presented through sufficient comparison with missing related work. I agree the novelty of the proposed method.**
> > >
> > > We sincerely appreciate the follow-up comments, particularly in recognizing the importance of our task and the novelty of our method. While the concept of dataset expansion may not be entirely new, we believe the contributions the reviewer has recognized hold substantial value for the community.
> > >
> > > We agree with the reviewer that it is beneficial to provide a discussion about the missing related work [A-E]. Following the suggestion, we will incorporate the discussion on [A-E] (cf. the original rebuttal) into the "Related Work":
> > >
> > > - Moreover, recent methods [A-E] also explored generative models to generate new data for model training. However, these methods cannot ensure that the synthesized data bring sufficient new information and accurate labels for the target small datasets. Moreover, training GANs from scratch, especially with very limited data, often fails to converge or produce meaningful results, making the methods like [D,E] less effective in small-data scenarios. As such, our method emerges as a more effective way to expand small datasets.
> > >
> > > We believe this revision can further ensure a thorough contrast of our work with prior studies.
> > >
> > > ---
> > >
> > > > **Q2: Comparisons to augmentation methods in model fine-tuning. The performance gap seems to be marginal compared to the scratch model.**
> > >
> > > We thank the reviewer for the comment. Here, we first provide a detailed explanation for the modest performance gains in model fine-tuning.
> > >
> > > - As highlighted in our original rebuttal, both dataset expansion and pre-trained model fine-tuning address the small-data problem. That is, model pre-training on large-scale datasets can mitigate the challenges associated with limited data training, which naturally diminishes the headroom of performance improvement via dataset expansion. As a result, the performance gains realized by integrating dataset expansion with model fine-tuning are understandably less significant than those achieved when training from scratch.
> > >
> > > Following the suggestion, we provide additional results related to model fine-tuning.
> > >
> > > - **Superiority of our method over augmentation in model fine-tuning**: While the gains in model fine-tuning can be modest due to the aforementioned reasons, our method still delivers noticeable performance improvement. The table below shows that when expanding CIFAR100-S for CLIP model fine-tuning, our GIF-SD yields a significant advantage over both RandAugment and an advanced training data generation method [A] suggested by Reviewer **1CUd**. These results further verify the superiority of our approach.
> > >
> > > | CIFAR100-S | CLIP fine-tuning |
> > > | -- |:--:|
> > > | Original dataset | 75.2 |
> > > | 5x-expanded by RandAugment   | 77.7 (+2.5) |
> > > | 5x-expanded by SD+LE+CF [A]   | 77.0 (+1.8) |
> > > | 5x-expanded by GIF-SD (ours)  | 79.4 (+4.2) |
> > >
> > > - **Broader applicability of dataset expansion compared to model fine-tuning**: A salient advantage of our dataset expansion is its adaptability to different image domains, whereas model fine-tuning is largely constrained by the correlation between the pre-training and fine-tuning datasets. When there is a significant disparity in image nature  (e.g., from natural images to medical images), the effectiveness of fine-tuning diminishes. This is evident in the following table, where fine-tuning a CLIP pre-trained model on medical image datasets only shows modest improvements, lagging behind the gains from our dataset expansion.  Such a limitation in model fine-tuning was also observed in prior works like  [B]. This finding further pinpoints the importance of dataset expansion, especially in scenarios where suitable in-domain pre-trained models are not readily available.
> > >
> > > | Methods | PathMNIST |  BreastMNIST | OrganSMNIST|
> > > | -- |:--:| :---:| :--:|
> > > | Training from scratch on original dataset | 72.4 | 55.8 | 76.3 |
> > > | CLIP fine-tuning on original dataset | 78.4 (+6.0) | 67.2 (+11.4) | 78.9 (+2.6) |
> > > | Training from scratch on 5x-expanded dataset by GIF-SD | 86.9 (+14.5) | 77.4 (+21.6) | 80.7 (+4.4) |
> > >
> > > In sum, these new results further verify the superiority and effectiveness of our method. In light of this constructive suggestion, we will incorporate this discussion into Section 5.1.
> > >
> > > **Reference**:
> > > - [A] Is synthetic data from generative models ready for image recognition? In ICLR, 2023
> > > - [B] Transfusion: Understanding transfer learning for medical imaging. In NeurIPS, 2019
> > >
> > > ---
> > >
> > > We are deeply grateful for the reviewer's sustained engagement. Thanks to the reviewer's constructive suggestions, the overall quality of our paper has been further improved. In light of these improvements, we humbly request the reviewer to re-evaluate our paper. Should there be any further questions, we are glad to address them.

---

> > > > ### Comment · Reviewer_7xaH · 2023-08-15
> > > > **Thanks!**
> > > >
> > > > Although this method is simple, and lacks the paper in several ways noted by reviewers, I have found it helpful for performance even in finetuning setup.
> > > > So I raise my score.

---

> > > > > ### Author Response · Authors · 2023-08-16
> > > > >
> > > > > Thanks for recognizing the value of our work! The positive feedback has greatly boosted our confidence. We're deeply appreciative of the reviewer's continuous engagement and insightful suggestions, which have undeniably enhanced the quality of our work.

---

### Author Rebuttal · Authors · 2023-08-09

**General Response**
---

We deeply appreciate all the reviewers for dedicating time and effort to reviewing our paper. Here, we first address the general question below, and subsequently, we will provide detailed responses to each reviewer's specific questions and comments.

---

>**G1. Concern about the time and cost of dataset expansion (Reviewers 7xaH and J4xT)**

The primary goal of dataset expansion is to mitigate the time and cost of human data collection/annotation and boost model performance in small-data scenarios. Please note that improving model performance on small-scale datasets inevitably incurs certain costs. For instance, in the context of transfer learning, the total cost—including time and resources for collecting and pre-training on large-scale datasets (such as large-scale medical image datasets)—could be significantly high. In this work, we introduce a complementary task paradigm to mitigate the human labor and financial costs associated with data collection, and also bolsters model performance.

In this context, dataset expansion and model training are treated as two distinct phases, and we differentiate the analysis for each phase as detailed below.

**I. Dataset expansion**: As discussed in Section 5.2 (Lines 343-350), GIF-SD can expand an image by 5 times in just 33 seconds using a V100 GPU, so it can significantly save time and cost of dataset expansion compared to manual data collection/annotation. Specifically, manually annotating 10,000 images, according to Masterpiece Group (https://mpg-myanmar.com/annotation), would typically **take 2 weeks and cost around \$800**. In contrast, GIF-SD can generate the same volume of labeled data in a **mere 2 hours, costing roughly \$40 for renting 8 V100 GPUs**. Moreover, if higher efficiency is pursued with an acceptable performance drop, GIF-MAE can create 10,000 labeled data in **just 80 seconds, at a cost of about \$0.48 for renting 8 V100 GPUs**. Note that once the dataset is expanded, it can be directly utilized to train various models, removing the need for regeneration with each model and thereby further enhancing efficiency.

| Methods | Expansion speed | Time (10,000 images) | Costs (10,000 images) |
| ---------- |:---------------:| :---------------:| :---------------:|
| Human data collection | 120.96s per image | 2 weeks | $800 |
| GIF-MAE (ours) | 0.008s per image | 80 seconds | $0.48 |
| GIF-SD (ours) | 6.6s per image | 2 hours | $40 |

**II. Model training**: The training time varies based on the specific datasets. However, it is pivotal to note that all dataset expansion methods were compared based on the same expansion ratio, thus ensuring consistent training time/cost and fair comparisons. We acknowledge that training on an expanded dataset will inevitably take longer than training on the original dataset. However, as shown in Table 1 (cf. Section 5.1), the significant improvement in model performance (i.e., by 36.9% on average over six natural image datasets and by 13.5% on average over three medical datasets) makes the increased investment in training time worthwhile. More importantly, a detailed analysis presented in Appendix F.1.3 (cf. Table 10) shows that, even with the same training consumption (in terms of sample number × training epochs), our proposed method still proves advantageous. More specifically, as shown in the following table, training the model on the original CIFAR100-S dataset for 5x more epochs performs much worse than the model trained on our 5x-expanded dataset. This comparison further underscores the effectiveness of our method in achieving higher accuracy without inflating training costs.


| CIFAR100-S | Training epoch |    Consumption (data number x epoch)    | Accuracy |
| ---------- |:---------------:|  :---------------:  | :---------------:|
| Training on original dataset | 100 | 1 million | 35.0 |
| Training on original dataset with RandAugment | 100 | 1 million |39.6 |
| Training on original dataset with RandAugment | 600 | 6 million| 51.1 |
| Training on 5x-expanded dataset by GIF-SD (ours) | 100 | 6 million| 61.1 |

To summarize, based on the comprehensive analysis, we firmly believe that the efficiency and cost-effectiveness of our method outweigh the concerns related to computational consumption.


---

>**G2. Question on potential influences of the synthetic data (Reviewer J4xT)**

Ethical considerations, especially in AI research and data generation, are indeed paramount. Given the importance of this topic, we have opted to address it in the general response, even though it was specifically raised by Reviewer J4xT. In fact, our approach is constructed with care to avoid negative ethical implications, as evidenced in the following points:

- **Controlled generation**: In our approach, the generation of synthetic data is driven by our expansion guidances, which ensure that new data is derived directly and meaningfully from the original dataset. This controlled mechanism minimizes the risks of creating unrelated or potentially harmful images.

- **No personal or sensitive data**: It is also worth noting that our method primarily focuses on public available datasets like CIFAR, Stanford Cars, and similar, which do not contain personal or sensitive information. As such, the risks related to privacy breaches or misrepresentations are substantially diminished.

- **Ethical commitment**: While our research aims to showcase the technical capabilities of our method, we agree with the significance of upholding ethical standards in technology. In future applications, if applying our method to more sensitive domains, we commit to seek ethical evaluations to ensure responsible use.

We hope these clarifications underscore our commitment to responsible AI and address the concern.

---

### Author Response · Authors · 2023-08-20

We deeply appreciate all reviewers’ time and efforts in reviewing our paper and providing constructive feedback. Besides the detailed response to each reviewer, here we would like to  1) further thank the reviewers for their recognition of our work; 2) summarize the main contributions of our work; and 3) highlight the new results added during the rebuttal.

**(1) We are glad that the reviewers appreciate and recognize our key contributions**.

- Dataset expansion is an interesting and useful research topic for small-scale domains [**7xaH**]
- The dataset expansion task has the potential to attract wide attention, and could contribute to the development of academia and industry [**UbXT,6GPi**]
- The task is meaningful. This paper may be one of the first studies to explore the effectiveness of synthetic images in boosting classification performance [**1CUd**]
- The proposed methodology is simple and easy to utilize, and the method for guided imagination is novel [**7xaH**]
- The proposed framework is intuitive, and the criteria to maintain the class information and to encourage diversity are well-motivated [**1CUd,6GPi**]
- This paper presents a promising solution to the challenge of generating reliable supplementary training data, distinguishing itself from conventional data augmentation methods [**J4xT**]
- The experimental evaluations and ablation studies are quite extensive and informative. Extensive experiments demonstrate the proposed method helps improve performance in small benchmark, model generalization and model finetuning [**7xaH,1CUd,UbXT,J4xT,6GPi**]


**(2) We summarize our main contributions below**.

- **Important task**. Automatic dataset expansion, especially in small-data scenarios, holds immense importance. Our work provides a unique perspective on formally defining and tackling this task. The proposed dataset expansion significantly reduces human efforts and expenses associated with manual data collection (cf. Section 5.2, Lines 343-350), and markedly improves the model performance in small-data scenarios (cf. Table 1 in Section 5.1). These benefits are highly important for real-world applications, since manual data collection is highly expensive in small-data scenarios  (e.g., medical image domains).
- **Novel method and Informative insights**. Although using generative models to create training data is not a new idea, our method introduces a distinct design. The cornerstone of our innovation is the concept of guided imagination (cf. Section 3.1), coupled with two critical expansion criteria (cf. Section 3.2). These insights, including the two pivotal expansion criteria, sample-wise expansion, channel-level noise optimization, are solidly rooted in both empirical observations and theoretical analysis (cf. Section 3, Section 5.2, Appendix B, Theorem 4.1).  Based on these criteria and insights, our approach can guide generative models to create informative new samples with novel content and correct class labels for effective dataset expansion. In contrast, existing methods cannot ensure the synthesized data bring sufficient new information and accurate labels for the target small datasets. As such, our method emerges as a more effective way to expand small datasets in real applications.
- **Extensive experiments**. Extensive and systematic experiments have demonstrated the effectiveness and superiority of our method across both natural and medical images, and in various scenarios, including small-data tasks, long-tailed recognition, model fine-tuning, and out-of-distribution generalization (cf. Section 5.1 and Appendix F).

**(3) In this rebuttal, we have added more supporting results following the reviewers’ suggestions**.


- More results in the fine-tuning setting [**7xaH,1CUd,J4xT**]
- Comparison results with a more advanced generative method [**7xaH,1CUd,J4xT**]
- Comparison results in terms of time cost across various methods and human data collection [**7xaH,J4xT**]
- Out-of-distribution results of CLIP model fine-tuning on CIFAR100-C  [**7xaH**]
- Relation analysis between the domain gap (in terms of FID) and model performance [**1CUd**]
- Safety check of the synthetic images with empirical assessments  [**J4xT**]

As the author-reviewer discussion draws to a close, we would like to know if there are any last-minute comments or questions we can address. Thanks again for the constructive feedback.

---

### Decision · Program_Chairs · 2023-09-21

**Decision:**

Accept (poster)

**Comment:**

This paper initially received mixed ratings, and the authors submitted rebuttals. Reviewers read the rebuttals and further discussed with the authors. Most reviewers were satisfied by the authors' rebuttals and responses. As a result, all reviewers except one raised their ratings. The AC read the paper, the authors' rebuttals and the discussions between authors and reviewers. The AC agrees with the majority of the reviewers on their final ratings, and the decision is to accept.